# What Improves the Generalization of Graph Transformer? A Theoretical Dive into Self-Attention and Positional Encoding

## Abstract

Graph Transformers, which incorporate self-attention and positional encoding, have recently emerged as a powerful architecture for various graph learning tasks. Despite their impressive performance, the complex non-convex interactions across layers and the recursive graph structure have made it challenging to establish a theoretical foundation for learning and generalization. This study introduces the first theoretical investigation of a shallow Graph Transformer for semi-supervised node classification, comprising a self-attention layer with relative positional encoding and a two-layer perception. Focusing on a graph data model with discriminative nodes that determine node labels and non-discriminative nodes that are class-irrelevant, we characterize the sample complexity required to achieve a zero generalization error by training with stochastic gradient descent (SGD). This paper provides the quantitative characterization of the sample complexity and number of iterations for convergence dependent on the fraction of discriminative nodes, the dominant patterns, the fraction of erroneous labels, and the initial model errors. Furthermore, we demonstrate that self-attention and positional encoding enhance generalization by making the attention map sparse and promoting the core neighborhood during training, which explains the superior feature representation of Graph Transformers. Our theoretical results are supported by empirical experiments on synthetic and real-world benchmarks.

## 1 Introduction

Graph Transformer (Dwivedi & Bresson, 2021; Kreuzer et al., 2021; Ying et al., 2021) was developed for graph machine learning as a response to the impressive performance of Transformers demonstrated in various domains (Vaswani et al., 2017; Kenton & Toutanova, 2019; Brown et al., 2020; Dosovitskiy et al., 2020; Chen et al., 2019). It is designed specifically to handle graph data by constructing positional embeddings that capture important graph information and using nodes as input tokens for the Transformer model. Empirical results have shown that Graph Transformers (GT) outperform classical graph neural networks (GNN), such as graph convolutional networks (GCN), in graph-level learning tasks such as molecular property prediction (Rong et al., 2020; Kreuzer et al., 2021; Wu et al., 2021), image classification (Gabrielsson et al., 2022; Rampášek et al., 2022), as well as node-level tasks like document analysis (Zhang & Zhang, 2020; Hu et al., 2020c;b; ZHANG et al., 2022; Chen et al., 2023), semantic segmentation (Rampášek et al., 2022; Hussain et al., 2022), and social network analysis (Zhao et al., 2021; Dwivedi & Bresson, 2021; Chen et al., 2022).

Despite the notable empirical advancements, some critical theoretical aspects of Graph Transformers remain much less explored. These include fundamental inquiries such as:

- *Under what conditions can a Graph Transformer achieve adequate generalization?*
- *What is the advantage of self-attention and positional encoding in graph learning?*

Some recent works (Ying et al., 2021; Chen et al., 2023) theoretically study GTs by comparing their expressive power with other graph neural networks without self-attention. Meanwhile, other studies (Kreuzer et al., 2021; Rampášek et al., 2022; Gabrielsson et al., 2022) explain the design of PE in terms of graph topology and spectral theory. However, these analyses only establish the existence of a desired GT model, rather than its achievability through practical learning methods. Additionally,

none of the existing works have theoretically examined the generalization of GTs, which is essential to explain their superior performance and guide the model and algorithm design.

To the best of our knowledge, this paper presents the first learning and generalization analysis of a basic shallow GT trained using stochastic gradient descent (SGD). We focus on a semi-supervised binary node classification problem on structured graph data, where each node feature corresponds to either a discriminative or a non-discriminative pattern, and each ground truth node label is determined by the dominant discriminative pattern in the core neighborhood. We explicitly characterize the required number of training samples, i.e., the sample complexity, and the number of SGD iterations to achieve a desired generalization error. Our sample complexity bound indicates that graphs with a larger fraction of discriminative nodes tend to have superior generalization performance. Moreover, our analysis reveals that better generalization performance can be achieved by using graph sampling methods that prioritize class-relevant nodes. Our **technical contributions** are highlighted below:

**First, this paper establishes a novel framework for the optimization and generalization analysis of shallow GTs.** We consider a shallow GT model with non-convex interactions across layers, including learnable self-attention and PE parameters, and Relu, softmax activation functions, while the state-of-the-art works on GNNs (Maskey et al., 2022; Tang & Liu, 2023; Zhang et al., 2023b) exclude attention layers due to such difficulties. This paper develops a novel and extendable feature-learning framework for analyzing the optimization and generalization of GTs.

**Secondly, this paper theoretically characterizes the benefits of the self-attention layer of GTs.** Our analysis shows that self-attention evolves in a way that promotes class-relevant nodes during training. Thus, a GT trained produces a sparse attention map. Compared with GCNs without self-attention, GTs have a lower sample complexity and faster convergence rate for better generalization.

**Third, this paper theoretically demonstrates that positional embedding improves the generalization by promoting the nodes in the core neighborhood**. Different from the state-of-the-art theoretical studies on Transformers that either ignore PE in analyzing generalization (Li et al., 2023a; Tian et al., 2023; Tang & Liu, 2023) or only characterize the expressive power of PE (Rampášek et al., 2022; Gabrielsson et al., 2022), this paper analyzes the generalization of a GT with a trainable relative positional embedding and proves that, with no prior knowledge, positional embedding trained with SGD can identify and promote the core neighborhood. This, in turn, leads to fewer training iterations and a smaller sample complexity.

## 2 RELATED WORKS

**Theoretical study on GTs.** Previous research has applied tools of topology theory, spectral theory, and expressive power to explain the success of GTs. For example, Ying et al. (2021); Chen et al. (2023) illustrates that proper weights of the Transformer layer can represent basic operations of popular GNN models and capture more multi-hop information. Rampášek et al. (2022) explains the necessity of PEs in distinguishing links that cannot be learned by 1-Weisfeiler-Leman test (Weisfeiler & Leman, 1968). Kreuzer et al. (2021); Gabrielsson et al. (2022) depict that the PE can measure the physical interactions between nodes and reconstruct the raw graph as a bijection.

**Theoretical analyses of GNNs.** The works in (Xu et al., 2019; Cong et al., 2021; Zhang et al., 2023a) characterize the expressive power of GNNs by studying the Weisfeiler-Leman test, inter-nodal distances, and graph biconnectivity. Verma & Zhang (2019); Cong et al. (2021); Zhou & Wang (2021) analyze the stability of training GCNs. References (Liao et al., 2021; Garg et al., 2020; Oono & Suzuki, 2020; Zhang et al., 2020b) characterize the generalization gap via concentration bound for transductive learning or dependent variables. In (Li et al., 2022) and (Zhang et al., 2023b), the authors explore the generalization of GNNs with node sampling.

**Learning neural networks on structured data.** Shi et al. (2021); Brutzkus & Globerson (2021); Allen-Zhu & Li (2022); Zhang et al. (2023b); Chowdhury et al. (2023) study one-hidden-layer fully-connected networks or convolutional neural networks given data containing discriminative and background patterns. This framework is extended to self-supervised learning and ensemble learning (Wen & Li, 2021; 2022; Allen-Zhu & Li, 2023). The learning and generalization of one-layer single-head Vision Transformers are studied in (Jelassi et al., 2022; Li et al., 2023a; Oymak et al., 2023; Li et al., 2023b) based on the spatial or pattern-space association between tokens.

## 3 PROBLEM FORMULATION AND LEARNING ALGORITHM

Let $\mathcal{G} = (\mathcal{V}, \mathcal{E})$ denote an un-directed graph, where $\mathcal{V}$ is the set of nodes with size $|\mathcal{V}| = N$ and $\mathcal{E}$ is the set of edges. $\boldsymbol{X} \in \mathbb{R}^{d \times N}$ denotes the matrix of the features of $N$ nodes, where the $n$-th column of $\boldsymbol{X}$, denoted by $\boldsymbol{x}_n \in \mathbb{R}^d$, represents the feature of node $n$. Assume $\|\boldsymbol{x}_n\| = 1$ for all nodes without loss of generality. We study a binary node classification problem [1]. The label of node $n$ is $y_n \in \{+1, -1\}$. Let $\mathcal{L} \subset \mathcal{V}$ denote the set of labeled nodes. Given $\boldsymbol{X}$ and labels in $\mathcal{L}$, the objective of semi-supervised learning for node classification is to predict the unknown labels in $\mathcal{V} - \mathcal{L}$. The learning process is implemented on a basic one-layer Graph Transformer in (1)[2], which includes a single-head self-attention layer and a two-layer perception with a relative positional embedding.

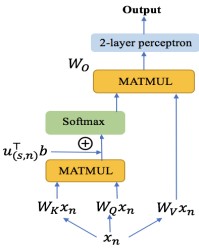

Figure 1: Graph Transformer in (1)

$$F(\boldsymbol{x}_n) = \boldsymbol{a}^\top \mathrm{Relu}(\boldsymbol{W}_O \sum_{s \in \mathcal{T}^n} \boldsymbol{W}_V \boldsymbol{x}_s \mathrm{softmax}_n((\boldsymbol{W}_K \boldsymbol{x}_s)^\top \boldsymbol{W}_Q \boldsymbol{x}_n + \boldsymbol{u}_{(s,n)}^\top \boldsymbol{b}))$$

(1)

where $\boldsymbol{x}_n, \boldsymbol{x}_s \in \mathbb{R}^d$ and $\mathcal{T}^n$ is the set of nodes for the aggregation computation of node $n$. $\mathrm{softmax}_n(g(s, n)) = \exp(g(s, n)) / \sum_{j \in \mathcal{T}^n} \exp(g(j, n))$ if we denote $g(s, n) = \boldsymbol{x}_s^\top \boldsymbol{W}_K^\top \boldsymbol{W}_Q \boldsymbol{x}_n + \boldsymbol{u}_{(s,n)}^\top \boldsymbol{b}$. $\boldsymbol{W}_K \in \mathbb{R}^{m_a \times d}$, $\boldsymbol{W}_Q \in \mathbb{R}^{m_a \times d}$, and $\boldsymbol{W}_V \in \mathbb{R}^{m_b \times d}$ are key, query, and value parameters to compute the self-attention representation by multiplying $\boldsymbol{X}$. $\boldsymbol{W}_O \in \mathbb{R}^{m \times m_b}$ and $\boldsymbol{a} \in \mathbb{R}^m$ are the hidden and output weights in the two-layer feedforward network. We define the one-hot distance vector $\boldsymbol{u}_{(s,n)} \in \mathbb{R}^Z$, where the non-zero index reflects the *truncated distance* between nodes $s$ and $n$. It is an indicator of the shortest-path distance (SPD) between nodes. Then,

$$\boldsymbol{u}_{(s,n)} = \begin{cases} \boldsymbol{e}_i, & \text{if the SPD of } s \text{ and } n \text{ equals } i - 1 \text{ and } i \leq Z, \\ \boldsymbol{e}_Z, & \text{if the SPD of } s \text{ and } n \text{ equals } i - 1 \text{ and } i > Z, \end{cases}$$

(2)

where $\boldsymbol{e}_i$ is the $i$-th standard basis in $\mathbb{R}^Z$. This architecture originates from (Vaswani et al., 2017) and is widely used in (Kreuzer et al., 2021; Zhao et al., 2021; ZHANG et al., 2022; Rampášek et al., 2022) for node classification on graphs. The PE $\boldsymbol{u}_{(s,n)}^\top \boldsymbol{b}$ is motivated by (Ying et al., 2021; Rampášek et al., 2022; Gabrielsson et al., 2022; Wu et al., 2022; Zhang et al., 2023c), which is one of the most commonly used PEs in GTs. [3]

Denote $\psi = (\boldsymbol{a}, \boldsymbol{W}_O, \boldsymbol{W}_V, \boldsymbol{W}_K, \boldsymbol{W}_Q, \boldsymbol{b})$ as the set of parameters to train. The semi-supervised learning problem solves the following empirical risk minimization problem $f_N(\psi)$,

$$\min_{\psi} : f_N(\psi) = \frac{1}{|\mathcal{L}|} \sum_{n \in \mathcal{L}} \ell(\boldsymbol{x}_n, y_n; \psi), \quad \ell(\boldsymbol{x}_n, y_n; \psi) = \max\{1 - y_n \cdot F(\boldsymbol{x}_n), 0\}, \quad (3)$$

where $\ell(\boldsymbol{x}_n, y_n; \psi)$ is the Hinge loss function. Assume $(\boldsymbol{x}_n, y_n)$ are identically distributed but *dependent* samples drawn from some unknown distribution $\mathcal{D}$. The sample dependence results from the dependence of node labels on neighboring node features. The test/generalization performance of a learned model $\psi$ is evaluated by the population risk $f(\psi)$, where

$$f(\psi) = f(\boldsymbol{a}, \boldsymbol{W}_O, \boldsymbol{W}_V, \boldsymbol{W}_K, \boldsymbol{W}_Q, \boldsymbol{b}) = \mathbb{E}_{(\boldsymbol{x}, y) \sim \mathcal{D}}[\max\{1 - y \cdot F(\boldsymbol{x}), 0\}]. \quad (4)$$

**Training Algorithm**: The training problem (3) is solved via a mini-batch stochastic gradient descent (SGD), as summarized in Algorithm 1. At each iteration $t$, the gradient is computed using a mini-batch $\mathcal{B}_t$ with $|\mathcal{B}_t| = B$ and step size $\eta$ with all parameters in $\psi$ except $\boldsymbol{a}$. At iteration $t$, we uniformly sample a subset $\mathcal{S}^{n,t}$ of nodes from the whole graph for aggregation of each node $n$.

Following the framework "pre-training & fine-tuning" for node classification using (Zhang et al., 2020a; Zhang & Zhang, 2020; Hu et al., 2020b; Liu et al., 2021), we set $\boldsymbol{W}_V^{(0)}$, $\boldsymbol{W}_Q^{(0)}$, and $\boldsymbol{W}_K^{(0)}$ come from an initial model. Every entry of $\boldsymbol{W}_O^{(0)}$ is generated from $\mathcal{N}(0, \xi^2)$. Every entry of $\boldsymbol{a}^{(0)}$ is sampled from $\{+1/\sqrt{m}, -1/\sqrt{m}\}$ with equal probability. $\boldsymbol{b}^{(0)} = \boldsymbol{0}$. $\boldsymbol{a}$ is fixed during the training [4].

---

[1] Extension to graph classification and multi-classification is briefly discussed in Appendix E.4 and E.5.

[2] Since the queries and keys are normalized, we remove the $\sqrt{m_a}$ scaling in the softmax function as in (Li et al., 2023a; Tian et al., 2023; Tarzanagh et al., 2023; Oymak et al., 2023).

[3] As the first work on the generalization of GT, we mainly study this PE for simplicity of the presentation. The analytical framework is extendable to GTs with other PEs. We briefly introduce the formulation and analysis of absolute PE, such as Laplacian vectors and node degree, in Appendix E.2.

[4] It is common to fix the output layer weights as the random initialization in the theoretical analysis of neural networks, including NTK (Allen-Zhu et al., 2019; Arora et al., 2019) and feature learning (Karp et al., 2021;

## 4 THEORETICAL RESULTS

### 4.1 THEORETICAL INSIGHTS

Before formally introducing our data model in Section 4.2 and the formal theoretical results in Section 4.3, we first summarize our key insights. We consider a data model where node features are noisy versions of *discriminative* patterns that directly determine the node labels and *non-discriminative* patterns that do not affect the labels. $\gamma_d$ is the fraction of discriminative nodes, $\tau$ is the noise level in node features. The node labels are determined by a majority vote of discriminative patterns in a so-called *core neighborhood*. A small $\epsilon_S$ corresponds to a clear-cutting vote in sampled nodes in the core neighborhood. $\sigma$ and $\delta$ are the initial model error. $p$ is the fraction of errors in observed labels.

**(P1). A new theoretical framework of a convergence and generalization analysis using SGD for GT.** This paper develops a new framework to analyze GTs based on a more general graph data model than existing works like (Zhang et al., 2023b). We show that with a proper initialization, the learning model converges with zero generalization error. The sample complexity bound is linear in $\gamma_d^{-2}$, $(\Theta(1) - \epsilon_S)^{-2}$. The required number of iterations is proportional to $(1 - 2p)^{\frac{3}{5}}$ and $(\Theta(1) - \delta - \tau)^{-1}$. The result indicates that a larger fraction of discriminative nodes and a smaller confusion ratio improve the sample complexity. A smaller fraction of label errors and smaller pattern/embedding noises accelerate the convergence.

**(P2). Self-attention helps GTs perform better than Graph convolutional networks**. We theoretically illustrate that the attention weights, i.e., softmax values of each node in the self-attention module, become increasingly sparse during the training and are concentrated at discriminative nodes. GTs can then learn more distinguishable representations for different classes, outperforming GCNs.

**(P3) Positional embedding promotes the core neighborhood.** We prove that starting from zero initialization, the positional embedding eventually finds the core neighborhood and assigns nodes in the core neighborhood with higher weights, which improves the generalization.

### 4.2 DATA MODEL ASSUMPTIONS

Each node feature $\boldsymbol{x}_n$ is a noisy version of one of $M$ ($2 \leq M < m_a, m_b$) distinct patterns $\{\boldsymbol{\mu}_1, \boldsymbol{\mu}_2, \cdots, \boldsymbol{\mu}_M\}$ in $\mathbb{R}^d$ with noise level $\tau$, i.e., $\min_{j \in [M]} \|\boldsymbol{x}_n - \boldsymbol{\mu}_j\| \leq \tau, \forall n \in \mathcal{V}$. $\boldsymbol{\mu}_1$ and $\boldsymbol{\mu}_2$ are two *discriminative patterns* that correspond to the label 1 and $-1$, respectively All other patterns $\boldsymbol{\mu}_3, \boldsymbol{\mu}_4, \cdots, \boldsymbol{\mu}_M$ are referred to as *non-discriminative patterns* that do not determine the labels. Let $\kappa = \min_{1 \leq i \neq j \leq M} \|\boldsymbol{\mu}_i - \boldsymbol{\mu}_j\| > 0$ denote the minimum distance between different patterns. We assume $\kappa \geq 4\tau$. Denote the set of nodes that are noisy versions of $\boldsymbol{\mu}_l$ as $\mathcal{D}_l$, $l \in [M]$, and $\cup_{l=1}^M \mathcal{D}_l = \mathcal{V}$. Let $\gamma_d = |\mathcal{D}_1 \cup \mathcal{D}_2|/|\mathcal{V}| = \Theta(1)$ represent the fraction of nodes that contain discriminative patterns[5]. We assume the dataset is balanced, i.e., the gap between the numbers of positive and negative labels is at most $O(\sqrt{N})$.

If node $n$ has the ground truth label $\tilde{y}^n = 1$, the nodes in $\mathcal{D}_1$ are are called *class-relevant* nodes for node $n$, and nodes in $\mathcal{D}_2$ called *confusion* nodes for node $n$. Conversely, if $\tilde{y}^n = -1$, $\mathcal{D}_2$ and $\mathcal{D}_1$ are class-relevant and confusion nodes for node $n$, respectively. We use notations $\mathcal{D}_*^n$ and $\mathcal{D}_\#^n$ for the class-relevant and confusion nodes for node $n$ without specifying $\mathcal{D}_1$ and $\mathcal{D}_2$. We define *distance-z neighborhood* of node $n$, denoted by $\mathcal{N}_z^n$, as the set of nodes that are away from node $n$ with distance $z$. The average winning margin of each node $n$ and the *core* distance $z_m$ are defined as follows.

**Definition 1.** *The winning margin for each node $n$ of distance-z and the average winning margin for all the nodes of distance-z are defined as*

$$\Delta_n(z) = |\mathcal{D}_*^n \cap \mathcal{N}_z^n| - |\mathcal{D}_\#^n \cap \mathcal{N}_z^n|, \quad \bar{\Delta}(z) = \frac{1}{N} \sum_{n \in \mathcal{V}} \Delta_n(z), \tag{5}$$

*for any $z \in [Z - 1]$. The core distance is defined as*

$$z_m = \arg \max_{z \in [Z-1]} \bar{\Delta}(z). \tag{6}$$

Allen-Zhu & Li, 2022; Li et al., 2023a) type of approaches. The optimization problem of $\boldsymbol{W}_Q, \boldsymbol{W}_K, \boldsymbol{W}_V, \boldsymbol{W}_O$, and $\boldsymbol{b}$ with non-linear activations is still highly non-convex and challenging.

[5]The ground truth label of each node $n \in \mathcal{V}$ follows a categorical distribution with probability $(\nu_1, \nu_2, \cdots, \nu_M)$, where $\nu_1 = \nu_2 = \gamma_d/2$ and $\nu_3 + \nu_4 + \cdots + \nu_M = 1 - \gamma_d$

**Assumption 1.** $\Delta_n(z_m) > 0$, *for all nodes* $n$.

Figure 2 provides an example of winning margin. Assumption 1 indicates that the ground-truth label $\tilde{y}^n$ for every node $n \in \mathcal{V}$ is consistent with a majority voting of $\boldsymbol{\mu}_1$ and $\boldsymbol{\mu}_2$ patterns in the core neighborhood $\mathcal{N}_{z_m}$, i.e., if $\tilde{y}^n = 1$ (or $\tilde{y}^n = -1$), then there are more nodes that correspond to $\boldsymbol{\mu}_1$ (or $\boldsymbol{\mu}_2$) in $\mathcal{N}_{z_m}^n$. We also assume $|\mathcal{N}_{z_m}^n|$ not too small to facilitate the sampling. We consider the general setup that the observed labels $\{y_n\}_{n \in \mathcal{L}}$ contain errors with the error fraction $p$. We set $|\mathcal{N}_{z_m}^n| \geq N/\text{poly}(Z)$ for all $n$ to avoid a trivial size of the core neighborhood.

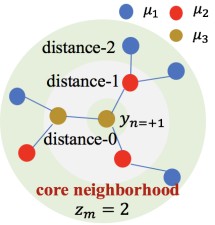

Assumption 2 in Appendix B requires the pre-trained model maps the query, key, and value embeddings to be close to orthogonal vectors with an error of $\sigma < O(1/M)$ for queries and keys and $\delta < 0.5$ for values. It is the same as Assumption 1 in (Li et al., 2023a). Such assumptions on the orthogonality of embeddings or data are widely employed in state-of-the-art generalization analysis for Transformers (Oymak et al., 2023; Tian et al., 2023). [6]

Figure 2: Example of the winning margin. Node $n$ has a non-discriminative feature $\boldsymbol{\mu}_3$ and label +1. Then $\Delta_n(1) = -2$, and $\Delta_n(2) = 3$.

### 4.3 Main Theoretical Results for Graph Transformer

Table 1: Some important notations

| | | | |
|---|---|---|---|
| $\mathcal{V}$ | The set of all the nodes | $\mathcal{D}_*^n, \mathcal{D}_\#^n$ | Sets of class-relevant nodes and confusion nodes for node $n$ |
| $\mathcal{L}$ | The set of labeled nodes | $\mathcal{T}^n$ | The set of nodes for aggregation for $n$ |
| $\mathcal{D}_l$ | The set of nodes of the pattern $\boldsymbol{\mu}_l$ | $\mathcal{S}_*^{n,t}, \mathcal{S}_\#^{n,t}$ | Sampled class-relevant and confusion nodes out of $\mathcal{T}^n$ at iteration $t$ |
| $\gamma_d$ | The fraction of discriminative nodes | $\overline{\Delta}(z)$ | Average winning margin of all nodes at the distance-$z$ neighborhood |
| $\mathcal{N}_z^n$ | Distance-$z$ neighborhood of node $n$ | $z_m$ | The core distance that has the largest winning margin |
| $\epsilon_\mathcal{S}$ | confusion ratio, the average fraction of confusion nodes in sampled nodes of distance-$z_m$ neighborhood | | |

We define *confusion ratio* $\epsilon_\mathcal{S}$ as the average fraction of confusion nodes in the distance-$z_m$ neighborhood over all iterations and all labeled nodes. Some notations are summarized in Table 1.

**Definition 2.** *The confusion ratio* $\epsilon_\mathcal{S}$ *is*

$$\epsilon_\mathcal{S} = \mathbb{E}_{t \geq 0, n \in (\cup_{l=3}^M \mathcal{D}_l) \cap \mathcal{L}} (|\mathcal{S}_\#^{n,t} \cap \mathcal{N}_{z_m}^n| / |(\mathcal{S}_*^{n,t} \cup \mathcal{S}_\#^{n,t}) \cap \mathcal{N}_{z_m}^n|), \quad (7)$$

*where* $\mathcal{S}_*^{n,t}$ *and* $\mathcal{S}_\#^{n,t}$ *denote the sampled class-relevant and confusion nodes in* $\mathcal{T}^n$ *for node* $n$ *in training iteration* $t$, *respectively.*

We then introduce our major theoretical results.

**Theorem 4.1.** *(Generalization Guarantee of Graph Transformers) As long as* $\tau \leq \min(\sigma, \delta)$; *and a model with* $m$ *at least* $M^2 \log N$ *for* $\epsilon \in (0, 1/2)$, *and the mini-batch size* $B$ *and the number of sampled nodes* $|\mathcal{S}^{n,t}|$ *for each iteration* $t$ *larger than* $\Omega(1)$. *Then, after* $T$ *iterations such that*

$$T = \Theta(\eta^{-1/2}(1 - 2p)^{-1/2}(1 - \delta - \tau)^{-1/2}), \quad (8)$$

*as long as the number of known labels satisfies*

$$|\mathcal{L}| \geq \max\{\Omega((1 - 2\epsilon_\mathcal{S}(1 - \gamma_d) - (\sigma + \tau))^{-2}(1 + \delta_{z_m}^2) \cdot \log N), BT\}, \quad (9)$$

*where* $\delta_{z_m} = \max_{n \in \mathcal{V}} |\mathcal{N}_{z_m}^n|$ *measures the maximum number of nodes in distance-$z_m$ neighborhood, for some* $\epsilon_\mathcal{S} \in (0, 1/2)$ *and* $p \in (0, 1/2)$, *then with a probability of at least* $0.99$, *the returned model trained by Algorithm 1 achieves zero generalization error as* $f(\psi) = 0$.

**Remark 1.** (Generalization improvement by good graph properties) The first term in (9) dominates when $p$ is not very close to[7] $1/2$, i.e., the fraction of erroneous training labels is small. Then the

---

[6] We conduct experiments to verify the existence of discriminative nodes and the core neighborhood with four real datasets in Appendix A.1. We also show Assumption 1 and 2 are not strong by comparing existing works in Appendix E.1.

[7] The exact condition is when $p < 1/2 - \delta_{z_m}^{-4}/2$.

sample complexity in (9) scales with $1/\gamma_d^2$, $(1 - \epsilon_{\mathcal{S}})^{-2}$ and $(\Theta(1) - \sigma - \tau)^{-2}$. Hence, a larger fraction of nodes of discriminative patterns (a larger $\gamma_d$), a smaller fraction of confusion patterns in the core neighborhood (a smaller $\epsilon_{\mathcal{S}}$), a smaller pattern noise and embedding noise (a smaller $\tau$ and $\sigma$) can reduce the sample complexity. The required number of iterations also reduces with a smaller fraction of label errors $p$ and the pattern and embedding noises $\tau$, $\sigma$.

**Remark 2.** (Impact of graph sampling) A graph sampling method that can sample more class-relevant nodes in the distance-$z_m$ neighborhood can improve the learning by reducing $\epsilon_{\mathcal{S}}$.

### 4.4 WHAT DOES SELF-ATTENTION IMPROVE? A COMPARISON WITH GCN

We show that the attention weights become concentrated on class-relevant nodes in Lemma 1. It increases the distance between output vectors from different classes, which in turn improves the test accuracy. In contrast, Theorem 4.2 shows that without the self-attention layer, GCN requires more iterations and training samples.

**Lemma 1.** *(Sparse attention map) The attention weights for each node become increasingly concentrated on those correlated with class-relevant nodes during the training, i.e.,*

$$\sum_{i \in \mathcal{S}_*^{n,t}} softmax_n(\boldsymbol{x}_i^\top \boldsymbol{W}_K^{(t)}{}^\top \boldsymbol{W}_Q^{(t)} \boldsymbol{x}_n + \boldsymbol{u}_{(i,n)}^\top \boldsymbol{b}^{(t)}) \to \begin{cases} 1 - \eta^C, & n: discriminative, \\ 1 - \epsilon_{\mathcal{S}} - \eta^C, & n: non\text{-}discriminative. \end{cases} \quad (10)$$

*at a sublinear rate of $O(1/t)$ as t increases for a large $C > 0$ and all $n \in \mathcal{V}$.*

Lemma 1 indicates that the outputs of the self-attention layer for all nodes, which are weighted summations of value vectors, evolve in the direction of the class-relevant value features along the training. Then it promotes learning class-relevant features while ignoring other features. Lemma 1 is a generalization of Proposition 2 in (Li et al., 2023a), which considers a shallow ViT with one self-attention layer without positional embedding or graph structure. Here, we extend the analysis to node classification on graphs with PE.

Theorem 4.2 indicates that without the self-attention layer, the resulting GCN requires more training iterations and samples to achieve zero generalization, even if the core distance $z_m$ is known, and the learning is performed on the core neighborhood only. Specifically,

**Theorem 4.2.** *(Generalization of GCN) When fixing $\boldsymbol{W}_K = \boldsymbol{W}_Q = 0$ and $\boldsymbol{b} = 0$ in (1), and all $\mathcal{S}^{n,t}$ ($n \in \mathcal{L}$) and $\mathcal{T}^n$ ($n \in \mathcal{V} - \mathcal{L}$) are subsets of $\mathcal{N}_{z_m}^n$, the resulting GCN (Kipf & Welling, 2017; Nt & Maehara, 2019) learning on the core neighborhood $\mathcal{N}_{z_m}^n$ can achieve a zero generalization with the same condition in Theorem 4.1, but the number of iterations and the sample complexity should satisfy*

$$T = \Theta(\eta^{-1/2}(1 - 2p)^{-1/2}\gamma_d^{-2}(1 - \delta - \tau)^{-1/2}), \quad (11)$$

$$|\mathcal{L}| \geq \max\{\Omega((\gamma_d^2 - (\sigma + \tau))^{-2}(1 + \delta_{z_m}^2) \log N), BT\}, \quad (12)$$

When $m \gg m_a$, $m_b$, i.e., the number of parameters is almost the same for GCN and GT, Theorem 4.2 shows that GCN requires $\Theta(\gamma_d^{-2})$ times more training samples and iterations[8] to achieve zero generalization error than those using GT in (9) and (8), respectively. This explains the advantage of using self-attention layers as in insight (P2).

### 4.5 HOW DOES POSITIONAL ENCODING GUIDE GRAPH LEARNING PROCESS?

In this section, we study how PE affects learning performance. Our insight is that the learnable parameter for the PE promotes the core neighborhood for classification and, thus, improves the sample complexity and required number of iterations for generalization. To see this, first, Lemma 2 shows that the largest entry in $\boldsymbol{b}^{(T)}$ indeed corresponds to the core distance $z_m$. Therefore, PE "attracts the attention" of GT to the $z_m$-distance neighborhood. Then, Theorem 4.3 indicates that learning with the positional embedding has the same generalization performance as an artificial learning process when the core neighborhood $\mathcal{N}_{z_m}^n$ is known, and the learning is performed on $\mathcal{N}_{z_m}^n$ only.

---

[8]All the sample complexity and iteration bounds in this paper are obtained based on sufficient conditions for zero generalization. Rigorously speaking, necessary conditions are also required to compare the generalization of different network architectures. However, necessary conditions are rarely considered in the literature due to technical challenges. Here we still believe it is a fair comparison of sufficient conditions because we employ the same tools to analyze different neural network architectures.

**Lemma 2.** *Starting from $\boldsymbol{b}^{(0)} = 0$, if $T$ satisfies (8), the returned model trained by Algorithm 1 satisfies*

$$b_{z_m}^{(T)} - b_z^{(T)} \geq \Omega(\gamma_d(\bar{\Delta}(z_m) - \bar{\Delta}(z))), \tag{13}$$

Lemma 2 shows that $b_{z_m}$ is the largest one among all $1 \leq z \leq Z - 1$ because $\bar{\Delta}(z_m)$ is the largest by (6). Because the softmax function employs $e^{b_z}$ when computing the attention map, nodes at the $z_m$-distance neighborhood dominate the attention weights.

**Theorem 4.3.** *(The equivalent effect of the positional embedding)[9] when $\boldsymbol{b} = 0$ in (1), and all $\mathcal{S}^{n,t}$ ($n \in \mathcal{L}$) and $\mathcal{T}^n$ ($n \in \mathcal{V} - \mathcal{L}$) are subsets of $\mathcal{N}_{z_m}^n$, a zero generalization can be achieved when the sample complexity and the number of iterations satisfy (9) and (8) in Theorem 4.1.*

The learning process described in Theorem 4.3 is artificial because $z_m$ is generally unknown. Theorem 4.3 shows that learning with position embedding has an equivalent generalization performance to learning from the core neighborhood $\mathcal{N}_{z_m}^n$ only.

### 4.6 PROOF SKETCH

The main proof idea of Theorem 4.1 is to unveil a joint learning mechanism of GTs for our graph data model: (i) identifying discriminative features and the core neighborhood using PE and (ii) determining the labels of non-discriminative nodes through a majority vote in the core neighborhood by self-attention. Several lemmas are introduced to support the proof.

Specifically, by supportive Lemmas 8 and 9, we first characterize two groups of neurons that respectively activate the self-attention layer output of $\boldsymbol{\mu}_1$ and $\boldsymbol{\mu}_2$ nodes from initialization. Then, Lemma 4 shows that the neurons of $\boldsymbol{W}_O$ in these two groups grow along the two directions of the discriminative pattern embeddings. Lemma 7 indicates that the updates of $\boldsymbol{W}_V$ consist of neuron weights from these two groups. Meanwhile, Lemma 5 states that $\boldsymbol{W}_Q$ and $\boldsymbol{W}_K$ evolve to promote the magnitude of query and key embeddings of discriminative nodes. Lemma 6 depicts the training trajectory of the learning parameter of PE that emphasizes the core neighborhood. Different from the proof in (Li et al., 2023a; Tian et al., 2023; Li et al., 2023b; Tarzanagh et al., 2023) that does not consider PE and graph structure, we make the proof of each lemma tractable by studying gradient growth per distance-$z$ neighborhood for each $z$ rather than directly characterizing the gradient growth over the whole graph. Such a technique enables a dynamic tracking of per-parameter gradient updates. As a novel aspect, we prove Lemma 6 by showing that its most significant gradient component is proportional to the average winning margin in the core neighborhood.

**Proof of Theorem 4.1** We can build the generalization guarantee in Theorem 4.1 from the above. First, Lemma 5 and 6 collaborate to illustrate that attention weights correlated with class-relevant nodes become close to 1 when $\eta t = \Theta(1)$. Second, we compute the network output by Lemmas 4 and 7. By enforcing the output to be either $\geq 1$ or $\leq -1$ to achieve zero Hinge loss, we derive the sample complexity bound and the required number of iterations by concentration inequalities.

**The proof of Theorem 4.2 and 4.3** follow a similar idea as Theorem 4.1. When the self-attention layer weights are fixed at 0 in Theorem 4.2, since that $\gamma_d = \Theta(1)$ and a given core neighborhood still ensure non-trivial attention weights correlated with class-relevant nodes along the training, the updates of $\boldsymbol{W}_O$ and $\boldsymbol{W}_V$ are order-wise the same as Lemmas 4 and 7. Then, we can apply Lemmas 4 and 7 to derive the required number of samples and iterations for zero generalization. Likewise, given a known core neighborhood in Theorem 4.3, the remaining parameters follow the same order-wise update as Lemmas 4, 5 and 7. Hence, Theorems 4.2 and 4.3 can be proved.

## 5 NUMERICAL EXPRIMENTS

### 5.1 EXPERIMENTS ON SYNTHETIC DATA

**Graph data generation**: The graph contains 1000 nodes in total. $M = 10$, $\boldsymbol{\mu}_1$ to $\boldsymbol{\mu}_M$ are selected as orthonormal vectors in $\mathbb{R}^d$, where $d$ is 20. Node features that correspond to pattern $\boldsymbol{\mu}_i$ are sampled from Gaussian distributions $\mathcal{N}(\boldsymbol{\mu}_i, c_0^2 \cdot I)$, where $c_0 = 0.01$, and $\boldsymbol{I} \in \mathbb{R}^d$ is the identity matrix. $\gamma_d/2$ fraction of nodes are selected as noisy versions of class-discriminative $\boldsymbol{\mu}_1$ and $\boldsymbol{\mu}_2$, respectively. The remaining nodes are evenly distributed among other non-discriminative $M - 2$ patterns. $\gamma_d = 0.4$

---

[9]We discuss the application of Theorem 4.3 to analyze the generalization of one-layer GAT in Appendix E.3.

unless otherwise specified. Our graph construction method is motivated by and extends from that in (Zhang et al., 2023b). Every non-discriminative node is labeled with +1 or -1 with equal probability. If labeled +1, that non-discriminative node is randomly connected with $120 \cdot (1 - \epsilon_{\mathcal{S}})$ nodes of $\boldsymbol{\mu}_1$ and $120 \cdot \epsilon_{\mathcal{S}}$ of $\boldsymbol{\mu}_2$ for some $\epsilon_{\mathcal{S}}$ in $[0, 1/2)$. If labeled -1, it is randomly connected with $120 \cdot (1 - \epsilon_{\mathcal{S}})$ nodes of $\boldsymbol{\mu}_2$ and $120 \cdot \epsilon_{\mathcal{S}}$ of $\boldsymbol{\mu}_1$. We also add edges among $\boldsymbol{\mu}_1$ nodes themselves, and edges among $\boldsymbol{\mu}_2$ nodes themselves to make each node degree at least 120. There is no edge between $\boldsymbol{\mu}_1$ nodes and $\boldsymbol{\mu}_2$ nodes. The ground-truth label for $\boldsymbol{\mu}_1$ or $\boldsymbol{\mu}_2$ nodes is +1 or -1, respectively. $p = 0$ if not otherwise specified.

**Learner network and algorithm**: The learner network is a one-layer GT defined in equation 1. Set dimensions of embeddings to be $m_a = m_b = 20$. The number of neurons $m$ of $\boldsymbol{W}_O$ is 400. $\delta = 0.2$, $\sigma = 0.1$, and $\xi = 0.01$. $\boldsymbol{W}_Q^{(0)} = \boldsymbol{W}_Q^{(0)} = \delta^2 \boldsymbol{I}/c_0^2$, $\boldsymbol{W}_V^{(0)} = \sigma^2 \boldsymbol{U}/c_0^2$, where each entry of $\boldsymbol{W}_O^{(0)}$ follows $\mathcal{N}(0, \xi^2)$. $\boldsymbol{U}$ is an $m_a \times m_a$ orthonormal matrix. The step size $\eta = 0.01$. $\mathcal{S}^{n,t}$ contains node $n$ and 60 uniformly sampled nodes from distance-1 and distance-2 neighborhood for each node $n$ at iteration $t$.

**Sample complexity and convergence rate**: We first study the impact of the fraction $\gamma_d$ of discriminative nodes on the sample complexity. Let $\epsilon_{\mathcal{S}} = 0.05$. We implement 20 independent experiments with the same $\gamma_d$ and $|\mathcal{L}|$ while randomly generating graph structure, node features, and sampled labels. An experiment is successful if the Hinge testing loss is smaller than $10^{-3}$. A black block means all the trials fail, while a white block means they all succeed. Figure 3 (a) shows that the sample complexity is indeed almost linear in $\gamma_d^{-2}$, as indicated in 9. We next set $\gamma_d = 0.4$ and vary $\epsilon_{\mathcal{S}}$. Figure 3 (b) shows that the sample complexity is linear in $(1 - \epsilon_{\mathcal{S}})^{-2}$, which is consistent with our result in (9). We then change $p$ and evaluate the prediction error when the number of training iterations change, when $\gamma_d = 0.4$, $\epsilon_{\mathcal{S}} = 0$, and $|\mathcal{L}| = 400$. Figure 4 shows that a larger $p$ requires more iterations to achieve the same generalization, and the increase is significant when $p$ increases to 0.5, which is consistent with (8).

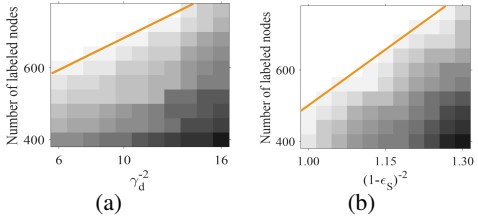

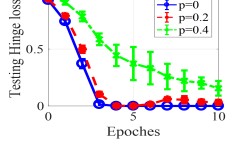

(a)    (b)

Figure 3: The impact of $\gamma_d$ and $\epsilon_{\mathcal{S}}$ on the sample complexity of GT.

Figure 4: The test Hinge loss against the number of epochs for different $p$.

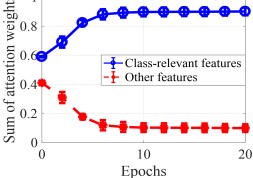

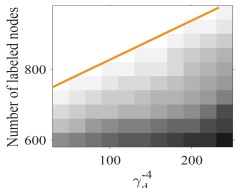

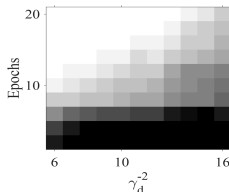

Figure 5: Concentration of attention weights

Figure 6: Sample complexity against $\gamma_d$

Figure 7: The required # of iterations against $\gamma_d$

**Attention map and comparison with GCN**: We then verify the sparsity of the attention map during the training. Let $|\mathcal{L}| = 400$, $\gamma_d = 0.2$. In Figure 5, the blue circled line shows the summation of attention weights on class-relevant nodes averaged over all labeled nodes increases to be close to 1 during training, which justifies (10), since when $\epsilon_{\mathcal{S}} = 0$, the left side of (10) converges to $1 - \eta^C$ for $C > 0$ for all nodes. Meanwhile, the summation of attention weights on other nodes decreases to be close to 0, as shown in the red dotted line. We also compare the performance on GT in (1) and a one-layer GCN with a similar architecture, and $\boldsymbol{W}_K$ and $\boldsymbol{W}_Q$ being 0, $\epsilon_{\mathcal{S}} = 0.2$. Figures 6 and 7 show the sample complexity and the required number of iterations of GCN are almost linear in $\gamma_d^{-4}$ and $\gamma_d^{-2}$, consistent with theoretical results in (12) and (11), respectively. In contrast, the theoretical sample complexity and the number of iterations of GT are respectively linear in $\gamma_d^{-2}$ (also see Figure 3) and independent of $\gamma_d$, which are order-wise smaller than GCN.

## 5.2 EXPERIMENTS ON REAL-WORLD DATASET

**Dataset and neural network model**: We evaluate node classification tasks on three benchmarks, a seven-classification citation graph PubMed (Kipf & Welling, 2017), a five-classification Actor co-occurrence graph (Chien et al., 2021), and a four-classification computer vision graph PascalVOC-SP-1G (Dwivedi et al., 2022), which are a homophilous, heterophilous, and a long-range graph, respectively. Please refer to Appendix A for detailed information on these datasets and results on large-scale dataset Ogbn-Arxiv (Hu et al., 2020a). The network contains four layers of four-head Transformer blocks. We implement the SPD-based PE as defined in (2) with $Z = 20$ and uniformly sample 20 nodes across the whole graph for feature aggregation of each node during every iteration.

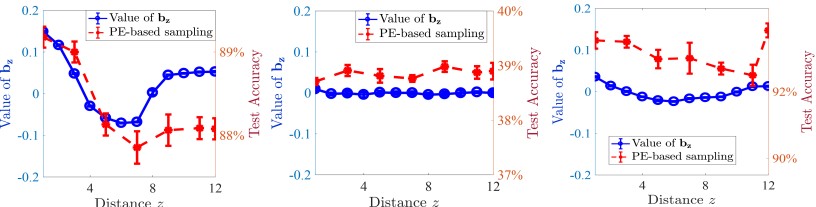

Figure 8: The values of entries of $b$ and the test accuracy of PE-based sampling. Left to right: PubMed, Actor, PascalVOC-SP-1G.

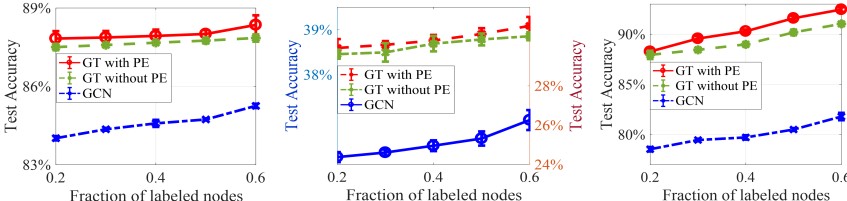

Figure 9: Test accuracy of GT with/without PE and GCN when the number of labeled nodes varies. Left to right: PubMed, Actor, PascalVOC-SP-1G.

**Success of PE**: The blue circled lines in Figure 8 show the average values of each dimension of the last-layer learned PE vector $b^{(T)}$ in these three datasets. We additionally train multiple models with the same setup, except that only distance-$z$ nodes are used for training and label prediction, i.e., $\mathcal{S}^{n,t}$ (for all labeled nodes $n$ and iteration $t$) and $\mathcal{T}^n$ (for all unlabeled nodes $n$) belong to $\mathcal{N}_z^n$. $|\mathcal{S}^{n,t}|$ is still 20. The red dashed curves show the test accuracy of these models. One can see that the test accuracy of these models has a similar trend as that of $b_z$ values. This justifies the success of PE and the existence of a core neighborhood defined in Definition 1. Moreover, SPD-based PE also correctly reflects the homophily, heterophily, and long-range dependency of these three datasets.

**Comparison of GTs with/without PE and GCN**. We use a four-layer GCN defined in (Kipf & Welling, 2017). The model size of GCN is slightly larger than GT by $\le 10\%$. Figure 9 shows that GT with PE has a better performance than that without PE and is better than GCN. This verifies Theorem 4.2 and discussions in Section 4.5.

## 6 CONCLUSION, LIMITATION, AND FUTURE WORK

This paper presents a novel theoretical analysis of Graph Transformers by explicitly characterizing the sample complexity and required number of SGD iterations needed to achieve zero generalization for node classification tasks. The analysis is based on a new graph data model that includes class-discriminative features that determine classes and class-irrelevant features, as well as a core neighborhood that determines the labels based on a majority vote of class-discriminative features. This paper shows that the sample complexity and training iterations are reduced when the fraction of class-discriminative nodes increases and/or the sampled nodes have a clear-cutting vote in the core neighborhood. This paper also proves that attention weights are concentrated on those of class-relevant nodes, and the positional embedding promotes the core neighborhood. All the theoretical results are centered on simplified shallow Transformer architectures, while experimental results on real-world datasets and deep neural network architectures support our theoretical findings. One future direction is theoretically analyzing more complex network architectures with milder assumptions. The authors find no ethical or immediate negative societal consequences of this work.

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

# APPENDIX

**The appendix contains five sections. We add some extra experiments in Section A. In Section B, we introduce some definitions and assumptions in accordance with the main paper for ease of proof. Section C first lists some key lemmas and then provides the proof of Theorem 4.1, Theorem 4.2, Theorem 4.3, Lemma 1, and Lemma 2. Section D shows the proof of lemmas of this paper. We finally add the extension of our analysis and other discussions in Section E.**

## A ADDITIONAL EXPERIMENTS

### A.1 VERIFYING ASSUMPTIONS MADE ON THE GRAPH DATA MODEL

For the assumption on the graph data model, we conduct several experiments to verify this assumption on the real-world dataset Cora, PubMed, Actor, and PascalVOC-SP-1G.

**Existence of discriminative nodes.** We first compute the eigenvalue of the covariance matrix of the feature matrix of data of all classes in Figures 10, 11, 12, and 13. One can observe that the feature matrix is almost low-rank, which indicates that node features from the same class can be represented by a few eigenvectors. Therefore, for each class, we select the top three eigenvectors and compute the 3-dimensional representations of each node feature with these three eigenvectors. Then, we select all nodes with features that are less than $\pi/4$ angle away from the mean of 3-dimensional representations as discriminative nodes. Non-discriminative nodes are the remaining nodes of each class. Tables 2, 3, 4, and 6 show the fraction of discriminative nodes in each class. One can see a large fraction of the node features in each class is close to its top three eigenvectors.

**The core distance (Assumption 1).** We further probe the core distance of each dataset by computing the fraction of nodes of which the label is aligned with the majority vote of the discriminative nodes in the distance-$z$ neighborhood. To extend the definition from binary classification in our formulation to multi-classification tasks, we use the average number of confusion nodes per class in the distance-$z$ neighborhood as $|\mathcal{D}_\#^n \cap \mathcal{N}_z^n|$, the number of confusion nodes in the distance-$z$ neighborhood of node $n$. Figure 14 shows the value of a normalized $\bar{\Delta}(z)$ for $z = 1, 2, \cdots, 12$, where $\bar{\Delta}(z)$ is divided by $|\mathcal{N}_z^n|$ to control the gap of different numbers of nodes in different neighborhoods. The empirical result indicates that (1) homophilous graphs Cora and PubMed have a decreasing value of the normalized $bar\Delta(z)$ as $z$ increases. The gap between the largest and the smallest normalized $\bar{\Delta}(z)$ is large. This implies the core distance is 1 for Cora and PubMed and is aligned with the PE-based sampling performance of PubMed in Figure 8. (2) the heterophilous graph Actor has the largest normalized $\bar{\Delta}(z)$ at $z = 1$, but the difference from other $z$ is very small. This is consistent with the result in Figure 8 where the PE-based sampling has a close performance of less than $0.5\%$ across $z$. (3) the long-range graph PascalVOC-SP-1G has the normalized $\bar{\Delta}(z)$ when $z = 1$, but the value when $z = 12$ is also remarkable. This corresponds to Figure 8 where the testing performance of PascalVOC-SP-1G is the highest when $z = 1$ or $z = 12$.

We then verify the balanced dataset assumption and show a difference of no more than $O(\sqrt{N})$ could be achieved in practical datasets. Table 9 shows that for Cora and Actor, this condition holds since the largest gap between the average number of nodes and the number of any class of nodes is smaller than $O(\sqrt{N}) = 10\sqrt{N}$.

| class 1 | class 2 | class 3 | class 4 | class 5 | class 6 | class 7 |
|---------|---------|---------|---------|---------|---------|---------|
| 82.05% | 88.02% | 82.54% | 78.12% | 78.17% | 83.56% | 76.11% |

Table 2: The fraction of discriminative nodes in each class of Cora

| class 1 | class 2 | class 3 |
|---------|---------|---------|
| 82.18% | 93.34% | 80.48% |

Table 3: The fraction of discriminative nodes in each class of PubMed

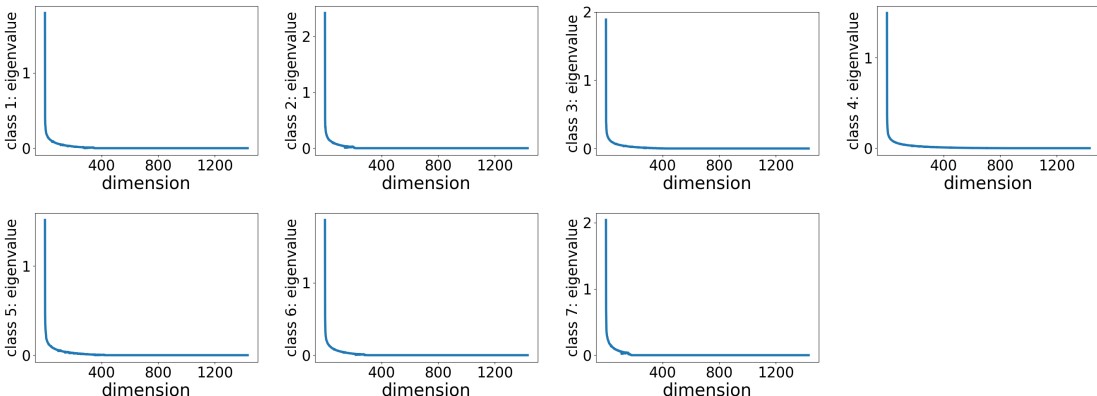

Figure 10: Eigenvalues of the covariance matrix of the feature matrix of all classes of Cora

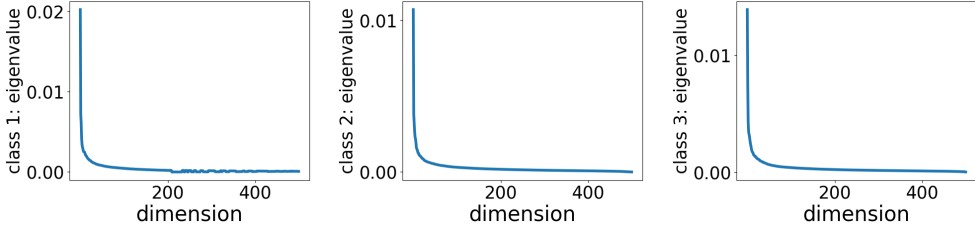

Figure 11: Eigenvalues of the covariance matrix of the feature matrix of all classes of PubMed

## A.2 EXPERIMENTS ON SYNTHETIC DATASET

This section compares the required number of iterations for Graph Transformer and GCN by their orders in $\gamma_d$. The experiment setup follows Section 5.1. We set the number of known labels to be 800. For Graph Transformer, $\epsilon_{\mathcal{S}} = 0.05$. For GCN, $\epsilon_{\mathcal{S}} = 0.2$. Figure 15 (a) shows that the required number of iterations is independent of $\gamma_d$. In contrast, Figure 15 (b), which is exactly Figure 7 indicates the number of iterations is linear in $\gamma_d^{-2}$.

## A.3 EXPERIMENTS ON REAL-WORLD DATASETS

We first add an introduction to the dataset PascalVOC-SP-1G we evaluate. This belongs to the Long Range Graph Benchmark, PascalVOC-SP (Dwivedi et al., 2022), which is a computer vision dataset for node classification containing $11,355$ graphs, $5,443,545$ nodes, and $30,777,444$ edges in total. Since this dataset is large, we pick the 2nd graph from the whole dataset and name this graph PascalVOC-SP-1G, which contains $479$ nodes and $2,718$ edges for node classification. The dimension of the node feature is $14$. The number of classes is $3$. Note that the size of the graph is not small compared with WebKB datasets (Pei et al., 2020), including Cornell, Texas, and Wisconsin, which contain $183$, $183$, and $251$ nodes in each dataset, respectively.

Meanwhile, to verify the scalability of our conclusion, we conduct the experiments on the large-scale graph dataset Ogbn-Arxiv (Hu et al., 2020a), which is a citation network with for node classification. The detailed statistics of these four datasets can be found in Table A.3.

We show the results of the Ogbn-Arxiv in Figure 17 and 18, where the dimension of $\boldsymbol{b}^{(T)}$ is set to be 5. We still plot $b_z^{(T)}$ with blue-circled lines for these datasets. Red dashed curves denote the test accuracy of the models learned with nodes all sampled from the distance-$z$ neighborhood for

| class 1 | class 2 | class 3 | class 4 | class 5 |
|---------|---------|---------|---------|---------|
| 42.09%  | 53.33%  | 57.85%  | 60.93%  | 64.79%  |

Table 4: The fraction of discriminative nodes in each class of Actor

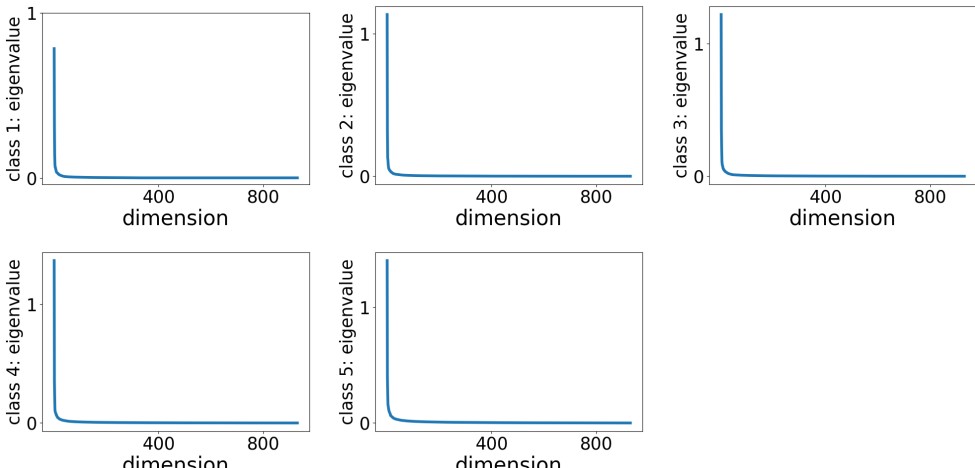

Figure 12: Eigenvalues of the covariance matrix of the feature matrix of all classes of Actor

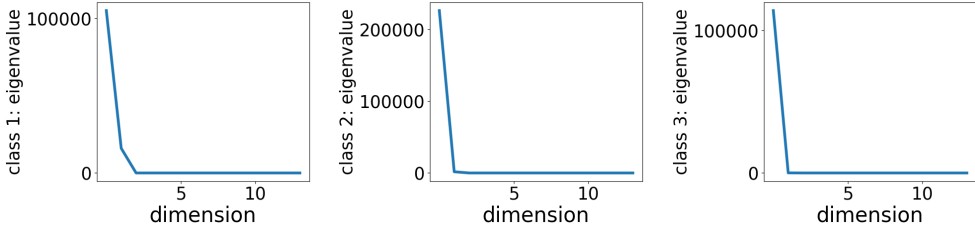

Figure 13: Eigenvalues of the covariance matrix of the feature matrix of all classes of PascalVOC-SP-1G

$z \in \{1, 2, \cdots, 5\}$. The result of Ogbn-Arxiv shows a large $b_z^{(T)}$ when $z$ is around 1. One can also observe that the testing accuracy using only distance-$z$ nodes has a similar trend as $b_z^{(T)}$ with the largest accuracy around $z = 1$. This is consistent with our conclusions on PubMed from Figure 17 in Section 5.2 since Ogbn-Arxiv and PubMed are both citation networks that are homophilous.

Figure 18 showcases that for Ogbn-Arxiv, GT with PE has a better performance than that without PE and GCN. The conclusion is consistent with Figure 9

## B PRELIMINARIES

We first formally state the Algorithm 1. The notations used in the Appendix is summarized in Table 8.

The loss function of a single data is defined in the following.

$$\text{Loss}(\boldsymbol{x}_l, y_l) = \max\{1 - y_l \cdot F(\boldsymbol{x}_l), 0\}. \tag{14}$$

The formal algorithm is as follows. At each iteration $t$, the gradient is computed using a mini-batch $\mathcal{B}_t$ with $|\mathcal{B}_t| = B$ and step size $\eta$. We first pre-train $\boldsymbol{W}_O$ for $T_0$ steps and then implement a full training with all parameters in $\psi$ except $\boldsymbol{a}$ for $T(\geq T_0)$ steps. At iteration $t$, we uniformly sample a subset $\mathcal{S}^{n,t}$ of nodes from the whole graph for aggregation of each node $n$. We set that $\boldsymbol{W}_V^{(0)}, \boldsymbol{W}_Q^{(0)}$, and $\boldsymbol{W}_K^{(0)}$ come from an initial model. Every entry of $\boldsymbol{W}_O^{(0)}$ is generated from $\mathcal{N}(0, \xi^2)$. Every entry

| class 1 | class 2 | class 3 |
|---------|---------|---------|
| 98.62%  | 100%    | 100%    |

Table 5: The fraction of discriminative nodes in each class of PascalVOC-SP-1G

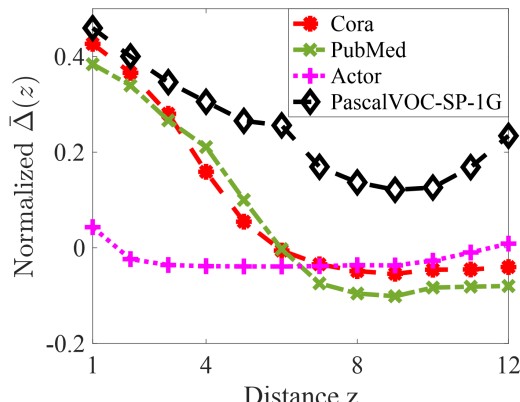

Figure 14: Normalized $\bar{\Delta}(z)$ for Cora, PubMed, Actor, and PascalVOC-SP-1G.

| Cora | PubMed 2 | Actor | PascalVOC-SP-1G |
|---|---|---|---|
| 95.68% | 85.73% | 86.31% | 98.54% |

Table 6: The fraction of nodes satisfying $\Delta_n(z_m) > 0$

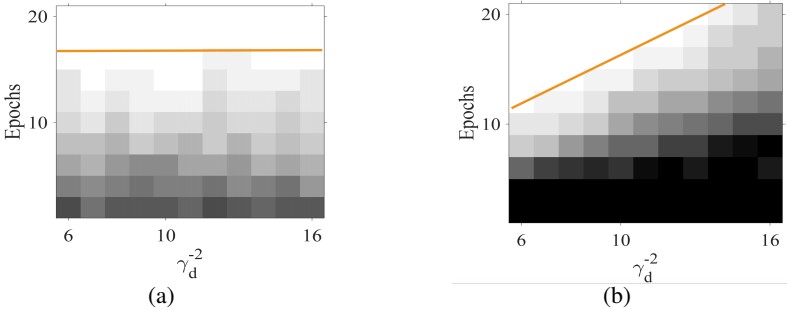

Figure 15: The required number of iterations against $\gamma_d^{-2}$ (a) Graph Transformer (b) GCN.

Table 7: The statistics of datasets.

| Dataset | #Nodes | #Edges | #Classes | #Features | Type |
|---|---|---|---|---|---|
| PubMed | $19,717$ | $44,324$ | 3 | 500 | Citation network |
| Actor | $7,600$ | $26,659$ | 5 | 932 | Actors in movies |
| PascalVOC-SP-1G | 479 | $2,718$ | 3 | 14 | Computer vision |
| Ogbn-Arxiv | $169,343$ | $1,166,243$ | 40 | 128 | Citation network |

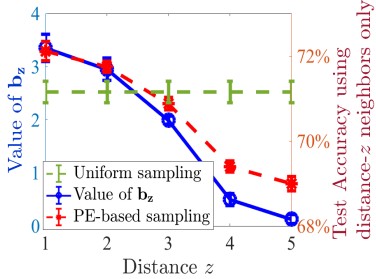 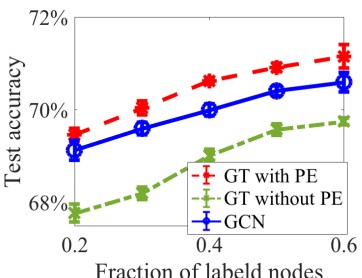

Figure 17: The values of entries of $\boldsymbol{b}$ and the test accuracy of PE-based sampling for Ogbn-Arxiv.
()

Figure 18: Test accuracy of GT with/without PE and GCN when the number of label nodes varies for Ogbn-Arxiv.
()

Figure 18: The required number of iterations against $\gamma_d^{-2}$ (a) Graph Transformer (b) GCN.

Table 8: Summary of notations

| | |
|---|---|
| $F(\boldsymbol{x}_l), \text{Loss}(\boldsymbol{x}_l, y_l)$ | The network output for the node $\boldsymbol{x}_l$ and the loss function of a single node. |
| $\boldsymbol{p}_j(t), \boldsymbol{q}_j(t), \boldsymbol{r}_j(t)$ | The features in value, key, and query vectors at the iteration $t$ for pattern $j$, respectively. We have $\boldsymbol{p}_j(0) = \boldsymbol{p}_j$, $\boldsymbol{q}_j(0) = \boldsymbol{q}_j$, and $\boldsymbol{r}_j(0) = \boldsymbol{r}_j$. |
| $\boldsymbol{z}_j(t), \boldsymbol{n}_j(t), \boldsymbol{o}_j(t)$ | The error terms in the value, key, and query vectors of the $j$-th node compared to their features at iteration $t$. |
| $\mathcal{W}_l(0), \mathcal{U}_l(0)$ | The set of lucky neurons for node $l$. |
| $\phi_n(t), \nu_n(t), p_n(t), \lambda$ | Approximate value of some attention weights at iteration $t$. $\lambda$ is the threshold between inner products of tokens from the same pattern and different patterns. |
| $\mathcal{S}_j^{n,t}$ | $\mathcal{S}_j^{n,t}$ is the set of sampled nodes of pattern $j$ at iteration $t$ to compute the aggregation of node $n$. |
| $\delta_z$ | The maximum number of nodes in distance-$z$ neighborhood for all nodes, which is no larger than $\sqrt{N}$. |

of $\boldsymbol{a}^{(0)}$ is sampled from $\{+1/\sqrt{m}, -1/\sqrt{m}\}$ with equal probability. $\boldsymbol{b}^{(0)} = \boldsymbol{0}$. $\boldsymbol{a}$ does not update during the training.

**Assumption 2.** *Li et al. (2023a) Define* $\boldsymbol{P} = (\boldsymbol{p}_1, \boldsymbol{p}_2, \cdots, \boldsymbol{p}_M) \in \mathbb{R}^{m_a \times M}$, $\boldsymbol{Q} = (\boldsymbol{q}_1, \boldsymbol{q}_2, \cdots, \boldsymbol{q}_M) \in \mathbb{R}^{m_b \times M}$ *and* $\boldsymbol{R} = (\boldsymbol{r}_1, \boldsymbol{r}_2, \cdots, \boldsymbol{r}_M) \in \mathbb{R}^{m_b \times M}$ *as three feature matrices, where* $\mathcal{P} = \{\boldsymbol{p}_1, \boldsymbol{p}_2, \cdots, \boldsymbol{p}_M\}$, $\mathcal{Q} = \{\boldsymbol{q}_1, \boldsymbol{q}_2, \cdots, \boldsymbol{q}_M\}$ *and* $\mathcal{R} = \{\boldsymbol{r}_1, \boldsymbol{r}_2, \cdots, \boldsymbol{r}_M\}$ *are three sets of orthonormal bases. Define the noise terms* $\boldsymbol{z}_j(t)$, $\boldsymbol{n}_j(t)$ *and* $\boldsymbol{o}_j(t)$ *with* $\|\boldsymbol{z}_j(0)\| \leq \sigma + \tau$ *and* $\|\boldsymbol{n}_j(0)\|, \|\boldsymbol{o}_j(0)\| \leq \delta + \tau$ *for* $j \in [L]$. $\boldsymbol{q}_1 = \boldsymbol{r}_1$, $\boldsymbol{q}_2 = \boldsymbol{r}_2$. *Suppose* $\|\boldsymbol{W}_V^{(0)}\|, \|\boldsymbol{W}_K^{(0)}\|, \|\boldsymbol{W}_Q^{(0)}\| \leq 1$, $\sigma, \tau < O(1/M)$ *and* $\delta < 1/2$. *Then, for* $\boldsymbol{x}_l \in \mathcal{S}_j^n$

1. $\boldsymbol{W}_V^{(0)}\boldsymbol{x}_l = \boldsymbol{p}_j + \boldsymbol{z}_j(0)$.

2. $\boldsymbol{W}_K^{(0)}\boldsymbol{x}_l = \boldsymbol{q}_j + \boldsymbol{n}_j(0)$.

3. $\boldsymbol{W}_Q^{(0)}\boldsymbol{x}_l = \boldsymbol{r}_j + \boldsymbol{o}_j(0)$.

| | average # of each class | $10\sqrt{N}$ | largest gap to the average |
|---|---|---|---|
| Cora | 386.86 | 520.38 | 431.14 |
| Actor | 1520 | 667 | 871.78 |

Table 9: The fraction of discriminative nodes in each class of Actor

---

**Algorithm 1** Training with Stochastic Gradient Descent (SGD)

---

1: **Input:** Training data $\{(\boldsymbol{X}, y_n)\}_{n \in \mathcal{L}}$, the step size $\eta$, the number of iterations $T$, batch size $B$.

2: **Initialization:** Each entry of $\boldsymbol{W}_O^{(0)}$ and $\boldsymbol{a}^{(0)}$ from $\mathcal{N}(0, \xi^2)$ and Uniform($\{+1/\sqrt{m}, -1/\sqrt{m}\}$), respectively. $\boldsymbol{W}_V^{(0)}, \boldsymbol{W}_K^{(0)}$ and $\boldsymbol{W}_Q^{(0)}$ are initialized from a fair model. $\boldsymbol{b}^{(0)} = 0$.

3: **Node sampling:** At each iteration $t$, sample $\mathcal{S}^{n,t}$ for each node $n$ to replace $\mathcal{T}^n$ in (1) when computing the $\ell(\cdot)$ function in (3).

4: **Training by SGD:** For $t = 0, 1, \cdots, T-1$ and $\boldsymbol{W}^{(t)} \in \{\boldsymbol{W}_O^{(t)}, \boldsymbol{W}_V^{(t)}, \boldsymbol{W}_K^{(t)}, \boldsymbol{W}_Q^{(t)}, \boldsymbol{b}^{(t)}\}$

$$\boldsymbol{W}^{(t+1)} = \boldsymbol{W}^{(t)} - \eta \cdot \frac{1}{B} \sum_{n \in \mathcal{B}_t} \nabla_{\boldsymbol{W}^{(t)}} \ell(\boldsymbol{x}_n, y_n; \boldsymbol{a}^{(0)}, \boldsymbol{W}_O^{(t)}, \boldsymbol{W}_V^{(t)}, \boldsymbol{W}_K^{(t)}, \boldsymbol{W}_Q^{(t)}, \boldsymbol{b}^{(t)}) \qquad (15)$$

5: **Output:** $\boldsymbol{W}_O^{(T)}, \boldsymbol{W}_V^{(T)}, \boldsymbol{W}_K^{(T)}, \boldsymbol{W}_Q^{(T)}, \boldsymbol{b}^{(T)}$.

---

Assumption 2 is a straightforward combination of Assumption 1 in (Li et al., 2023a) and the equation $\min_{j \in [M]} \|\boldsymbol{x}_n - \boldsymbol{\mu}_j\| \leq \tau, \forall n \in \mathcal{V}$ by applying the triangle inequality to bound the error terms for tokens. We then provide a condition which is equivalent to the equation $\min_{j \in [M]} \|\boldsymbol{x}_n - \boldsymbol{\mu}_j\| \leq \tau, \forall n \in \mathcal{V}$ since $\tau < O(1/M)$, i.e., if nodes $i$ and $j$ correspond to the same pattern $k \in [M]$, i.e., $i \in \mathcal{D}_k$ and $j \in \mathcal{D}_k$, we have $\boldsymbol{x}_i^\top \boldsymbol{x}_j \geq 1$. If nodes $i$ and $j$ correspond to the different feature $k, l \in [M], k \neq l$ i.e., $i \in \mathcal{D}_k$ and $j \in \mathcal{D}_l, k \neq l$, we have $\boldsymbol{x}_i^\top \boldsymbol{x}_j \leq \lambda < 1$. Here, we scale up all nodes a bit to make the threshold of linear separability 1 for the simplicity of presentation.

**Definition 3.** *Define*

$$\boldsymbol{V}_n(t) = \boldsymbol{W}_V^{(t)} \sum_{s \in \mathcal{T}^n} \boldsymbol{x}_n \, \text{softmax}_n(\boldsymbol{x}_s^\top \boldsymbol{W}_K^{(t)\top} \boldsymbol{W}_Q^{(t)} \boldsymbol{x}_n + \boldsymbol{u}_{(s,n)}^\top \boldsymbol{b}^{(t)}). \qquad (16)$$

*for the node $n$. Define $\mathcal{W}_n(0), \mathcal{U}_n(0)$ as the sets of lucky neurons such that*

$$\mathcal{W}_n(0) = \{i : \boldsymbol{W}_{O_{(i,\cdot)}}^{(0)} \boldsymbol{V}_n(0) > 0, l \in \mathcal{S}_1^{n,t}\}, \qquad (17)$$

$$\mathcal{U}_n(0) = \{i : \boldsymbol{W}_{O_{(i,\cdot)}}^{(0)} \boldsymbol{V}_n(0) > 0, l \in \mathcal{S}_2^{n,t}\}. \qquad (18)$$

**Definition 4.** *When $n \in \mathcal{D}_1 \cup \mathcal{D}_2$, we have*

1. $\phi_n(t) = (\sum_{z \in \mathcal{Z}} |\mathcal{N}_z^n \cap \mathcal{S}_*^{n,t}| e^{\|\boldsymbol{q}_1(t)\|^2 + (\sigma+\tau)\|\boldsymbol{q}_1(t)\| + b_z^{(t)}} + \sum_{z \in \mathcal{Z}} |(\mathcal{N}_z^n \cap \mathcal{S}^{n,t}) - \mathcal{S}_1^{n,t}| e^{b_z^{(t)}})^{-1}$.

2. $\nu_n(t) = (\sum_{z \in \mathcal{Z}} |\mathcal{N}_z^n \cap \mathcal{S}_*^{n,t}| e^{\|\boldsymbol{q}_1(t)\|^2 - (\sigma+\tau)\|\boldsymbol{q}_1(t)\| + b_z^{(t)}} + \sum_{z \in \mathcal{Z}} |(\mathcal{N}_z^n \cap \mathcal{S}^{n,t}) - \mathcal{S}_1^{n,t}| e^{b_z^{(t)}})^{-1}$.

3. $p_n(t) = \sum_{z \in \mathcal{Z}} |\mathcal{N}_z^n \cap \mathcal{S}_*^{n,t}| e^{\|\boldsymbol{q}_1(t)\|^2 - (\sigma+\tau)\|\boldsymbol{q}_1(T)\| + b_z^{(t)}} \nu_n(t)$.

*When $n \notin \mathcal{D}_1 \cup \mathcal{D}_2$, we have*

1. $\phi_n(t) = (\sum_{z \in \mathcal{Z}} (|\mathcal{N}_z^n \cap \mathcal{S}_*^{n,t}| + |\mathcal{N}_z^n \cap \mathcal{S}_\#^{n,t}|) e^{\|\boldsymbol{q}_1(t)\|^2 + (\sigma+\tau)\|\boldsymbol{q}_1(t)\| + b_z^{(t)}} + \sum_{z \in \mathcal{Z}} |(\mathcal{N}_z^n \cup \mathcal{S}^{n,t})/(\mathcal{S}_1^{n,t} \cup \mathcal{S}_2^{n,T})| e^{b_z^{(t)}})^{-1}$.

2. $\nu_n(t) = (\sum_{z \in \mathcal{Z}} (|\mathcal{N}_z^n \cap \mathcal{S}_*^{n,t}| + |\mathcal{N}_z^n \cap \mathcal{S}_\#^{n,t}|) e^{\|\boldsymbol{q}_1(t)\|^2 - (\sigma+\tau)\|\boldsymbol{q}_1(t)\| + b_z^{(t)}} + \sum_{z \in \mathcal{Z}} |(\mathcal{N}_z^n \cup \mathcal{S}^{n,t})/(\mathcal{S}_1^{n,t} \cup \mathcal{S}_2^{n,t})| e^{b_z^{(t)}})^{-1}$.

3. $p_n(t) = \sum_{z \in \mathcal{Z}} |\mathcal{N}_z^n \cap \mathcal{S}_*^{n,t}| e^{\|\boldsymbol{q}_1(t)\|^2 - (\sigma+\tau)\|\boldsymbol{q}_1(t)\| + b_z^{(t)}} \nu_n(t)$.

We then cite useful results of the concentration bounds on sub-gaussian variables.

**Definition 5.** *(Vershynin, 2010) We say $X$ is a sub-Gaussian random variable with sub-Gaussian norm $K > 0$, if $(\mathbb{E}|X|^p)^{\frac{1}{p}} \leq K\sqrt{p}$ for all $p \geq 1$. In addition, the sub-Gaussian norm of X, denoted $\|X\|_{\psi_2}$, is defined as $\|X\|_{\psi_2} = \sup_{p \geq 1} p^{-\frac{1}{2}} (\mathbb{E}|X|^p)^{\frac{1}{p}}$.*

**Lemma 3.** *(Vershynin (2010) Proposition 5.1, Hoeffding's inequality) Let $X_1, X_2, \cdots, X_N$ be independent centered sub-gaussian random variables, and let $K = \max_i \|\boldsymbol{X}_i\|_{\psi_2}$. Then for every $\boldsymbol{a} = (a_1, \cdots, a_N) \in \mathbb{R}^N$ and every $t \geq 0$, we have*

$$\mathbb{P}\Big\{\Big|\sum_{i=1}^{N} a_i X_i\Big| \geq t\Big\} \leq e \cdot \exp(-\frac{ct^2}{K^2\|\boldsymbol{a}\|^2}). \tag{19}$$

*where $c > 0$ is an absolute constant.*

## C  KEY LEMMAS AND PROOF OF THE MAIN THEOREMS

We first present our key lemmas, followed by the proof of the main theorems.

For $l \in \mathcal{S}_1^{n,t}$ for the data with $y_n = 1$, define

$$\boldsymbol{V}_l(t) = \sum_{s \in \mathcal{S}^{n,t}} \boldsymbol{W}_V^{(t)} \boldsymbol{x}_s \text{softmax}_n(\boldsymbol{x}_s^\top \boldsymbol{W}_K^{(t)\top} \boldsymbol{W}_Q^{(t)} \boldsymbol{x}_n + \boldsymbol{u}_{(s,n)}^\top \boldsymbol{b}^{(t)}). \tag{20}$$

We later can show that

$$\boldsymbol{V}_l(t) = \sum_{s \in \mathcal{S}_1^{n,t}} \text{softmax}_n(\boldsymbol{x}_s^\top \boldsymbol{W}_K^{(t)\top} \boldsymbol{W}_Q^{(t)} \boldsymbol{x}_n + \boldsymbol{u}_{(s,n)}^\top \boldsymbol{b}^{(t)})\boldsymbol{p}_1 + \boldsymbol{z}(t) + \sum_{j \neq 1} W_j(t)\boldsymbol{p}_j$$
$$- \eta \sum_{b=1}^{t} \Big(\sum_{i \in \mathcal{W}_l(b)} V_i(b)\boldsymbol{W}_{O_{(i,\cdot)}}^{(b)\top} + \sum_{i \notin \mathcal{W}_l(b)} V_i(b)\lambda \boldsymbol{W}_{O_{(i,\cdot)}}^{(b)\top}\Big). \tag{21}$$

We have the following Lemmas:

**Lemma 4.** *For the lucky neuron $i \in \mathcal{W}_l(0)$ and $b \in [T]$, we have that the major component of $\boldsymbol{W}_{O_{(i,\cdot)}}^{(t)}$ is in the direction of $\boldsymbol{p}_1$, i.e.,*

$$\boldsymbol{W}_{O_{(i,\cdot)}}^{(t)} \boldsymbol{p}_1 \gtrsim \frac{\xi}{aB} \sum_{n \in \mathcal{B}_b} \frac{\eta t^2(1-2p)}{4B} \sum_{n \in \mathcal{B}_b} \frac{m}{a} p_n(t) + \xi, \tag{22}$$

$$\boldsymbol{W}_{O_{(i,\cdot)}}^{(t)} \boldsymbol{p} \lesssim \frac{1}{\sqrt{B}} \boldsymbol{W}_{O_{(i,\cdot)}}^{(t)} \boldsymbol{p}_1, \quad \text{for } \boldsymbol{p} \in \{\boldsymbol{p}_2, \boldsymbol{p}_3, \cdots, \boldsymbol{p}_M\}, \tag{23}$$

$$\|\boldsymbol{W}_{O_{(i,\cdot)}}^{(t)}\|^2 \geq (\frac{\xi}{aB} \sum_{n \in \mathcal{B}_b} \frac{\eta t^2(1-2p)}{4B} \sum_{n \in \mathcal{B}_b} \frac{m}{a} p_n(t) + \xi)^2, \tag{24}$$

*and for the noise $\boldsymbol{z}_l(t)$,*

$$\|\boldsymbol{W}_{O_{(i,\cdot)}}^{(t)} \boldsymbol{z}_l(t)\| \leq (\sigma + \tau)\|\boldsymbol{W}_{O_{(i,\cdot)}}^{(t)}\|. \tag{25}$$

*For $i \in \mathcal{U}_l(0)$, we also have equations as in (22) to (25), including*

$$\boldsymbol{W}_{O_{(i,\cdot)}}^{(t)} \boldsymbol{p}_2 \gtrsim \frac{\xi}{aB} \sum_{n \in \mathcal{B}_b} \frac{\eta t^2(1-2p)}{4B} \sum_{n \in \mathcal{B}_b} \frac{m}{a} p_n(t) + \xi, \tag{26}$$

$$\boldsymbol{W}_{O_{(i,\cdot)}}^{(t)} \boldsymbol{p} \lesssim \frac{1}{\sqrt{B}} \boldsymbol{W}_{O_{(i,\cdot)}}^{(t)} \boldsymbol{p}_1, \quad \text{for } \boldsymbol{p} \in \{\boldsymbol{p}_1, \boldsymbol{p}_3, \boldsymbol{p}_4, \cdots, \boldsymbol{p}_M\}, \tag{27}$$

$$\|\boldsymbol{W}_{O_{(i,\cdot)}}^{(t)}\|^2 \geq (\frac{\xi}{aB} \sum_{n \in \mathcal{B}_b} \eta t^2 \frac{\eta t^2(1-2p)}{4B} \sum_{n \in \mathcal{B}_b} \frac{m}{a} p_n(t) + \xi)^2. \tag{28}$$

*For the noise $\boldsymbol{z}_l(t)$,*

$$\|\boldsymbol{W}_{O_{(i,\cdot)}}^{(t)} \boldsymbol{z}_l(t)\| \leq (\sigma + \tau)\|\boldsymbol{W}_{O_{(i,\cdot)}}^{(t)}\|. \tag{29}$$

*For unlucky neurons $i$ and $j \in \mathcal{W}_l(0)$, $k \in \mathcal{U}_l(0)$, $\boldsymbol{p} \in \mathcal{P}$, we have*

$$\boldsymbol{W}_{O_{(i,\cdot)}}^{(t)} \boldsymbol{p} \leq \frac{1}{\sqrt{B}} \min\{\boldsymbol{W}_{O_{(j,\cdot)}}^{(t)} \boldsymbol{p}_1, \boldsymbol{W}_{O_{(k,\cdot)}}^{(t)} \boldsymbol{p}_2\}, \tag{30}$$

$$\|\boldsymbol{W}_{O_{(i,\cdot)}}^{(t)} \boldsymbol{z}_l(t)\| \leq (\sigma + \tau)\|\boldsymbol{W}_{O_{(j,\cdot)}}^{(t)}\|, \tag{31}$$

$$\|\boldsymbol{W}_{O_{(i,\cdot)}}^{(t)}\|^2 \leq \frac{1}{B} \min\{\|\boldsymbol{W}_{O_{(j,\cdot)}}^{(t)}\|^2, \|\boldsymbol{W}_{O_{(k,\cdot)}}^{(t)}\|^2\}. \tag{32}$$

**Lemma 5.** *There exists $K(t), Q(t) > 0$, $t = 0, 1, \cdots, T-1$ such that for $r \in \mathcal{S}_*^{n,t}$, if $u_{(r,l)_{z_0}} = 1$, defining*

$$\boldsymbol{q}_i(t) = \sqrt{\prod_{l=0}^{t-1}(1 + K(l))}\boldsymbol{q}_i, \tag{33}$$

$$\boldsymbol{r}_i(t) = \sqrt{\prod_{l=0}^{t-1}(1 + Q(l))}\boldsymbol{r}_i, \tag{34}$$

*where $i = 1, 2$. Then, we have*

$$softmax_l(\boldsymbol{x}_r{}^\top \boldsymbol{W}_K^{(t+1)}\boldsymbol{W}_Q^{(t+1)}\boldsymbol{x}_l + \boldsymbol{u}_{(r,l)}^\top \boldsymbol{b}^{(t+1)})$$

$$\gtrsim \frac{e^{(1+K(t))\|\boldsymbol{q}_1(t)\|^2 - (\delta+\tau)\|\boldsymbol{q}_1(t)\| + b_{z_0}^{(t)}}}{\sum_{z \in \mathcal{Z}}|\mathcal{N}_z^n \cap \mathcal{S}_*^{n,T}|e^{(1+K(t))\|\boldsymbol{q}_1(T)\|^2 - (\sigma+\tau)\|\boldsymbol{q}_1(T)\| + b_z^{(T)}} + \sum_{z \in \mathcal{Z}}|(\mathcal{N}_z^n \cap \mathcal{S}^{n,T}) - \mathcal{S}_1^{n,T}|e^{b_z^{(T)}}}. \tag{35}$$

*Similarly, for $r \notin \mathcal{S}_*^{l,t}$, we have*

$$softmax_l(\boldsymbol{x}_r{}^\top \boldsymbol{W}_K^{(t+1)}{}^\top \boldsymbol{W}_Q^{(t+1)}\boldsymbol{x}_l + \boldsymbol{u}_{(r,l)}^\top \boldsymbol{b}^{(t)})$$

$$\lesssim \frac{e^{b_{z_0}^{(t)}}}{\sum_{z \in \mathcal{Z}}|\mathcal{N}_z^n \cap \mathcal{S}_*^{n,T}|e^{(1+K(t))\|\boldsymbol{q}_1(T)\|^2 - (\sigma+\tau)\|\boldsymbol{q}_1(T)\| + b_z^{(T)}} + \sum_{z \in \mathcal{Z}}|(\mathcal{N}_z^n \cap \mathcal{S}^{n,T}) - \mathcal{S}_1^{n,T}|e^{b_z^{(T)}}}. \tag{36}$$

**Lemma 6.** *During the training, we fix $b_0^{(t)} = b_0^{(0)} = \Omega(1)$. For $z \geq 1$,*

$$b_{z_m}^{(t)} - b_z^{(t)}$$

$$\gtrsim \eta \frac{1}{B}\sum_{b=1}^{t}\sum_{n \in \mathcal{B}_b}\eta\frac{(1-2p)^3}{B}\sum_{b=1}^{t}\sum_{n \in \mathcal{B}_b}\frac{p_n(b)m^2}{a^2}(\frac{\xi\eta t^2 m}{a^2})^2\|\boldsymbol{p}_1\|^2 \cdot \frac{\gamma_d}{2} \tag{37}$$

$$\cdot (\frac{|\mathcal{S}_*^{l,t} \cap \mathcal{N}_{z_m}^l| - |\mathcal{S}_\#^{l,t} \cap \mathcal{N}_{z_m}^l|}{K|\mathcal{S}^{l,t}|} - \frac{|\mathcal{S}_*^{l,t} \cap \mathcal{N}_z^l| - |\mathcal{S}_\#^{l,t} \cap \mathcal{N}_z^l|}{K|\mathcal{S}^{l,t}|}).$$

$$b_z^{(t)} \geq \eta \frac{1}{B}\sum_{b=1}^{t}\sum_{n \in \mathcal{B}_b}\eta\frac{(1-2p)^3}{B}\sum_{b=1}^{t}\sum_{n \in \mathcal{B}_b}\frac{p_n(b)m^2}{a^2}(\frac{\xi\eta t^2 m}{a^2})^2\|\boldsymbol{p}_1\|^2 \cdot \frac{\gamma_d}{2}\frac{|\mathcal{S}_*^{l,t} \cap \mathcal{N}_z^l| - |\mathcal{S}_\#^{l,t} \cap \mathcal{N}_z^l|}{K|\mathcal{S}^{l,t}|}. \tag{38}$$

**Lemma 7.** *For the update of $\boldsymbol{W}_V^{(t)}$, there exists $\lambda \leq \Theta(1)$ such that*

$$\boldsymbol{W}_V^{(t)}\boldsymbol{x}_j = \boldsymbol{p}_1 - \eta\sum_{b=1}^{t}(\sum_{i \in \mathcal{W}_n(0)}V_i(b)\boldsymbol{W}_{O_{(i,\cdot)}}^{(b)}{}^\top + \sum_{i \notin \mathcal{W}_n(0)}\lambda V_i(b)\boldsymbol{W}_{O_{(i,\cdot)}}^{(b)}{}^\top) + \boldsymbol{z}_j(t), \quad j \in \mathcal{S}_1^{n,t}, \tag{39}$$

$$\boldsymbol{W}_V^{(t)}\boldsymbol{x}_j^n = \boldsymbol{p}_2 - \eta\sum_{b=1}^{t}(\sum_{i \in \mathcal{U}(0)}V_i(b)\boldsymbol{W}_{O_{(i,\cdot)}}^{(b)}{}^\top + \sum_{i \notin \mathcal{U}(0)}\lambda V_i(b)\boldsymbol{W}_{O_{(i,\cdot)}}^{(b)}{}^\top) + \boldsymbol{z}_j(t), \quad j \in \mathcal{S}_2^{n,t}, \tag{40}$$

$$\boldsymbol{W}_V^{(t+1)}\boldsymbol{x}_j^n = \boldsymbol{p}_l - \eta\sum_{b=1}^{t}\sum_{i=1}^{m}\lambda V_i(b)\boldsymbol{W}_{O_{(i,\cdot)}}^{(b)}{}^\top + \boldsymbol{z}_j(t), \quad j \in \mathcal{S}^{n,t}\backslash(\mathcal{S}_1^{n,t} \cup \mathcal{S}_2^{n,t}), \tag{41}$$

$$\|\boldsymbol{z}_j(t)\| \leq (\sigma + \tau), \tag{42}$$

*with*

$$W_l(t) \leq \nu_n(t)|\mathcal{S}_j^n|, \quad l \in \mathcal{S}_j^n, \tag{43}$$

$$V_i(t) \lesssim \frac{1-2p}{2B}\sum_{n \in \mathcal{B}_{b+}} -\frac{1}{a}p_n(t), \quad i \in \mathcal{W}_l(0), \tag{44}$$

$$V_i(t) \gtrsim \frac{1-2p}{2B}\sum_{n \in \mathcal{B}_{b-}}\frac{1}{a}p_n(t), \quad i \in \mathcal{U}_l(0), \tag{45}$$

$$V_i(t) \geq -\frac{1}{\sqrt{B}a}, \quad \text{if } i \text{ is an unlucky neuron.} \tag{46}$$

**Lemma 8.** *(Li et al., 2023a) If the number of neurons $m$ is larger enough such that*

$$m \geq M^2 \log N, \tag{47}$$

*the number of lucky neurons at the initialization $|\mathcal{W}_l(0)|$, $|\mathcal{U}_l(0)|$ satisfies*

$$|\mathcal{W}_l(0)|, \; |\mathcal{U}+l(0)| \geq \Omega(m). \tag{48}$$

**Lemma 9.** *Under the condition that $m \gtrsim M^2 \log N$, we have the following result. For $i \in \mathcal{W}_l(0)$ and $l \in \mathcal{D}_1$, we have*

$$\mathbb{1}[\boldsymbol{W}_{O_{(i,\cdot)}}^{(t)} \boldsymbol{V}_l(t)] = 1; \tag{49}$$

*For $i \in \mathcal{U}_l(0)$ and $l \in \mathcal{D}_2$, we have*

$$\mathbb{1}[\boldsymbol{W}_{O_{(i,\cdot)}}^{(t)} \boldsymbol{V}_l(t)] = 1; \tag{50}$$

**Proof of Theorem 4.1:**
Denote the set of neurons with positive $a_i$ as $\mathcal{K}_+$ and the set of neurons with negative $a_i$ as $\mathcal{K}_-$. For $y_n = 1$, recall from (10) and Definition 4, we have

$$F(\boldsymbol{x}_n) = \sum_{i \in \mathcal{W}_n(0)} \frac{1}{a} \text{Relu}(\boldsymbol{W}_{O_{(i)}}^{(t)} \boldsymbol{V}_n(t)) + \sum_{i \in \mathcal{K}_+/\mathcal{W}_n(0)} \frac{1}{a} \text{Relu}(\boldsymbol{W}_{O_{(i)}}^{(t)} \boldsymbol{V}_n(t))$$
$$- \sum_{i \in \mathcal{K}_-} \frac{1}{a} \text{Relu}(\boldsymbol{W}_{O_{(i)}}^{(t)} \boldsymbol{V}_n(t)). \tag{51}$$

Therefore,

$$\sum_{i \in \mathcal{W}_n(0)} \frac{1}{a} \text{Relu}(\boldsymbol{W}_{O_{(i)}}^{(t)} \boldsymbol{V}_n(t))$$
$$= \sum_{i \in \mathcal{W}_n(0)} \frac{1}{a} \text{Relu}(\boldsymbol{W}_{O_{(i)}}^{(t)} \boldsymbol{V}_n(t)) + \sum_{i \in \mathcal{W}_n(t)} \frac{1}{a} \text{Relu}(\boldsymbol{W}_{O_{(i)}}^{(t)} \boldsymbol{V}_n(t))$$
$$\gtrsim \frac{1}{a} \cdot \boldsymbol{W}_{O_{(i,\cdot)}}^{(t)} \Big( \sum_{s \in \mathcal{S}_1^{n,t}} \boldsymbol{p}_s \text{softmax}_n(\boldsymbol{x}_s{}^\top \boldsymbol{W}_K^{(t)\top} \boldsymbol{W}_Q^{(t)} \boldsymbol{x}_n) + \boldsymbol{z}(t) + \sum_{l \neq s} W_l(u) \boldsymbol{p}_l$$
$$- \eta t \big( \sum_{j \in \mathcal{W}_n(0)} V_j(t) \boldsymbol{W}_{O_{(j,\cdot)}}^{(t)}{}^\top + \sum_{j \notin \mathcal{W}_n(0)} V_j(t) \lambda \boldsymbol{W}_{O_{(j,\cdot)}}^{(t)}{}^\top \big) \Big) |\mathcal{W}_n(0)| + 0 \tag{52}$$
$$\gtrsim \frac{m}{a} \Big( \frac{1}{B} \sum_{n \in \mathcal{B}_b} \frac{\xi \eta t^2 m}{a^2} \big( \frac{1-2p}{4B} \sum_{n \in \mathcal{B}_b} p_n(b) - \sigma - \tau \big) p_n(t) + \eta m \frac{1-2p}{2B} \sum_{b=1}^{t} \sum_{n \in \mathcal{B}_{b+}} \frac{1}{a} p_n(b)$$
$$\cdot \big( \frac{\xi}{aB} \sum_{n \in \mathcal{B}_b} \frac{\eta t^2 (1-2p)}{4B} \sum_{n \in \mathcal{B}_b} \frac{m}{a} p_n(t))^2 \Big),$$

where the second step results from the formulation of $\boldsymbol{V}_n(t)$ in (21) and the last step is by (140). Meanwhile, we have

$$\sum_{i \in \mathcal{K}_+/\mathcal{W}_n(0)} \frac{1}{a} \text{Relu}(\boldsymbol{W}_{O_{(i)}}^{(t)} \boldsymbol{V}_n(t)) \geq 0. \tag{53}$$

To deal with the upper bound of the third term in (51), we have

$$\Big| \sum_{i \in \mathcal{K}_-} \frac{1}{a} \text{Relu}(\boldsymbol{W}_{O_{(i)}}^{(t)} \boldsymbol{V}_n(t)) \Big| \lesssim \sum_{i \in \mathcal{K}_+} \frac{1}{a} \text{Relu}(\boldsymbol{W}_{O_{(i)}}^{(t)} \boldsymbol{V}_n(t)). \tag{54}$$

Note that at the $t$-th iteration,

$$
\begin{aligned}
&K(t)\\
&\gtrsim \eta \frac{1}{B}\sum_{n\in\mathcal{B}_b}\frac{m}{a}\Big(\frac{1-2p}{B}\sum_{n\in\mathcal{B}_b}\frac{\xi\eta(t+1)^2m}{a^2}(\frac{1}{4B}\sum_{n\in\mathcal{B}_b}p_n(b)-\sigma-\tau)+\eta m\frac{1}{2B}\sum_{b=1}^{t}\sum_{n\in\mathcal{B}_b}\frac{p_n(b)}{a}\\
&\quad\cdot(1-(\sigma+\tau))(\frac{\xi}{aB}\sum_{n\in\mathcal{B}_b}\frac{(1-2p)\eta(t+1)^2}{4B}\sum_{n\in\mathcal{B}_b}\frac{m}{a}p_n(t))^2\Big)\phi_n(t)(|\mathcal{S}^{l,t}|-|\mathcal{S}_1^{l,t}|)\|\boldsymbol{q}_1(t)\|^2\\
&\gtrsim \frac{1}{e^{\|\boldsymbol{q}_1(t)\|^2-(\delta+\tau)\|\boldsymbol{q}_1(t)\|}}.
\end{aligned}
\tag{55}
$$

Since that

$$
\begin{aligned}
\boldsymbol{q}_1(T) &\gtrsim (1+\min_{l=0,1,\cdots,T-1}\{K(l)\})^T\\
&\gtrsim (1+\frac{1}{e^{\|\boldsymbol{q}_1(T)\|^2-(\delta+\tau)\|\boldsymbol{q}_1(T)\|}})^T.
\end{aligned}
\tag{56}
$$

To find the order-wise lower bound of $\boldsymbol{q}_1(T)$, we need to check the equation

$$
\boldsymbol{q}_1(T) \lesssim (1+\frac{1}{e^{\|\boldsymbol{q}_1(T)\|^2-(\delta+\tau)\|\boldsymbol{q}_1(T)\|}})^T.
\tag{57}
$$

One can obtain

$$
\Theta(\sqrt{\log T(1-\delta-\tau)}) = \boldsymbol{q}_1(T) \le \Theta(T).
\tag{58}
$$

We require that

$$
\begin{aligned}
&\frac{m}{a}\Big(\frac{1}{B}\sum_{n\in\mathcal{B}_b}\frac{\xi\eta T^2m}{a^2}(\frac{1-2p}{4B}\sum_{n\in\mathcal{B}_b}p_n(b)-\sigma-\tau)p_n(T)+\eta m\frac{1-2p}{2B}\sum_{b=1}^{T}\sum_{n\in\mathcal{B}_{b+}}\frac{1}{a}p_n(b)\\
&\quad\cdot(\frac{\xi}{aB}\sum_{n\in\mathcal{B}_b}\frac{\eta T^2(1-2p)}{4B}\sum_{n\in\mathcal{B}_b}\frac{m}{a}p_n(T))^2\Big)\\
&:=a_0\eta^3T^5+a_1\eta T^2,\\
&>1,
\end{aligned}
\tag{59}
$$

where the first step is by letting $a=\sqrt{m}$ and $m\gtrsim M^2\log N$. We replace $p_n(b)$ with $p_n(T)$ because when $b$ achieves the level of $T$, $b^{o_1}p_n(b)^{o_2}$ is the same order as $b^{o_1}$ for $o_1,o_2\ge 0$. Thus,

$$
\sum_{b=1}^{T}b^{o_1}p_n(b)^{o_2} \gtrsim T^{o_1+1}p_n(\Theta(1)\cdot T)^{o_2} \gtrsim T^{o_1+1}p_n(T)^{o_2}.
\tag{60}
$$

We also require

$$
B \gtrsim \Theta(1),
\tag{61}
$$

Note that $p_n(t)$ is dependent on other $\sum_{z\in\mathcal{Z}}\delta_z$ nodes. Hence, we know that each $p_n(T)$ is dependent on other $1+\sum_{z\in\mathcal{Z}}\delta_z^2$ variables of $p_j(T)$ for $j,n\in\mathcal{V}$. It is easy to find that $p_n(T)$ is a 1-sub-gaussian random variable because its absolute value is upper bounded by 1. By Lemma 7 in (Zhang et al., 2020b), we can obtain

$$
\mathbb{E}_{\mathcal{D}}[e^{s(\sum_{n\in\mathcal{L}}p_n(T)-|\mathcal{L}|\mathbb{E}_{\mathcal{D}}[p_n(T)])}] \le e^{|\mathcal{L}|(1+\sum_{z\in\mathcal{Z}}\delta_z^2)s^2}.
\tag{62}
$$

When $\eta T=\Theta(1)$, we have $|b_z(T)|=\Theta(1)$ and $|b_z(T)-b_{z'}(T)|\le\Theta(1)$. Therefore, when $n\in\mathcal{D}_1\cup\mathcal{D}_2$, we have

$$
\begin{aligned}
&p_n(T)\\
&=\frac{\sum_{z\in\mathcal{Z}}|\mathcal{N}_z^n\cap\mathcal{S}_*^{n,T}|e^{\|\boldsymbol{q}_1(T)\|^2-(\sigma+\tau)\|\boldsymbol{q}_1(T)\|+b_z^{(T)}}}{\sum_{z\in\mathcal{Z}}|\mathcal{N}_z^n\cap\mathcal{S}_*^{n,T}|e^{\|\boldsymbol{q}_1(T)\|^2-(\sigma+\tau)\|\boldsymbol{q}_1(T)\|+b_z^{(T)}}+\sum_{z\in\mathcal{Z}}|(\mathcal{N}_z^n\cap\mathcal{S}^{n,T})-\mathcal{S}_1^{n,T}|e^{b_z^{(T)}}}\\
&\ge 1-\eta^C.
\end{aligned}
\tag{63}
$$

When $n \notin \mathcal{D}_1 \cup \mathcal{D}_2$, we have

$$
\begin{aligned}
&p_n(T) \\
&= \sum_{z \in \mathcal{Z}} |\mathcal{N}_z^n \cap \mathcal{S}_*^{n,T}| e^{\|\boldsymbol{q}_1(T)\|^2 - (\sigma+\tau)\|\boldsymbol{q}_1(T)\| + b_z^{(T)}} \left( \sum_{z \in \mathcal{Z}} (|\mathcal{N}_z^n \cap \mathcal{S}_*^{n,T}| + |\mathcal{N}_z^n \cap \mathcal{S}_\#^{n,T}|) \right. \\
&\quad \left. \cdot e^{\|\boldsymbol{q}_1(T)\|^2 - (\sigma+\tau)\|\boldsymbol{q}_1(T)\| + b_z^{(T)}} + \sum_{z \in \mathcal{Z}} |(\mathcal{N}_z^n \cup \mathcal{S}^{n,T})/(\mathcal{S}_1^{n,T} \cup \mathcal{S}_2^{n,T})| e^{b_z^{(T)}} \right)^{-1} \\
&\geq \sum_{z \in \mathcal{Z}} |\mathcal{N}_z^n \cap \mathcal{S}_*^{n,T}| (T(1-\delta-\tau))^C e^{b_z^{(T)} - b_{z_m}^{(T)}} \left( \sum_{z \in \mathcal{Z}} (|\mathcal{N}_z^n \cap \mathcal{S}_*^{n,T}| + |\mathcal{N}_z^n \cap \mathcal{S}_\#^{n,T}|) \right. \quad (64) \\
&\quad \left. \cdot (T(1-\delta-\tau))^C e^{b_z^{(T)} - b_{z_m}^{(T)}} + \sum_{z \in \mathcal{Z}} |(\mathcal{N}_z^n \cup \mathcal{S}^{n,T})/(\mathcal{S}_1^{n,T} \cup \mathcal{S}_2^{n,T})| e^{b_z^{(T)} - b_{z_m}^{(T)}} \right)^{-1} \\
&\geq \frac{|\mathcal{N}_{z_m}^n \cap \mathcal{S}_\#^{n,T}|}{(|\mathcal{N}_{z_m}^n \cap \mathcal{S}_*^{n,T}| + |\mathcal{N}_{z_m}^n \cap \mathcal{S}_\#^{n,T}|)} - (T(1-\delta-\tau))^{-C} e^{-b_z(T)}.
\end{aligned}
$$

$$
\begin{aligned}
\zeta &= \mathbb{E}_\mathcal{D}[p_n(T)] \\
&\geq (1-\gamma_d) \cdot \mathbb{E}_\mathcal{D} \left[ \frac{\sum_{z \in \mathcal{Z}} (T(1-\delta-\tau))^C |\mathcal{S}_*^{n,T} \cap \mathcal{N}_z^n| e^{b_z(T)}}{\sum_{z \in \mathcal{Z}} (T(1-\delta-\tau))^C |(\mathcal{S}_1^{n,T} \cup \mathcal{S}_2^{n,T}) \cap \mathcal{N}_z^n| e^{b_z(T)} + \Theta(1)} \right] \\
&\quad + \gamma_d \cdot \frac{(T(1-\delta-\tau))^C}{(T(1-\delta-\tau))^C + \Theta(1)} \\
&\geq (1-\gamma_d)(1 - \epsilon_\mathcal{S} - (T(1-\delta-\tau))^{-C} e^{-b_z(T)}) + \gamma_d(1 - \eta^C) \\
&\gtrsim 1 - \epsilon_\mathcal{S}(1-\gamma_d) - \eta^C.
\end{aligned}
\tag{65}
$$

Hence, define

$$
p_n(T) \geq p_n'(T) := \begin{cases} 1, & \text{if } n \in \mathcal{D}_1 \cup \mathcal{D}_2 \\ \frac{|\mathcal{N}_{z_m}^n \cap \mathcal{S}_\#^{n,T}|}{(|\mathcal{N}_{z_m}^n \cap \mathcal{S}_*^{n,T}| + |\mathcal{N}_{z_m}^n \cap \mathcal{S}_\#^{n,T}|)}, & \text{if } n \notin \mathcal{D}_1 \cup \mathcal{D}_2. \end{cases}
\tag{66}
$$

Therefore,

$$
|\mathbb{E}_{t \geq 0, n \in \mathcal{V}}[p_n'(T)] - 1| \leq (1-\gamma_d) \mathbb{E}_{n \notin (\mathcal{D}_1 \cup \mathcal{D}_2)} \left[ \frac{|\mathcal{N}_{z_m}^n \cap \mathcal{S}_\#^{n,T}|}{(|\mathcal{N}_{z_m}^n \cap \mathcal{S}_*^{n,T}| + |\mathcal{N}_{z_m}^n \cap \mathcal{S}_\#^{n,T}|)} \right] = (1-\gamma_d)\epsilon_\mathcal{S}.
\tag{67}
$$

We can also derive

$$
\mathbb{E}_{n \in \mathcal{L}} \left[ (1 - p_n'(T)^2) \right] \leq 2\mathbb{E}_\mathcal{D} \left[ 1 - p_n'(T) \right] \leq 2(1-\gamma_d)\epsilon_\mathcal{S},
\tag{68}
$$

where the first inequality is by $1 - (p_n'(T))^2 \leq (1 - p_n'(T))(1 + p_n'(T)) \leq 2(1 - p_n'(T))$. We have

$$
\begin{aligned}
&\left| \frac{1}{|\mathcal{L}|} \sum_{n \in \mathcal{L}} (p_n'(T) - (\sigma+\tau)) p_n'(T) - 1 \right| \\
&\leq \left| \frac{1}{|\mathcal{L}|} \sum_{n \in \mathcal{L}} (p_n'(T) - (\sigma+\tau)) p_n'(T) - \mathbb{E}_{n \in \mathcal{L}}[(p_n'(T) - (\sigma+\tau)) p_n'(T)] \right| \\
&\quad + \mathbb{E}_{n \in \mathcal{L}}[|1 - p_n'(T)^2|] + \mathbb{E}_{n \in \mathcal{L}}[(\sigma+\tau) p_n'(T)] \\
&\lesssim \sqrt{\frac{(1 + \delta_{z_m}^2) \cdot \log N}{|\mathcal{L}|}} + 2(1-\gamma_d)\epsilon_\mathcal{S} + (\sigma+\tau),
\end{aligned}
\tag{69}
$$

$$
\left| \frac{1}{|\mathcal{L}|} \sum_{n \in \mathcal{L}} p_n(T)^2 - 1 \right| \lesssim \sqrt{\frac{(1 + \delta_{z_m}^2) \cdot \log N}{|\mathcal{L}|}} + 2(1-\gamma_d)\epsilon_\mathcal{S},
\tag{70}
$$

$$
\left| \frac{1}{|\mathcal{L}|} \sum_{n \in \mathcal{L}} p_n(T) - 1 \right| \lesssim \sqrt{\frac{(1 + \delta_{z_m}^2) \cdot \log N}{|\mathcal{L}|}} + (1-\gamma_d)\epsilon_\mathcal{S}.
\tag{71}
$$

We can then have

$$T = \frac{\eta^{-\frac{1}{2}}(1-\delta-\tau)^{-\frac{1}{2}}}{\sqrt{a_1}} = \frac{\eta^{-\frac{1}{2}}(1-\delta-\tau)^{-\frac{1}{2}}}{(1-2p)^{\frac{1}{2}}}. \tag{72}$$

As long as

$$|\mathcal{L}| \geq \max\{\Omega(\frac{(1+\delta_{z_m}^2) \cdot \log N}{(1-2(1-\gamma_d)\epsilon_{\mathcal{S}}-(\sigma+\tau))^2}), BT\}. \tag{73}$$

we can obtain

$$F(\boldsymbol{x}_n) > 1. \tag{74}$$

Similarly, we can derive that for $y_n = -1$,

$$F(\boldsymbol{x}_n) < -1. \tag{75}$$

Hence, for all $n \in \mathcal{V}$,

$$\text{Loss}(\boldsymbol{x}_n, y_n) = 0. \tag{76}$$

We also have

$$\mathbb{E}_{(\boldsymbol{x}_n, y_n) \sim \mathcal{D}}[\text{Loss}(\boldsymbol{x}_n, y_n)] = 0, \tag{77}$$

with the conditions of sample complexity and the number of iterations.

**Proof of Lemma 1:**
This Lemma is proved by (63) and (64).

**Proof of Theorem 4.2:**
The main proof idea is similar to the proof of Theorem 4.1. A major difference is that the aggregation matrix does not update, i.e., $p_n(t)$ stays at $t = 0$. Since that a given core neighborhood and a $\gamma_d = \Theta(1)$ fraction of discriminative nodes still ensures non-trivial attention weights correlated with class-relevant nodes along the training, the updates of $\boldsymbol{W}_O$ and $\boldsymbol{W}_V$ are order-wise the same as Lemmas 4 and 7.
Since that

$$p_n(0) = \begin{cases} \frac{\sum_{z \in \mathcal{Z}}|\mathcal{S}_*^{n,t} \cap \mathcal{N}_z^n|}{\sum_{z \in \mathcal{Z}}|\mathcal{S}_*^{n,t} \cap \mathcal{N}_z^n| + \sum_{z \in \mathcal{Z}}(|\mathcal{N}_z^n| - |\mathcal{S}_*^{n,t} \cap \mathcal{N}_z^n|)e^{-1}}, & \text{if } n \in \mathcal{S}_1^{n,t} \cup \mathcal{S}_2^{n,t} \\ \frac{\sum_{z \in \mathcal{Z}}|\mathcal{S}_*^{n,t} \cap \mathcal{N}_z^n|}{\sum_{z \in \mathcal{Z}}(|\mathcal{S}^{n,t}| - |\mathcal{S}^{n,t} \cap \mathcal{N}_z^n|) + \sum_{z \in \mathcal{Z}}|\mathcal{N}_z^n \cap \mathcal{S}_l^{n,t}|e}, & \text{if } n \notin (\mathcal{S}_1^{n,t} \cup \mathcal{S}_2^{n,t}) \end{cases} \tag{78}$$
$$= \Theta(1),$$

there exists $c_\gamma > 0$, such that

$$\mathbb{E}[p_n(0)] = \gamma_d \cdot \Theta(\gamma_d) + (1-\gamma_d)\Theta(\frac{\gamma_d}{2}) = c_\gamma \gamma_d, \tag{79}$$

$$\mathbb{E}[|p_n(0) \pm c_\gamma \gamma_d|^2] \leq \gamma_d \cdot \Theta(\gamma_d^2) + (1-\gamma_d) \cdot \Theta(|\gamma_d \pm \frac{1}{2}|^2 \gamma_d^2) \leq \Theta(\gamma_d^2). \tag{80}$$

Therefore,

$$\begin{aligned} &\left|\frac{1}{|\mathcal{L}|}\sum_{n=1}^{N} p_n(T)(p_n(T) - (\sigma+\tau)) - c_\gamma^2 \gamma_d^2\right| \\ \leq& \left|\frac{1}{|\mathcal{L}|}\sum_{n=1}^{N} p_n(0)(p_n(0) - (\sigma+\tau)) - \mathbb{E}\Big[p_n(0)(p_n(0) - (\sigma+\tau))\Big]\right| \\ &+ \left|\mathbb{E}\Big[|p_n(0)^2 - (\sigma+\tau)p_n(0) - c_\gamma^2 \gamma_d^2|\Big]\right| \\ \lesssim& \sqrt{\frac{\log N}{|\mathcal{L}|}} + (\sigma+\tau) + \sqrt{\mathbb{E}\Big[|p_n(0) + c_\gamma \gamma_d|^2\Big] \cdot \mathbb{E}\Big[|p_n(0) - c_\gamma \gamma_d|^2\Big]} \\ \lesssim& \sqrt{\frac{\log N}{|\mathcal{L}|}} + (\sigma+\tau) + \Theta(\gamma_d^2), \end{aligned} \tag{81}$$

where the first step is because $p_n(T)$ does not update since $\boldsymbol{W}_K^{(t)}$ and $\boldsymbol{W}_Q^{(t)}$ are fixed at initialization $\boldsymbol{W}_K^{(0)}$ and $\boldsymbol{W}_Q^{(0)}$, and the second step is by Cauchy-Schwarz inequality. Since that

$$\sqrt{\frac{\log N}{N}} + (\sigma + \tau) \leq \Theta(\gamma_d^2), \tag{82}$$

we have

$$|\mathcal{L}| \geq \Omega\left(\frac{(1 + \delta_{z_m}^2)\log N}{(\gamma_d^2 - (\sigma + \tau))^2}\right), \tag{83}$$

and

$$T = \frac{\eta^{-\frac{1}{2}}}{(1 - 2p)^{\frac{1}{2}}(1 - \delta - \tau)^{\frac{1}{2}}\gamma_d^2}. \tag{84}$$

**Proof of Lemma 2:**
When $t = T$, we have $\eta T \geq \Theta(1)$. Since that

$$\left|\sum_{b=1}^{T}\sum_{n\in\mathcal{B}_b}\frac{|\mathcal{S}_*^{n,T}\cap\mathcal{N}_z^n|}{|\mathcal{S}^{n,T}|} - \sum_{b=1}^{T}\sum_{n\in\mathcal{B}_b}\frac{|\mathcal{D}_*^n\cap\mathcal{N}_z^n|}{N}\right| \leq \epsilon_0, \tag{85}$$

with high probability for some $1 > \epsilon_0 > 0$, we can derive (13).

**Proof of Theorem 4.3:**
When $\boldsymbol{b} = 0$ is fixed during the training, but $\mathcal{S}^{n,t}$ and $\mathcal{T}^n$ are subsets of $\mathcal{N}_{z_m}^n$, the bound for $p_n(T)$ is still the same as in (63) and (64). Given a known core neighborhood in Theorem 4.3, the remaining parameters follow the same order-wise update as Lemmas 4, 5 and 7. The remaining proof steps just follow the remaining contents in the proof of Theorem 4.1.

## D  USEFUL LEMMAS

We prove Lemma 4, 5, 7, and 6 jointly by induction. Lemma 4 first studies the gradient update of lucky neurons in $\mathcal{W}_l(t)$ in directions of $\boldsymbol{p}_1$, $\boldsymbol{p}_2$, and other $\boldsymbol{p}$. We divide the updates into several terms and solve each of them. By applying a known result of PDE, we bound the component in the direction of $\boldsymbol{p}_1$, which is the most important one. The updates of other neurons follow the above procedure. Lemma 5 computes the gradient update of $\boldsymbol{W}_Q$ and $\boldsymbol{W}_K$ in different directions of $\boldsymbol{x}_l$. By controlling the gradient update to be positive in the directions of discriminative nodes, we get a lower bound of $B$. Meanwhile, we obtain the update of key and query embeddings. Lemma 7 is derived by considering different components of $\boldsymbol{W}_{O_{(i,\cdot)}}$ in the gradient. In proving Lemma 6, we characterize the update of different distance $z$ in terms of components from different neighborhoods. Combining concentration bounds, we remove the influence on unimportant terms and only retain one part, which represents the update of the average winning margin of the majority vote, i.e., the update of $\bar{\Delta}(z)$. For Lemma 9, We characterize the updates of the lucky neurons to the desired directions to show lucky neurons can activate the self-attention output of discriminative nodes along the training.

**Proof of Lemma 4:**
At the $t$-th iteration, if $s \in \mathcal{S}_1^{n,t}$, we can obtain

$$\begin{aligned}
\boldsymbol{V}_n(t) &= \sum_{s\in\mathcal{S}^{l,t}} \boldsymbol{W}_V^{(t)}\boldsymbol{x}_s\text{softmax}_n(\boldsymbol{x}_s^\top\boldsymbol{W}_K^{(t)^\top}\boldsymbol{W}_Q^{(t)}\boldsymbol{x}_n + \boldsymbol{u}_{(s,n)}^\top\boldsymbol{b}^{(t)}) \\
&= \sum_{s\in\mathcal{S}_1} \text{softmax}_n(\boldsymbol{x}_s^\top\boldsymbol{W}_K^{(t)^\top}\boldsymbol{W}_Q^{(t)}\boldsymbol{x}_n + \boldsymbol{u}_{(s,n)}^\top\boldsymbol{b}^{(t)})\boldsymbol{p}_1 + \boldsymbol{z}(t) + \sum_{j\neq 1}W_j^l(t)\boldsymbol{p}_j \\
&\quad - \eta\sum_{b=1}^{t}\left(\sum_{i\in\mathcal{W}_n(0)}V_i(b)\boldsymbol{W}_{O_{(i,\cdot)}}^{(b)^\top} + \sum_{i\notin\mathcal{W}_l(b)}V_i(b)\lambda\boldsymbol{W}_{O_{(i,\cdot)}}^{(b)^\top}\right),
\end{aligned} \tag{86}$$

$l \in [M]$, where the last step comes from Lemma 7. Then we can derive that for $k \in \mathcal{S}_j^{n,t}$,

$$W_k^n(t) \leq \frac{\sum_{z\in\mathcal{Z}}|\mathcal{S}_j^{n,t}\cap\mathcal{N}_z^n|e^{\delta\|\boldsymbol{q}_1(t)\|+b_z^{(t)}}}{\sum_{z\in\mathcal{Z}}|\mathcal{S}_j^{n,t}\cap\mathcal{N}_z^n|e^{\|\boldsymbol{q}_1(t)\|^2-(\sigma+\delta)\|\boldsymbol{q}_1(t)\|+b_z^{(t)}}}p_n(t), \tag{87}$$

which is much smaller than $\Theta(1)$ when $t$ is large. This is the reason why we ignore the impact of $W_l(t)$ on $\eta \sum_{b=0}^{t-1} (\sum_{i \in \mathcal{W}_l(0)} V_i(b){\boldsymbol{W}_{O_{(i,\cdot)}}^{(b)}}^{\top} + \sum_{i \notin \mathcal{W}_l(0)} V_i(b)\lambda {\boldsymbol{W}_{O_{(i,\cdot)}}^{(b)}}^{\top})$. Hence,

$$\frac{1}{B}\sum_{n \in \mathcal{B}_b}\frac{\partial \mathrm{Loss}(\boldsymbol{x}_n, y_n)}{{\partial \boldsymbol{W}_{O_{(i)}}}^{\top}} = -\frac{1}{B}\sum_{n \in \mathcal{B}_b}y_n\sum_{l \in \mathcal{S}^{n,t}}a_i \mathbb{1}[\boldsymbol{W}_{O_{(i)}}\boldsymbol{V}_l(t) \geq 0]\boldsymbol{V}_l(t)^{\top}. \tag{88}$$

Denote that for $j \in [M]$,

$$H_4 = \frac{1}{B}\sum_{n \in \mathcal{B}_b}\eta y_n a_i \mathbb{1}[\boldsymbol{W}_{O_{(i)}}^{(t)}\boldsymbol{V}_l(t) \geq 0](-\eta)\sum_{b=1}^{t}\sum_{k \in \mathcal{W}_l(b)}V_k(b)\boldsymbol{W}_{O_{(k,\cdot)}}^{(b)}\boldsymbol{p}_j, \tag{89}$$

$$H_4 = \frac{1}{B}\sum_{n \in \mathcal{B}_b}\eta y_n a_i \mathbb{1}[\boldsymbol{W}_{O_{(i)}}^{(t)}\boldsymbol{V}_l(t) \geq 0](-\eta)\sum_{b=1}^{t}\sum_{k \notin \mathcal{W}_l(b)}V_k(b)\boldsymbol{W}_{O_{(k,\cdot)}}^{(b)}\boldsymbol{p}_j, \tag{90}$$

and we can then derive

$$\left\langle {\boldsymbol{W}_{O_{(i)}}^{(t+1)}}^{\top}, \boldsymbol{p}_j \right\rangle - \left\langle {\boldsymbol{W}_{O_{(i)}}^{(t)}}^{\top}, \boldsymbol{p}_j \right\rangle$$

$$=\frac{1}{B}\sum_{l \in \mathcal{B}_b}\eta y_n a_i \mathbb{1}[\boldsymbol{W}_{O_{(i)}}^{(t)}\boldsymbol{V}_l(t) \geq 0]\boldsymbol{V}_l(t)^{\top}\boldsymbol{p}_j$$

$$=\frac{1}{B}\sum_{l \in \mathcal{B}_b}\eta y_n a_i \mathbb{1}[\boldsymbol{W}_{O_{(i)}}^{(t)}\boldsymbol{V}_l(t) \geq 0]\boldsymbol{z}_l(t)^{\top}\boldsymbol{p}_j$$

$$+\frac{1}{B}\sum_{l \in \mathcal{B}_b}\eta y_n a_i \mathbb{1}[\boldsymbol{W}_{O_{(i)}}^{(t)}\boldsymbol{V}_l(t) \geq 0]\sum_{s \in \mathcal{S}_l}\mathrm{softmax}_l(\boldsymbol{x}_s^{\top}{\boldsymbol{W}_K^{(t)}}^{\top}\boldsymbol{W}_Q^{(t)}\boldsymbol{x}_l + \boldsymbol{u}_{(s,l)}^{\top}\boldsymbol{b}^{(t)})\boldsymbol{p}_l^{\top}\boldsymbol{p}_j \tag{91}$$

$$+\frac{1}{B}\sum_{l \in \mathcal{B}_b}\eta y_n a_i \mathbb{1}[\boldsymbol{W}_{O_{(i)}}^{(t)}\boldsymbol{V}_l(t) \geq 0]\sum_{k \neq l}W_l(t)\boldsymbol{p}_k^{\top}\boldsymbol{p}_j + H_4 + H_4$$

$$:=H_1 + H_2 + H_3 + H_4 + H_4,$$

where

$$H_1 = \frac{1}{B}\sum_{n \in \mathcal{B}_b}\eta y_n a_i \mathbb{1}[\boldsymbol{W}_{O_{(i)}}^{(t)}\boldsymbol{V}_l(t) \geq 0]\boldsymbol{z}_l(t)^{\top}\boldsymbol{p}_j, \tag{92}$$

$$H_2 = \frac{1}{B}\sum_{n \in \mathcal{B}_b}\eta y_n a_i \mathbb{1}[\boldsymbol{W}_{O_{(i)}}^{(t)}\boldsymbol{V}_l(t) \geq 0]\sum_{s \in \mathcal{S}_l}\mathrm{softmax}_l(\boldsymbol{x}_s^{\top}{\boldsymbol{W}_K^{(t)}}^{\top}\boldsymbol{W}_Q^{(t)}\boldsymbol{x}_l + \boldsymbol{u}_{(s,l)}^{\top}\boldsymbol{b}^{(t)})\boldsymbol{p}_l^{\top}\boldsymbol{p}_j, \tag{93}$$

$$H_3 = \frac{1}{B}\sum_{n \in \mathcal{B}_b}\eta y_n a_i \mathbb{1}[\boldsymbol{W}_{O_{(i)}}^{(t)}\boldsymbol{V}_l(t) \geq 0]\sum_{j \neq l}W_l(t)\boldsymbol{p}_j^{\top}\boldsymbol{p}_j. \tag{94}$$

We then show the statements in different cases.
(1) When $j = 1$, since that $\mathrm{Pr}(y_n = 1) = \mathrm{Pr}(y_n = -1) = 1/2$, by Hoeffding's inequality in (19), one can obtain

$$\mathrm{Pr}\left(\left|\frac{1}{B}\sum_{n \in \mathcal{B}_b}y_n\right| \geq \sqrt{\frac{\log B}{B}}\right) \leq B^{-c}, \tag{95}$$

$$\mathrm{Pr}\left(\left|\boldsymbol{z}_l(t)^{\top}\boldsymbol{p}_1\right| \geq \sqrt{((\sigma + \tau))^2 \log m}\right) \leq m^{-c}. \tag{96}$$

Hence, with a high probability, we have

$$|H_1| \leq \frac{\eta((\sigma + \tau))}{a}\sqrt{\frac{\log m \log B}{B}}. \tag{97}$$

For $i \in \mathcal{W}_l(0)$, by the reasoning in (139) later, we can obtain

$$\boldsymbol{W}_{O_{(i,\cdot)}}^{(t)}\sum_{s \in \mathcal{S}^{l,t}}\boldsymbol{W}_V^{(t)}\boldsymbol{x}_s\mathrm{softmax}_l(\boldsymbol{x}_s^{\top}{\boldsymbol{W}_K^{(t)}}^{\top}\boldsymbol{W}_Q^{(t)}\boldsymbol{x}_l + \boldsymbol{u}_{(s,l)}^{\top}\boldsymbol{b}^{(t)}) > 0. \tag{98}$$

Denote $p_n(t) = |\mathcal{S}_1^{n,t}|\nu_n(t)e^{\|\boldsymbol{q}_1(t)\|^2 - 2\delta\|\boldsymbol{q}_1(t)\|}$. Hence, for $k \notin \mathcal{W}_l(0)$,

$$H_2 \gtrsim \eta \cdot \frac{1}{B} \sum_{n \in \mathcal{B}_b} \frac{1}{a} \|\boldsymbol{p}_1\|^2 \cdot p_n(t)(1 - 2p), \tag{99}$$

$$H_3 = 0, \tag{100}$$

$$H_4 \gtrsim \frac{1}{B} \sum_{b=1}^{t} \sum_{n \in \mathcal{B}_b} \frac{\eta^2}{a} \frac{1}{2B} \sum_{n \in \mathcal{B}_b} \frac{m}{a} p_n(t)(1 - 2p)\|\boldsymbol{p}_1\|^2 (1 - \epsilon_m - \frac{(\sigma + \tau)M}{\pi}) \boldsymbol{W}_{O_{(i,\cdot)}} \boldsymbol{p}_1, \tag{101}$$

$$|H_4| \lesssim \frac{1}{B} \sum_{b=1}^{t} \sum_{n \in \mathcal{B}_b} \frac{\eta^2}{a} (1 - \epsilon_m - \frac{(\sigma + \tau)}{\pi}) \frac{1}{2B} \sum_{n \in \mathcal{B}_b} \frac{m}{aM} p_n(t)\|\boldsymbol{p}_1\|^2 \boldsymbol{W}_{O_{(i,\cdot)}} \boldsymbol{p}_2$$
$$+ \frac{\eta^2 tm}{\sqrt{B}a^2} \boldsymbol{W}_{O_{(k,\cdot)}} \boldsymbol{p}_1. \tag{102}$$

Hence, if we combine (97), (99), (100), (101), and (102), we can derive

$$\left\langle \boldsymbol{W}_{O_{(i)}}^{(t+1)^\top}, \boldsymbol{p}_1 \right\rangle - \left\langle \boldsymbol{W}_{O_{(i)}}^{(t)^\top}, \boldsymbol{p}_1 \right\rangle$$

$$\gtrsim \frac{\eta}{a} \cdot \frac{1}{B} \sum_{n \in \mathcal{B}_b} (p_n(t)(1 - 2p) - (\sigma + \tau) + \eta \sum_{b=1}^{t} \frac{1}{2B} \sum_{n \in \mathcal{B}_b} \frac{m}{a} p_n(t)(1 - \epsilon_m - \frac{(\sigma + \tau)M}{\pi})$$

$$\cdot \boldsymbol{W}_{O_{(i,\cdot)}} \boldsymbol{p}_1(1 - 2p) - \eta \sum_{b=1}^{t} \frac{1}{2B} \sum_{n \in \mathcal{B}_b} \frac{m}{a} p_n(t)(1 - \epsilon_m - \frac{(\sigma + \tau)M}{\pi})$$

$$\cdot \boldsymbol{W}_{O_{(i,\cdot)}} \boldsymbol{p}_2(1 + (\sigma + \tau)) - \frac{\eta tm \boldsymbol{W}_{O_{(k,\cdot)}} \boldsymbol{p}_1}{\sqrt{B}a})$$

$$\gtrsim \frac{\eta}{aB} \sum_{n \in \mathcal{B}_b} (p_n(t)(1 - 2p) - (\sigma + \tau) + \frac{\eta t(1 - 2p)}{2B} \sum_{n \in \mathcal{B}_b} \frac{m}{a} p_n(t) \cdot (1 - \epsilon_m - \frac{(\sigma + \tau)M}{\pi})$$

$$\cdot \boldsymbol{W}_{O_{(i,\cdot)}} \boldsymbol{p}_1). \tag{103}$$

Since that $\boldsymbol{W}_{O_{(i,\cdot)}}^{(0)} \sim \mathcal{N}(0, \frac{\xi^2 \boldsymbol{I}}{m_a})$, from the property of Gaussian distribution, we have

$$\Pr(\|\boldsymbol{W}_{O_{(i,\cdot)}}^{(0)}\| \lesssim \xi) \lesssim \xi. \tag{104}$$

Therefore, with high probability for all $i \in [m]$, we can derive

$$\|\boldsymbol{W}_{O_{(i,\cdot)}}^{(0)}\| \gtrsim \xi. \tag{105}$$

When $\eta$ is very small, given $p_n(t)$ as the order of a constant, (103) leads to a PDE on the lower bound of $\boldsymbol{W}_{O_{(i,\cdot)}} \boldsymbol{p}_1$ since the last step of (103) is always positive. Denote $y(t)$ as a lower bound of $\boldsymbol{W}_{O_{(i,\cdot)}} \boldsymbol{p}_1$, we have

$$\frac{\partial y(t)}{\partial t}$$
$$= \Theta(\frac{1}{aB} \sum_{n \in \mathcal{B}_b} (p_n(t)(1 - 2p) - (\sigma + \tau)) + \frac{\eta t(1 - 2p)}{2B} \sum_{n \in \mathcal{B}_b} \frac{m}{a} p_n(t)y(t)). \tag{106}$$

Therefore, we can derive

$$y(t) = e^{\frac{1}{aB} \sum_{n \in \mathcal{B}_b} \frac{\eta t^2 (1 - 2p)}{4B} \sum_{n \in \mathcal{B}_b} \frac{m}{a} p_n(t)} (\int_{-\infty}^{t} \frac{1}{aB} \sum_{n \in \mathcal{B}_b} (p_n(t)(1 - 2p) - (\sigma + \tau))$$

$$\cdot e^{-\frac{1}{aB} \sum_{n \in \mathcal{B}_b} \frac{\eta u^2 (1 - 2p)}{4B} \sum_{n \in \mathcal{B}_b} \frac{m}{a} p_n(t)} du + C_0). \tag{107}$$

Note that

$$
\int_{-\infty}^{t} e^{-\frac{1}{aB}\sum_{n\in\mathcal{B}_b}\frac{\eta u^2(1-2p)}{4B}\sum_{n\in\mathcal{B}_b}\frac{m}{a}p_n(t)} du
$$

$$
\leq \int_{-\infty}^{\infty} e^{-\frac{1}{aB}\sum_{n\in\mathcal{B}_b}\frac{\eta u^2(1-2p)}{4B}\sum_{n\in\mathcal{B}_b}\frac{m}{a}p_n(t)} du \tag{108}
$$

$$
=\sqrt{2\pi}\cdot(\frac{1}{aB}\sum_{n\in\mathcal{B}_b}\frac{\eta(1-2p)}{4B}\sum_{n\in\mathcal{B}_b}\frac{m}{a}p_n(t))^{-1}
$$

$$
=\Theta(\eta^{-1}).
$$

$$
\int_{-\infty}^{t} e^{-\frac{1}{aB}\sum_{n\in\mathcal{B}_b}\frac{\eta u^2(1-2p)}{4B}\sum_{n\in\mathcal{B}_b}\frac{m}{a}p_n(t)} du
$$

$$
\geq \int_{-\infty}^{0} e^{-\frac{1}{aB}\sum_{n\in\mathcal{B}_b}\frac{\eta u^2(1-2p)}{4B}\sum_{n\in\mathcal{B}_b}\frac{m}{a}p_n(t)}) du \tag{109}
$$

$$
=\Theta(\eta^{-1}).
$$

Hence,

$$
y(0) = \frac{\eta^{-1}}{aB}\sum_{n\in\mathcal{B}_b}(p_n(t)(1-2p)-(\sigma+\tau))+C_0 = \Theta(\eta^{-1}\xi)+C_0 = \xi. \tag{110}
$$

$$
C_0 = \xi(1-\Theta(\eta^{-1})). \tag{111}
$$

$$
\boldsymbol{W}_{O_{(i,\cdot)}}^{(t+1)}\boldsymbol{p}_1 \gtrsim y(t)
$$

$$
\gtrsim e^{\frac{1}{aB}\sum_{n\in\mathcal{B}_b}\frac{\eta(t+1)^2(1-2p)}{4B}\sum_{n\in\mathcal{B}_b}\frac{m}{a}p_n(t)}\xi \tag{112}
$$

$$
\gtrsim \frac{\xi}{aB}\sum_{n\in\mathcal{B}_b}\frac{\eta(t+1)^2(1-2p)}{4B}\sum_{n\in\mathcal{B}_b}\frac{m}{a}p_n(t)+\xi.
$$

(2) When $\boldsymbol{p}_j\in\mathcal{P}/p^+$, we have

$$
H_2 = 0, \tag{113}
$$

$$
|H_3| \leq \frac{1}{B}\sum_{n\in\mathcal{B}_b}\nu_n(t)\frac{\eta}{a}\sqrt{\frac{\log m\log B}{B}}\|\boldsymbol{p}\|^2, \tag{114}
$$

$$
|H_4| \leq \frac{\eta^2}{a}\sum_{b=1}^{t}\sqrt{\frac{\log m\log B}{B}}\frac{1}{2B}\sum_{n\in\mathcal{B}_b}\frac{m}{a}p_n(b)\boldsymbol{W}_{O_{(i,\cdot)}}^{\top}\boldsymbol{p}_j. \tag{115}
$$

For $k\notin\mathcal{W}_l(0)$,

$$
|H_5| \lesssim \frac{\eta^2 tm}{\sqrt{B}a^2}\boldsymbol{W}_{O_{(k,\cdot)}}^{\top}\boldsymbol{p}_1 + \frac{\eta^2}{a}\sum_{b=1}^{t}\sqrt{\frac{\log m\log B}{B}}\frac{1}{2B}\sum_{n\in\mathcal{B}_b}\frac{m}{a}p_n(t)\boldsymbol{W}_{O_{(i,\cdot)}}^{\top}\boldsymbol{p}_2, \tag{116}
$$

with high probability. (115) is from (24). Then, combining (97), (113), (114), (115) and (116), we have

$$
\left|\left\langle\boldsymbol{W}_{O_{(i)}}^{(t+1)\top},\boldsymbol{p}_j\right\rangle-\left\langle\boldsymbol{W}_{O_{(i)}}^{(t)\top},\boldsymbol{p}_j\right\rangle\right|
$$

$$
\lesssim \frac{\eta}{a}\cdot\frac{1}{B}\sum_{n\in\mathcal{B}_b}(\nu_n(t)+(\sigma+\tau) \tag{117}
$$

$$
+\sum_{b=1}^{t}\frac{p_n(b)\eta m}{a}\boldsymbol{W}_{O_{(i,\cdot)}}^{\top}\boldsymbol{p}_j)\sqrt{\frac{\log m\log B}{B}}.
$$

Comparing (103) and (117), we have

$$
\boldsymbol{W}_{O_{(i,\cdot)}}^{(t+1)}\boldsymbol{p}_j \lesssim \frac{1}{\sqrt{B}}\boldsymbol{W}_{O_{(i,\cdot)}}^{(t+1)}\boldsymbol{p}_1. \tag{118}
$$

(3) If $i \in \mathcal{U}_l(0)$, from the derivation of (112) and (118), we can obtain

$$\boldsymbol{W}_{O_{(i,\cdot)}}^{(t+1)}\boldsymbol{p}_2 \gtrsim \frac{\xi}{aB}\sum_{n\in\mathcal{B}_b}\frac{\eta(t+1)^2(1-2p)}{4B}\sum_{n\in\mathcal{B}_b}\frac{m}{a}p_n(t) + \xi, \tag{119}$$

$$\boldsymbol{W}_{O_{(i,\cdot)}}^{(t+1)}\boldsymbol{p}_j \lesssim \frac{1}{\sqrt{B}}\boldsymbol{W}_{O_{(i,\cdot)}}^{(t+1)}\boldsymbol{p}_2, \quad \text{for } \boldsymbol{p} \in \mathcal{P}/\boldsymbol{p}_2. \tag{120}$$

(4) If $i \notin (\mathcal{W}_l(0) \cup \mathcal{U}_{l,n}(0))$,

$$|H_2 + H_3| \leq \frac{\eta}{a}\sqrt{\frac{\log m \log B}{B}}\|\boldsymbol{p}\|^2, \tag{121}$$

Following (115) and (116), we have

$$|H_4| \leq \sum_{b=1}^{t}\frac{\eta^2}{a}\sqrt{\frac{\log m \log B}{B}}\frac{1}{2B}\sum_{n\in\mathcal{B}_b}\frac{m}{a}p_n(b)\boldsymbol{W}_{O_{(i,\cdot)}}^{\top}\boldsymbol{p}, \tag{122}$$

$$|H_5| \lesssim \frac{\eta^2 tm}{\sqrt{B}a^2}\boldsymbol{W}_{O_{(k,\cdot)}}^{(t)}\boldsymbol{p}_1 + \sum_{b=1}^{t}\frac{\eta^2}{a}\sqrt{\frac{\log m \log B}{B}}\frac{1}{2B}\sum_{n\in\mathcal{B}_b}\frac{m}{a}p_n(b)\boldsymbol{W}_{O_{(i,\cdot)}}^{(t)}\boldsymbol{p}_2. \tag{123}$$

Thus, combining (121), (122), and (123), we can derive

$$\left|\left\langle\boldsymbol{W}_{O_{(i,\cdot)}}^{(t+1)^\top}, \boldsymbol{p}\right\rangle - \left\langle\boldsymbol{W}_{O_{(i,\cdot)}}^{(t)}{}^\top, \boldsymbol{p}\right\rangle\right|$$
$$\lesssim \frac{\eta}{a}\cdot(\|\boldsymbol{p}\| + (\sigma+\tau) + \sum_{b=1}^{t}\frac{1}{2B}\sum_{n\in\mathcal{B}_b}\frac{p_n(b)\eta m}{a}\boldsymbol{W}_{O_{(i,\cdot)}}^{\top}\boldsymbol{p}_j)\sqrt{\frac{\log m \log B}{B}}, \tag{124}$$

Comparing (103) and (124), we can obtain

$$\boldsymbol{W}_{O_{(i,\cdot)}}^{(t+1)}\boldsymbol{p}_j \lesssim \frac{1}{\sqrt{B}}\boldsymbol{W}_{O_{(j,\cdot)}}^{(t+1)}\boldsymbol{p}_1, \tag{125}$$

for $j \in \mathcal{W}_l(0)$.

(5) In this part, we study the bound of $\boldsymbol{W}_{O_{(i,\cdot)}}^{(t)}$ and the product with the noise term according to the analysis above.

By (42), for the lucky neuron $i$, since that the update of $\boldsymbol{W}_{O_{(i,\cdot)}}^{(t)}$ lies in the subspace spanned by $\mathcal{P}$, we can obtain

$$\|\boldsymbol{W}_{O_{(i,\cdot)}}^{(t+1)}\|^2 = \sum_{l=1}^{M}(\boldsymbol{W}_{O_{(i,\cdot)}}^{(t+1)}\boldsymbol{p}_l)^2 \geq (\boldsymbol{W}_{O_{(i,\cdot)}}^{(t+1)}\boldsymbol{p}_1)^2$$
$$\gtrsim(\frac{\xi}{aB}\sum_{n\in\mathcal{B}_b}\frac{\eta(t+1)^2(1-2p))}{4B}\sum_{n\in\mathcal{B}_b}\frac{m}{a}p_n(t))^2, \tag{126}$$

$$\|\boldsymbol{W}_{O_{(i,\cdot)}}^{(t+1)}\boldsymbol{z}_l(t)\| \leq \left|(\sigma+\tau)\|\boldsymbol{W}_{O_{(i,\cdot)}}^{(t+1)}\|\right|. \tag{127}$$

For the unlucky neuron $i$, we can similarly get

$$\|\boldsymbol{W}_{O_{(i,\cdot)}}^{(t+1)}\|^2 \leq \frac{1}{B}\|\boldsymbol{W}_{O_{(j,\cdot)}}^{(t+1)}\|^2, \tag{128}$$

where $j$ is a lucky neuron. The proof of Lemma 4 finishes here.

**Proof of Lemma 5**:

We first study the gradient of $\boldsymbol{W}_Q^{(t+1)}$ in part (a) and the gradient of $\boldsymbol{W}_K^{(t+1)}$ in part (b).

(a) from (14), we can obtain

$$
\eta \frac{1}{B} \sum_{l \in \mathcal{B}_b} \frac{\partial \mathbf{Loss}(\boldsymbol{X}^n, y_n)}{\partial \boldsymbol{W}_Q}
$$

$$
= \eta \frac{1}{B} \sum_{l \in \mathcal{B}_b} \frac{\partial \mathbf{Loss}(\boldsymbol{X}^n, y_n)}{\partial F(\boldsymbol{X}^n)} \frac{\partial F(\boldsymbol{X}^n)}{\partial \boldsymbol{W}_Q}
$$

$$
= \eta \frac{1}{B} \sum_{l \in \mathcal{B}_b} (-y_n) \sum_{i=1}^m a_i \mathbb{1}[\boldsymbol{W}_{O_{(i,\cdot)}} \sum_{s \in \mathcal{S}^{l,t}} \boldsymbol{W}_V \boldsymbol{x}_s \mathrm{softmax}_l(\boldsymbol{x}_s^\top \boldsymbol{W}_K^\top \boldsymbol{W}_Q \boldsymbol{x}_l + \boldsymbol{u}_{(s,l)}^\top \boldsymbol{b}^{(t)}) \geq 0]
$$

$$
\cdot \Big( \boldsymbol{W}_{O_{(i,\cdot)}} \sum_{s \in \mathcal{S}^{l,t}} \boldsymbol{W}_V \boldsymbol{x}_s \mathrm{softmax}_l(\boldsymbol{x}_s^\top \boldsymbol{W}_K^\top \boldsymbol{W}_Q \boldsymbol{x}_l + \boldsymbol{u}_{(s,l)}^\top \boldsymbol{b}^{(t)})
$$

$$
\cdot \sum_{r \in \mathcal{S}^{l,t}} \mathrm{softmax}_l(\boldsymbol{x}_r^\top \boldsymbol{W}_K^\top \boldsymbol{W}_Q \boldsymbol{x}_l + \boldsymbol{u}_{(r,l)}^\top \boldsymbol{b}^{(t)}) \boldsymbol{W}_K (\boldsymbol{x}_s - \boldsymbol{x}_r) \boldsymbol{x}_l^\top \Big) \tag{129}
$$

$$
= \eta \frac{1}{B} \sum_{n \in \mathcal{B}_b} (-y_l) \sum_{i=1}^m a_i \mathbb{1}[\boldsymbol{W}_{O_{(i,\cdot)}} \sum_{s \in \mathcal{S}^{l,t}} \boldsymbol{W}_V \boldsymbol{x}_s \mathrm{softmax}_l(\boldsymbol{x}_s^\top \boldsymbol{W}_K^\top \boldsymbol{W}_Q \boldsymbol{x}_l + \boldsymbol{u}_{(s,l)}^\top \boldsymbol{b}^{(t)}) \geq 0]
$$

$$
\cdot \Big( \boldsymbol{W}_{O_{(i,\cdot)}} \sum_{s \in \mathcal{S}^{l,t}} \boldsymbol{W}_V \boldsymbol{x}_s \mathrm{softmax}_l(\boldsymbol{x}_s^\top \boldsymbol{W}_K^\top \boldsymbol{W}_Q \boldsymbol{x}_l + \boldsymbol{u}_{(s,l)}^\top \boldsymbol{b}^{(t)})
$$

$$
\cdot (\boldsymbol{W}_K \boldsymbol{x}_s - \sum_{r \in \mathcal{S}^{l,t}} \mathrm{softmax}_l(\boldsymbol{x}_r^\top \boldsymbol{W}_K^\top \boldsymbol{W}_Q \boldsymbol{x}_l + \boldsymbol{u}_{(r,l)}^\top \boldsymbol{b}^{(t)}) \boldsymbol{W}_K \boldsymbol{x}_r) \boldsymbol{x}_l^\top \Big).
$$

(i) If $l \in \mathcal{S}_1^{n,t}$ or $l \in \mathcal{S}_2^{n,t}$, say $l \in \mathcal{S}_1^{n,t}$, we have the following derivation.

At the initial point, we can obtain

$$
\boldsymbol{W}_{O_{(i,\cdot)}}^{(0)} \sum_{s \in \mathcal{S}^{l,t}} \boldsymbol{W}_V^{(0)} \boldsymbol{x}_s \mathrm{softmax}_l((\boldsymbol{W}_K^{(0)} \boldsymbol{x}_s)^\top \boldsymbol{W}_Q^{(0)} \boldsymbol{x}_l + \boldsymbol{u}_{(s,l)}^\top \boldsymbol{b}^{(0)}) > 0, \tag{130}
$$

and

$$
\mathrm{softmax}_l((\boldsymbol{W}_K^{(0)} \boldsymbol{x}_s)^\top \boldsymbol{W}_Q^{(0)} \boldsymbol{x}_l + \boldsymbol{u}_{(s,l)}^\top \boldsymbol{b}^{(0)}) \geq \Omega(1) \cdot \sum_{r \in \mathcal{S}_2^{l,t}} \mathrm{softmax}_l((\boldsymbol{W}_K^{(0)} \boldsymbol{x}_r)^\top \boldsymbol{W}_Q^{(0)} \boldsymbol{x}_l + \boldsymbol{u}_{(s,l)}^\top \boldsymbol{b}^{(0)}), \tag{131}
$$

for $s \in \mathcal{S}_1^{l,t}$.

For $r, l \in \mathcal{S}_1^{l,t}$, if $u_{(r,l)_{z_0}} = 1$, by (35) we have

$$
\mathrm{softmax}_l(\boldsymbol{x}_r^\top \boldsymbol{W}_K^{(t)} \boldsymbol{W}_Q^{(t)} \boldsymbol{x}_l + \boldsymbol{u}_{(r,l)}^\top \boldsymbol{b}^{(t)})
$$

$$
\gtrsim \frac{e^{\|\boldsymbol{q}_1(t)\|^2 - (\delta+\tau)\|\boldsymbol{q}_1(t)\| + b_{z_0}^{(t)}}}{\sum_{z \in \mathcal{Z}} |\mathcal{N}_z^n \cap \mathcal{S}_*^{n,T}| e^{\|\boldsymbol{q}_1(T)\|^2 - (\sigma+\tau)\|\boldsymbol{q}_1(T)\| + b_z^{(T)}} + \sum_{z \in \mathcal{Z}} |(\mathcal{N}_z^n \cap \mathcal{S}^{n,T}) - \mathcal{S}_1^{n,T}| e^{b_z^{(T)}}}, \tag{132}
$$

Likewise, for $r \notin \mathcal{S}_1^{l,t}$ and $l \in \mathcal{S}_1^{l,t}$, we have

$$
\mathrm{softmax}_l(\boldsymbol{x}_r^\top \boldsymbol{W}_K^{(t+1)^\top} \boldsymbol{W}_Q^{(t+1)} \boldsymbol{x}_l + \boldsymbol{u}_{(r,l)}^\top \boldsymbol{b}^{(t)})
$$

$$
\lesssim \frac{e^{b_{z_0}^{(t)}}}{\sum_{z \in \mathcal{Z}} |\mathcal{N}_z^n \cap \mathcal{S}_*^{n,T}| e^{\|\boldsymbol{q}_1(T)\|^2 - (\sigma+\tau)\|\boldsymbol{q}_1(T)\| + b_z^{(T)}} + \sum_{z \in \mathcal{Z}} |(\mathcal{N}_z^n \cap \mathcal{S}^{n,T}) - \mathcal{S}_1^{n,T}| e^{b_z^{(T)}}}. \tag{133}
$$

Therefore, for $s, r, l \in \mathcal{S}_1^{n,t}$, let

$$
\boldsymbol{W}_K^{(t)} \boldsymbol{x}_s - \sum_{r \in \mathcal{S}^{l,t}} \mathrm{softmax}_l(\boldsymbol{x}_r^\top \boldsymbol{W}_K^{(t)^\top} \boldsymbol{W}_Q^{(t)} \boldsymbol{x}_l + \boldsymbol{u}_{(r,l)}^\top \boldsymbol{b}^{(t)}) \boldsymbol{W}_K^{(t)} \boldsymbol{x}_r := \boldsymbol{\beta}_1^l(t) \boldsymbol{q}_1(t) + \boldsymbol{\beta}_2^l(t), \tag{134}
$$

where

$$
\beta_1^l(t) \gtrsim \frac{\sum_{z \in \mathcal{Z}} (|\mathcal{N}_z^n \cap \mathcal{S}^{l,t}| - |\mathcal{N}_z^l \cap \mathcal{S}_1^{l,t}|) e^{b_z(t)}}{\sum_{z \in \mathcal{Z}} |\mathcal{N}_z^l \cap \mathcal{S}_*^{n,T}| e^{\|\boldsymbol{q}_1(T)\|^2 - (\sigma+\tau)\|\boldsymbol{q}_1(T)\| + b_z^{(T)}} + \sum_{z \in \mathcal{Z}} |(\mathcal{N}_z^n \cap \mathcal{S}^{n,T}) - \mathcal{S}_1^{n,T}| e^{b_z^{(T)}}}
$$

$$
\gtrsim \phi_l(t)(|\mathcal{S}^{l,t}| - |\mathcal{S}_1^{l,t}|), \tag{135}
$$

$$\beta_1^l(t) \lesssim e^{2(\delta+\tau)\|\boldsymbol{q}_1(t)\|}\phi_l(t)(|\mathcal{S}^{l,t}| - |\mathcal{S}_1^{l,t}|) \leq \phi_l(t)(|\mathcal{S}^{l,t}| - |\mathcal{S}_1^{l,t}|). \tag{136}$$

Meanwhile,

$$\boldsymbol{\beta}_2^l(t) \approx \Theta(1) \cdot \boldsymbol{o}_j^l(t) + Q_e(t)\boldsymbol{r}_2(t) + \sum_{n=3}^{M} \gamma_n' \boldsymbol{r}_n(t) - \sum_{a=1}^{M} \sum_{r \in \mathcal{S}^{l,t}} \mathrm{softmax}_l(\boldsymbol{x}_r^\top \boldsymbol{W}_K^{(t)\top} \boldsymbol{W}_Q^{(t)} \boldsymbol{x}_l)\boldsymbol{r}_a(t)$$

$$= \Theta(1) \cdot \boldsymbol{o}_j^l(t) + \sum_{n=1}^{M} \zeta_n' \boldsymbol{r}_n(t), \tag{137}$$

for some $Q_e(t) > 0$ and $\gamma_l' > 0$. Here

$$|\zeta_l'| \leq \beta_1^n(t)\frac{|\mathcal{S}_l^{n,t}|}{|\mathcal{S}^{n,t}| - |\mathcal{S}_1^{n,t}|}, \tag{138}$$

for $l \geq 2$. Note that $|\zeta_l'| = 0$ if $|\mathcal{S}^{n,t}| = |\mathcal{S}_1^{n,t}|$, $l \geq 2$.
For $i \in \mathcal{W}_l(0)$, by Lemma 9,

$$\boldsymbol{W}_{O_{(i,\cdot)}}^{(t)} \sum_{s \in \mathcal{S}^{l,t}} \boldsymbol{W}_V^{(t)} \boldsymbol{x}_s \mathrm{softmax}_l(\boldsymbol{x}_s^\top \boldsymbol{W}_K^{(t)\top} \boldsymbol{W}_Q^{(t)} \boldsymbol{x}_l + \boldsymbol{u}_{(s,l)}^\top \boldsymbol{b}^{(t)}) > 0. \tag{139}$$

Then we study how large the coefficient of $\boldsymbol{q}_1(t)$ in (129).
If $s \in \mathcal{S}_1^{l,t}$, from basic mathematical computation given (22) to (25),

$$\boldsymbol{W}_{O_{(i,\cdot)}}^{(t)} \boldsymbol{W}_V^{(t)} \boldsymbol{x}_s \mathrm{softmax}_l(\boldsymbol{x}_s^\top \boldsymbol{W}_K^{(t)\top} \boldsymbol{W}_Q^{(t)} \boldsymbol{x}_l + \boldsymbol{u}_{(s,l)}^\top \boldsymbol{b}^{(t)})$$

$$\gtrsim \frac{p_l(t)}{|\mathcal{S}_1^{n,t}|}\Big(\frac{1-2p}{B} \sum_{n \in \mathcal{B}_b} \frac{\xi\eta(t+1)^2 m}{a^2}(\frac{1}{4B} \sum_{n \in \mathcal{B}_b} p_n(b) - \sigma - \tau)$$

$$\cdot + \eta m \frac{1}{2B} \sum_{b=1}^{t} \sum_{n \in \mathcal{B}_b} \frac{p_n(b)}{a}(1 - (\sigma + \tau)) \cdot (\frac{\xi}{aB} \sum_{n \in \mathcal{B}_b} \frac{(1-2p)\eta(t+1)^2}{4B} \sum_{n \in \mathcal{B}_b} \frac{m}{a}p_l(t))^2\Big). \tag{140}$$

If $s \in \mathcal{S}_2^{l,t}$ and $j \in \mathcal{S}_1^{l,t}$, from (26) to (29), we have

$$\boldsymbol{W}_{O_{(i,\cdot)}}^{(t)} \boldsymbol{W}_V^{(t)} \boldsymbol{x}_s \mathrm{softmax}_l(\boldsymbol{x}_s^\top \boldsymbol{W}_K^{(t)\top} \boldsymbol{W}_Q^{(t)} \boldsymbol{x}_l + \boldsymbol{u}_{(s,l)}^\top \boldsymbol{b}^{(t)})$$

$$\lesssim \boldsymbol{W}_{O_{(i,\cdot)}}^{(t)} \boldsymbol{W}_V^{(t)} \boldsymbol{x}_j \mathrm{softmax}_l(\boldsymbol{x}_j^\top \boldsymbol{W}_K^{(t)\top} \boldsymbol{W}_Q^{(t)} \boldsymbol{x}_l + \boldsymbol{u}_{(j,l)}^\top \boldsymbol{b}^{(t)}) \cdot \phi_n(t)\frac{|\mathcal{S}_1^{n,t}|}{p_l(t)}. \tag{141}$$

If $i \in \mathcal{W}_l(0)$, $s \notin (\mathcal{S}_1^{l,t} \cup \mathcal{S}_2^{l,t})$, and $j \in \mathcal{S}_1^{l,t}$,

$$\boldsymbol{W}_{O_{(i,\cdot)}}^{(t)} \boldsymbol{W}_V^{(t)} \boldsymbol{x}_s \mathrm{softmax}_l(\boldsymbol{x}_s^\top \boldsymbol{W}_K^{(t)\top} \boldsymbol{W}_Q^{(t)} \boldsymbol{x}_l + \boldsymbol{u}_{(s,l)}^\top \boldsymbol{b}^{(t)})$$

$$\lesssim \boldsymbol{W}_{O_{(i,\cdot)}}^{(t)} \boldsymbol{W}_V^{(t)} \boldsymbol{x}_j \mathrm{softmax}_l(\boldsymbol{x}_j^\top \boldsymbol{W}_K^{(t)\top} \boldsymbol{W}_Q^{(t)} \boldsymbol{x}_l + \boldsymbol{u}_{(j,l)}^\top \boldsymbol{b}^{(t)})\phi_l(t) \cdot \frac{|\mathcal{S}_1^{l,t}|}{p_l(t)} \tag{142}$$

by (30) to (32).
Hence, for $i \in \mathcal{W}_l(0)$, $j \in \mathcal{S}_1^{g,t}$, combining (135) and (140), we can obtain

$$\boldsymbol{W}_{O_{(i,\cdot)}}^{(t)} \sum_{s \in \mathcal{S}^{l,t}} \boldsymbol{W}_V^{(t)} \boldsymbol{x}_s \mathrm{softmax}_l(\boldsymbol{x}_s^\top \boldsymbol{W}_K^{(t)\top} \boldsymbol{W}_Q^{(t)} \boldsymbol{x}_l + \boldsymbol{u}_{(s,l)}^\top \boldsymbol{b}^{(t)})\boldsymbol{q}_1(t)^\top$$

$$\cdot (\boldsymbol{W}_K^{(t)} \boldsymbol{x}_s - \sum_{r \in \mathcal{S}^{l,t}} \mathrm{softmax}_l(\boldsymbol{x}_r^\top \boldsymbol{W}_K^{(t)\top} \boldsymbol{W}_Q^{(t)} \boldsymbol{x}_l + \boldsymbol{u}_{(r,l)}^\top \boldsymbol{b}^{(t)})\boldsymbol{W}_K^{(t)} \boldsymbol{x}_r)\boldsymbol{x}_l^\top \boldsymbol{x}_j$$

$$\gtrsim \Big(\frac{1-2p}{B} \sum_{n \in \mathcal{B}_b} \frac{\xi\eta(t+1)^2 m}{a^2}(\frac{1}{4B} \sum_{n \in \mathcal{B}_b} p_n(b) - \sigma - \tau) + \eta m \frac{1}{2B} \sum_{b=1}^{t} \sum_{n \in \mathcal{B}_b} \frac{p_n(b)}{a}(1 - (\sigma + \tau))$$

$$\cdot (\frac{\xi}{aB} \sum_{n \in \mathcal{B}_b} \frac{(1-2p)\eta(t+1)^2}{4B} \sum_{n \in \mathcal{B}_b} \frac{m}{a}p_l(t))^2\Big)\phi_l(t)(|\mathcal{S}^{l,t}| - |\mathcal{S}_1^{l,t}|)\|\boldsymbol{q}_1(t)\|^2. \tag{143}$$

For $i \in \mathcal{U}_l(t)$ and $l \in \mathcal{S}_1^{l,t}$, $j \in \mathcal{S}_1^{g,t}$, and $k \in \mathcal{W}_l(0)$,

$$
\boldsymbol{W}_{O_{(i,\cdot)}}^{(t)} \sum_{s \in \mathcal{S}^{l,t}} \boldsymbol{W}_V^{(t)} \boldsymbol{x}_s \mathrm{softmax}_l({\boldsymbol{x}_s}^\top {\boldsymbol{W}_K^{(t)}}^\top \boldsymbol{W}_Q^{(t)} \boldsymbol{x}_l + \boldsymbol{u}_{(s,l)}^\top \boldsymbol{b}^{(t)}) \boldsymbol{q}_1(t)^\top
$$

$$
\cdot (\boldsymbol{W}_K^{(t)} \boldsymbol{x}_s - \sum_{r \in \mathcal{S}^{n,t}} \mathrm{softmax}_l({\boldsymbol{x}_r}^\top {\boldsymbol{W}_K^{(t)}}^\top \boldsymbol{W}_Q^{(t)} \boldsymbol{x}_l + \boldsymbol{u}_{(s,l)}^\top \boldsymbol{b}^{(t)}) \boldsymbol{W}_K^{(t)} \boldsymbol{x}_r) \boldsymbol{x}_l^\top \boldsymbol{x}_j
$$

$$
\lesssim \Big( \frac{1-2p}{B} \sum_{n \in \mathcal{B}_b} \frac{\xi \eta (t+1)^2 m}{a^2} \big( \frac{1}{4B} \sum_{n \in \mathcal{B}_b} p_n(b) - \sigma - \tau \big) + \eta m \frac{1}{2B} \sum_{b=1}^t \sum_{n \in \mathcal{B}_b} \frac{p_n(b)}{a} (1 - (\sigma + \tau))
$$

$$
\cdot \big( \frac{\xi}{aB} \sum_{n \in \mathcal{B}_b} \frac{(1-2p)\eta(t+1)^2}{4B} \sum_{n \in \mathcal{B}_b} \frac{m}{a} p_l(t) \big)^2 \Big) \phi_n(t) |\mathcal{S}_2^{n,t}| \cdot \beta_1(t) \|\boldsymbol{q}_1(t)\|^2.
$$

$$(144)$$

For $i \notin (\mathcal{W}_l(t) \cup \mathcal{U}_l(t))$ and $l \in \mathcal{S}_1^{l,t}$, $j \in \mathcal{S}_1^g$,

$$
\boldsymbol{W}_{O_{(i,\cdot)}}^{(t)} \sum_{s \in \mathcal{S}^{l,t}} \boldsymbol{W}_V^{(t)} \boldsymbol{x}_s \mathrm{softmax}_l({\boldsymbol{x}_s}^\top {\boldsymbol{W}_K}^{(t)^\top} \boldsymbol{W}_Q^{(t)} \boldsymbol{x}_l + \boldsymbol{u}_{(s,l)}^\top \boldsymbol{b}^{(t)}) \boldsymbol{q}_1(t)^\top
$$

$$
\cdot (\boldsymbol{W}_K^{(t)} \boldsymbol{x}_s - \sum_{r \in \mathcal{S}^{l,t}} \mathrm{softmax}_l({\boldsymbol{x}_r}^\top {\boldsymbol{W}_K}^{(t)^\top} \boldsymbol{W}_Q^{(t)} \boldsymbol{x}_l + \boldsymbol{u}_{(s,l)}^\top \boldsymbol{b}^{(t)}) \boldsymbol{x}_r) \boldsymbol{x}_l^\top \boldsymbol{x}_j
$$

$$
\lesssim \boldsymbol{W}_{O_{(k,\cdot)}}^{(t)} \sum_{s \in \mathcal{S}^{l,t}} \boldsymbol{W}_V^{(t)} \boldsymbol{x}_s \mathrm{softmax}_l({\boldsymbol{x}_s}^\top {\boldsymbol{W}_K}^{(t)^\top} \boldsymbol{W}_Q^{(t)} \boldsymbol{x}_l + \boldsymbol{u}_{(s,l)}^\top \boldsymbol{b}^{(t)}) \boldsymbol{q}_1(t)^\top
$$

$$
\cdot (\boldsymbol{W}_K^{(t)} \boldsymbol{x}_s - \sum_{r \in \mathcal{S}^{l,t}} \mathrm{softmax}_l({\boldsymbol{x}_r}^\top {\boldsymbol{W}_K}^{(t)^\top} \boldsymbol{W}_Q^{(t)} \boldsymbol{x}_l + \boldsymbol{u}_{(s,l)}^\top \boldsymbol{b}^{(t)}) \boldsymbol{x}_r) \boldsymbol{x}_l^\top \boldsymbol{x}_j \cdot \frac{1}{\sqrt{B}}.
$$

$$(145)$$

Therefore, by the update rule,

$$
\boldsymbol{W}_Q^{(t+1)} \boldsymbol{x}_j = \boldsymbol{W}_Q^{(t)} \boldsymbol{x}_j - \eta \frac{1}{B} \sum_{n \in \mathcal{B}_b} \Big( \frac{\partial \mathbf{Loss}(\boldsymbol{X}, y_n)}{\partial \boldsymbol{W}_Q} \Big| \boldsymbol{W}_Q^{(t)} \Big) \boldsymbol{x}_j
$$

$$
= \boldsymbol{r}_1(t) + K(t) \boldsymbol{q}_1(t) + \Theta(1) \cdot \boldsymbol{n}_j(t) + |K_e|(t) \boldsymbol{q}_2(t) + \sum_{l=3}^M \gamma_l' \boldsymbol{q}_l(t) \qquad (146)
$$

$$
= (1 + K(t)) \boldsymbol{q}_1(t) + \Theta(1) \cdot \boldsymbol{n}_j(t) + |K_e|(t) \boldsymbol{q}_2(t) + \sum_{l=3}^M \gamma_l' \boldsymbol{q}_l(t),
$$

where the last step is by

$$
\boldsymbol{q}_1(t) = k_1(t) \cdot \boldsymbol{r}_1(t), \qquad (147)
$$

and

$$
\boldsymbol{q}_2(t) = k_2(t) \cdot \boldsymbol{r}_2(t), \qquad (148)
$$

for $k_1(t) > 0$ and $k_2(t) > 0$ from induction, i.e., $\boldsymbol{q}_1(t)$ and $\boldsymbol{r}_1(t)$, $\boldsymbol{q}_1(t)$ and $\boldsymbol{r}_1(t)$ are from the same direction, respectively. Define $qc_t(\boldsymbol{x}) = \boldsymbol{x}^\top \boldsymbol{q}_1(t) / \|\boldsymbol{q}_1(t)\|$ and denote

$$
\Delta(l, i) = a_i \mathbb{1}[\boldsymbol{W}_{O_{(i,\cdot)}} \sum_{s \in \mathcal{S}^{l,t}} \boldsymbol{W}_V \boldsymbol{x}_s \mathrm{softmax}_l({\boldsymbol{x}_s}^\top \boldsymbol{W}_K^\top \boldsymbol{W}_Q \boldsymbol{x}_l + \boldsymbol{u}_{(s,l)}^\top \boldsymbol{b}) \geq 0]
$$

$$
\cdot \Big( \boldsymbol{W}_{O_{(i,\cdot)}} \sum_{s \in \mathcal{S}^{l,t}} \boldsymbol{W}_V \boldsymbol{x}_s \mathrm{softmax}_l({\boldsymbol{x}_s}^\top \boldsymbol{W}_K^\top \boldsymbol{W}_Q \boldsymbol{x}_l + \boldsymbol{u}_{(s,l)}^\top \boldsymbol{b} \qquad (149)
$$

$$
\cdot (\boldsymbol{W}_K \boldsymbol{x}_s - \sum_{r \in \mathcal{S}^{l,t}} \mathrm{softmax}_l({\boldsymbol{x}_r}^\top \boldsymbol{W}_K^\top \boldsymbol{W}_Q \boldsymbol{x}_l + \boldsymbol{u}_{(r,l)}^\top \boldsymbol{b}) \boldsymbol{W}_K \boldsymbol{x}_r) \boldsymbol{x}_l^\top \Big).
$$

We then have

$$
\begin{aligned}
&K(t) \\
&\gtrsim \eta \frac{1}{B} \Big( \Big| \sum_{l \in \mathcal{B}_b, l \in \mathcal{S}_1^{l,t}} (-y_l) \sum_{i \in \mathcal{W}_l(0)} qc_t(\Delta(l,i)) \Big| - \Big| \sum_{l \in \mathcal{B}_b, l \in \mathcal{S}_1^{l,t}} (-y_l) \sum_{i \in \mathcal{U}_{l,n}(0)} qc_t(\Delta(l,i)) \Big| \\
&\quad - \Big| \sum_{l \in \mathcal{B}_b, l \in \mathcal{S}_1^{l,t}} (-y_l) \sum_{i \notin \mathcal{W}_l(0) \cup \mathcal{U}_{l,n}(0)} qc_t(\Delta(l,i)) \Big| - \Big| \sum_{l \in \mathcal{B}_b, l \in \mathcal{S}_2^{l,t}} (-y_l) \sum_{i=1}^{m} qc_t(\Delta(l,i)) \Big| \\
&\quad - \Big| \sum_{l \in \mathcal{B}_b, l \in \mathcal{S}^{l,t} - \mathcal{S}_1^{l,t} - \mathcal{S}_2^{l,t}} (-y_l) \sum_{i=1}^{m} qc_t(\Delta(l,i)) \Big| \Big) \\
&\gtrsim \eta \frac{1}{B} \sum_{n \in \mathcal{B}_b} \frac{m}{a} \Big( \frac{1-2p}{B} \sum_{n \in \mathcal{B}_b} \frac{\xi \eta(t+1)^2 m}{a^2} \big( \frac{1}{4B} \sum_{n \in \mathcal{B}_b} p_n(b) - \sigma - \tau \big) + \eta m \frac{1}{2B} \sum_{b=1}^{t} \sum_{n \in \mathcal{B}_b} \frac{p_n(b)}{a} \\
&\quad \cdot (1 - (\sigma + \tau)) \big( \frac{\xi}{aB} \sum_{n \in \mathcal{B}_b} \frac{(1-2p)\eta(t+1)^2}{4B} \sum_{n \in \mathcal{B}_b} \frac{m}{a} p_l(t) \big)^2 \Big) \phi_l(t) (|\mathcal{S}^{l,t}| - |\mathcal{S}_1^{l,t}|) \|\boldsymbol{q}_1(t)\|^2 \\
&> 0,
\end{aligned}
$$

(150)

$$
|\gamma_l'| \lesssim \frac{1}{B} \sum_{n \in \mathcal{B}_b} K(t) \cdot \frac{|\mathcal{S}_l^{n,t}|}{|\mathcal{S}^{n,t}| - |\mathcal{S}_1^{n,t}|},
$$

(151)

$$
|K_e(t)| \lesssim \frac{1}{B} \sum_{n \in \mathcal{B}_b} \lambda \cdot K(t) \cdot \frac{|\mathcal{S}_2^{n,t}|}{|\mathcal{S}^{n,t}| - |\mathcal{S}_1^{n,t}|},
$$

(152)

as long as

$$
\begin{aligned}
&\Big( \frac{1-2p}{B} \sum_{n \in \mathcal{B}_b} \frac{\xi \eta(t+1)^2 m}{a^2} \big( \frac{1}{4B} \sum_{n \in \mathcal{B}_b} p_n(b) - \sigma - \tau \big) + \eta m \frac{1}{2B} \sum_{b=1}^{t} \sum_{n \in \mathcal{B}_b} \frac{p_n(b)}{a} (1 - (\sigma + \tau)) \\
&\quad \cdot \big( \frac{\xi}{aB} \sum_{n \in \mathcal{B}_b} \frac{(1-2p)\eta(t+1)^2}{4B} \sum_{n \in \mathcal{B}_b} \frac{m}{a} p_l(t) \big)^2 \Big) \phi_l(t) (|\mathcal{S}^{l,t}| - |\mathcal{S}_1^{l,t}|) \|\boldsymbol{q}_1(t)\|^2 \\
&\gtrsim \Big( \frac{1-2p}{B} \sum_{n \in \mathcal{B}_b} \frac{\xi \eta(t+1)^2 m}{a^2} \big( \frac{1}{4B} \sum_{n \in \mathcal{B}_b} p_n(b) - \sigma - \tau \big) + \eta m \frac{1}{2B} \sum_{b=1}^{t} \sum_{n \in \mathcal{B}_b} \frac{p_n(b)}{a} (1 - (\sigma + \tau)) \\
&\quad \cdot \big( \frac{\xi}{aB} \sum_{n \in \mathcal{B}_b} \frac{(1-2p)\eta(t+1)^2}{4B} \sum_{n \in \mathcal{B}_b} \frac{m}{a} p_l(t) \big)^2 \Big) \phi_l(t) |\mathcal{S}_2^{l,t}| \cdot \beta_1(t) \|\boldsymbol{q}_1(t)\|^2.
\end{aligned}
$$

(153)

To find the sufficient condition for (153), we compare the LHS with two terms of RHS in (153). Note that when $|\mathcal{S}^{n,t}| > |\mathcal{S}_1^{n,t}|$, by (136),

$$
\phi_n(t)(|\mathcal{S}^{n,t}| - |\mathcal{S}_1^{n,t}|) \gtrsim \beta_1^n(t),
$$

(154)

Moreover,

$$
1 \gtrsim \phi_n(t) |\mathcal{S}_2^{n,t}|.
$$

(155)

For the second term on RHS, we can derive the bound in the same way.

(ii) Then we provide a brief derivation of $\boldsymbol{W}_Q^{(t+1)}\boldsymbol{x}_j$ for $j \notin (\mathcal{S}_1^{n,t} \cup \mathcal{S}_2^{n,t})$ in the following.

To be specific, for $j \in \mathcal{S}_n/(\mathcal{S}_1^{n,t} \cup \mathcal{S}_2^{n,t})$,

$$
\left\langle \eta \frac{1}{B} \sum_{n \in \mathcal{B}_b} \frac{\partial \mathbf{Loss}(\boldsymbol{X}, y_n)}{\partial \boldsymbol{W}_Q^{(t)}} \boldsymbol{x}_j^n, \boldsymbol{q}_1(t) \right\rangle
$$

$$
\gtrsim \eta \frac{1}{B} \sum_{n \in \mathcal{B}_b} \frac{m}{a} \Big( \frac{1-2p}{B} \sum_{n \in \mathcal{B}_b} \frac{\xi \eta (t+1)^2 m}{a^2} (\frac{1}{4B} \sum_{n \in \mathcal{B}_b} p_n'(b) - \sigma - \tau) + \eta m \frac{1}{2B} \sum_{b=1}^{t} \sum_{n \in \mathcal{B}_b} \frac{p_n'(b)}{a}
$$

$$
\cdot (1 - (\sigma + \tau))(\frac{\xi}{aB} \sum_{n \in \mathcal{B}_b} \frac{(1-2p)\eta(t+1)^2}{4B} \sum_{n \in \mathcal{B}_b} \frac{m}{a} p_n'(t))^2 \Big) \phi_n(t)(|\mathcal{S}^{l,t}| - |\mathcal{S}_1^{l,t}|) \|\boldsymbol{q}_1(t)\|^2,
$$

$$(156)$$

where

$$
p_n'(t)
$$

$$
= \frac{\sum_{z \in \mathcal{Z}} |\mathcal{S}_1^{n,t} \cap \mathcal{N}_z^n| e^{\boldsymbol{q}_1(t)^\top \sum_{b=1}^t K(b)\boldsymbol{q}_1(0) - (\delta + \tau)] \|\boldsymbol{q}_1(t)\| + b_z^{(t)}}}{\sum_{z \in \mathcal{Z}} |(\mathcal{S}_1^{n,t} \cup \mathcal{S}_2^{n,t}) \cap \mathcal{N}_z^n| e^{\boldsymbol{q}_1(t)^\top \sum_{b=1}^t K(b)\boldsymbol{q}_1(b) - (\delta + \tau)] \|\boldsymbol{q}_1(t)\| + b_z^{(t)}} + |\mathcal{S}^{n,t}| - |\mathcal{S}_1^{n,t}| - |\mathcal{S}_2^{n,t}|}.
$$

$$(157)$$

When $K(b)$ is close to $0^+$, we have

$$
\prod_{b=1}^{t} \sqrt{1 + K(b)\|\boldsymbol{q}(0)\|^2} \gtrsim e^{\sum_{b=1}^{t} K(b)\|\boldsymbol{q}_1(0)\|^2} \geq \sum_{b=1}^{t} K(b)\|\boldsymbol{q}_1(0)\|^2,
$$

$$(158)$$

where the first step comes from $\log(1 + x) \approx x$ when $x \to 0^+$. Therefore, one can derive that

$$
\left\langle \eta \frac{1}{B} \sum_{n \in \mathcal{B}_b} \frac{\partial \mathbf{Loss}(\boldsymbol{X}^n, y_n)}{\partial \boldsymbol{W}_Q^{(t)}} \boldsymbol{x}_j^n, \boldsymbol{q}_1(t) \right\rangle \gtrsim \Theta(1) \cdot K(t).
$$

$$(159)$$

At the same time, the value of $p_n'(t)$ will increase to 1 along the training, making the component of $\boldsymbol{q}_1(t)$ the major part in $\eta \frac{1}{B} \sum_{n \in \mathcal{B}_b} \frac{\partial \mathbf{Loss}(\boldsymbol{X}^n, y_n)}{\partial \boldsymbol{W}_Q^{(t)}} \boldsymbol{x}_j^n$. This is also the same for $\boldsymbol{q}_2(t)$.

Hence, if $j \in \mathcal{S}_l^{n,t}$ for $l \geq 3$,

$$
\boldsymbol{W}_Q^{(t+1)}\boldsymbol{x}_j = \boldsymbol{q}_l(t) + \Theta(1) \cdot \boldsymbol{n}_j(t) + \Theta(1) \cdot K(t)(\boldsymbol{q}_1(t) + \boldsymbol{q}_2(t)) + \sum_{l=2}^{M} \gamma_l' \boldsymbol{q}_l(t).
$$

$$(160)$$

Similarly, for $j \in \mathcal{S}_2^{n,t}$,

$$
\boldsymbol{W}_Q^{(t+1)}\boldsymbol{x}_j = (1 + K(t)\frac{|\mathcal{S}_2^{n,t}|}{|\mathcal{S}_1^{n,t}|})\boldsymbol{q}_2(t) + \Theta(1) \cdot \boldsymbol{n}_j(t) + \Theta(1) \cdot K(t)\boldsymbol{q}_1(t) + \sum_{l=2}^{M} \gamma_l' \boldsymbol{q}_l(t).
$$

$$(161)$$

(b) For the gradient of $\boldsymbol{W}_K$, we have

$$
\frac{1}{B} \sum_{n \in \mathcal{B}_b} \frac{\partial \mathbf{Loss}(\boldsymbol{x}_n, y_n)}{\partial F(\boldsymbol{x}_n)} \frac{\partial F(\boldsymbol{x}_n)}{\partial \boldsymbol{W}_K}
$$

$$
= \frac{1}{B} \sum_{n \in \mathcal{B}_b} (-y_n) \sum_{i=1}^{m} a_i \mathbb{1}[\boldsymbol{W}_{O_{(i,\cdot)}} \sum_{s \in \mathcal{S}^{l,t}} \boldsymbol{W}_V \boldsymbol{x}_s \mathrm{softmax}_l(\boldsymbol{x}_s^\top \boldsymbol{W}_K^\top \boldsymbol{W}_Q \boldsymbol{x}_l + \boldsymbol{u}_{(s,l)}^\top \boldsymbol{b}^{(t)}) \geq 0]
$$

$$(162)$$

$$
\cdot \Big( \boldsymbol{W}_{O_{(i,\cdot)}} \sum_{s \in \mathcal{S}^{l,t}} \boldsymbol{W}_V \boldsymbol{x}_s \mathrm{softmax}_l(\boldsymbol{x}_s^\top \boldsymbol{W}_K^\top \boldsymbol{W}_Q \boldsymbol{x}_l + \boldsymbol{u}_{(s,l)}^\top \boldsymbol{b}^{(t)}) \boldsymbol{W}_Q^\top \boldsymbol{x}_l
$$

$$
\cdot (\boldsymbol{x}_s - \sum_{r \in \mathcal{S}^{l,t}} \mathrm{softmax}_l(\boldsymbol{x}_r^\top \boldsymbol{W}_K^\top \boldsymbol{W}_Q \boldsymbol{x}_l + \boldsymbol{u}_{(s,l)}^\top \boldsymbol{b}^{(t)}) \boldsymbol{x}_r)^\top \Big).
$$

Hence, for $j \in \mathcal{S}_1^{n,t}$, we can follow (146) to derive

$$
\boldsymbol{W}_K^{(t+1)}\boldsymbol{x}_j \approx (1 + Q(t))\boldsymbol{q}_1(t) + \Theta(1) \cdot \boldsymbol{o}_j(t) + |Q_e(t)|\boldsymbol{r}_2(t) + \sum_{l=3}^{M} \gamma_l' \boldsymbol{r}_l(t),
$$

$$(163)$$

where

$$Q(t) \geq K(t)(1 - \lambda) > 0, \tag{164}$$

for $\lambda < 1$, and

$$|\gamma_l| \lesssim \frac{1}{B} \sum_{n \in \mathcal{B}_b} Q(t) \cdot \frac{|\mathcal{S}_l^{n,t}|}{|\mathcal{S}^{n,t}| - |\mathcal{S}_*^{n,t}|}, \tag{165}$$

$$|Q_e(t)| \lesssim \frac{1}{B} \sum_{n \in \mathcal{B}_b} Q(t) \cdot \frac{|\mathcal{S}_\#^{n,t}|}{|\mathcal{S}^{n,t}| - |\mathcal{S}_*^{n,t}|}. \tag{166}$$

Similarly, for $j \in \mathcal{S}_2^{n,t}$, we can obtain

$$\boldsymbol{W}_K^{(t+1)} \boldsymbol{x}_j \approx (1 + Q(t)) \boldsymbol{q}_2(t) + \Theta(1) \cdot \boldsymbol{o}_j(t) + |Q_e(t)| \boldsymbol{r}_1(t) + \sum_{l=3}^{M} \gamma_l' \boldsymbol{r}_l(t), \tag{167}$$

For $j \in \mathcal{S}_l^{n,t}, \ l = 3, 4, \cdots, M$, we can obtain

$$\boldsymbol{W}_K^{(t+1)} \boldsymbol{x}_j \approx \boldsymbol{q}_l(t) + \Theta(1) \cdot \boldsymbol{o}_j(t) + \Theta(1) \cdot |Q_f(t)| \boldsymbol{r}_1(t) + \Theta(1) \cdot Q_f(t) \boldsymbol{r}_2(t) + \sum_{i=3}^{M} \gamma_i' \boldsymbol{r}_i(t), \tag{168}$$

where

$$|Q_f(t)| \lesssim Q(t). \tag{169}$$

Therefore, for $l \in \mathcal{S}_1^{n,t}$, if $j \in \mathcal{S}_1^{n,t}$,

$$\begin{aligned}
&\boldsymbol{x}_j^\top {\boldsymbol{W}_K^{(t+1)}}^\top \boldsymbol{W}_Q^{(t+1)} \boldsymbol{x}_l \\
&\gtrsim (1 + K(t))(1 + Q(t)) \|\boldsymbol{q}_1(t)\|^2 - (\delta + \tau) \|\boldsymbol{q}_1(t)\| + K_e(t) Q_e(t) \|\boldsymbol{q}_2(t)\| \|\boldsymbol{r}_2(t)\| \\
&\quad + \sum_{l=3}^{M} \gamma_l \gamma_l' \|\boldsymbol{q}_l(t)\| \|\boldsymbol{r}_l(t)\| \\
&\gtrsim (1 + K(t))(1 + Q(t)) \|\boldsymbol{q}_1(t)\|^2 - (\delta + \tau) \|\boldsymbol{q}_1(t)\| \\
&\quad - \sqrt{\sum_{l=2}^{M} \left( \frac{1}{B} \sum_{n \in \mathcal{B}_b} Q(t) \frac{|\mathcal{S}_l^{n,t}|}{|\mathcal{S}^{n,t}| - |\mathcal{S}_*^{n,t}|} \right)^2 \|\boldsymbol{r}_l(t)\|^2} \cdot \sqrt{\sum_{l=2}^{M} \left( \frac{1}{B} \sum_{n \in \mathcal{B}_b} K(t) \frac{|\mathcal{S}_l^{n,t}|}{|\mathcal{S}^{n,t}| - |\mathcal{S}_*^{n,t}|} \right)^2 \|\boldsymbol{q}_l(t)\|^2} \\
&\gtrsim (1 + K(t) + Q(t)) \|\boldsymbol{q}_1(t)\|^2 - (\delta + \tau) \|\boldsymbol{q}_1(t)\|,
\end{aligned} \tag{170}$$

where the second step is from Cauchy-Schwarz inequality.
If $j \notin \mathcal{S}_1^{n,t}$,

$$\begin{aligned}
&\boldsymbol{x}_j^\top {\boldsymbol{W}_K^{(t+1)}}^\top \boldsymbol{W}_Q^{(t+1)} \boldsymbol{x}_l \\
&\lesssim (1 + K(t)) Q_f(t) \|\boldsymbol{q}_1(t)\|^2 + K_e(t) Q_f(t) \|\boldsymbol{q}_2(t)\|^2 + \gamma_l \|\boldsymbol{q}_l(t)\|^2 + (\delta + \tau) \|\boldsymbol{q}_1(t)\| \\
&\lesssim Q_f(t) \|\boldsymbol{q}_1(t)\|^2 + (\delta + \tau) \|\boldsymbol{q}_1(t)\|.
\end{aligned} \tag{171}$$

Therefore, for $r, l \in \mathcal{S}_1^{l,t}$, if $u_{(r,l)_{z_0}} = 1$, we have

$$\begin{aligned}
&\text{softmax}_l (\boldsymbol{x}_r^\top \boldsymbol{W}_K^{(t+1)} \boldsymbol{W}_Q^{(t+1)} \boldsymbol{x}_l + \boldsymbol{u}_{(r,l)}^\top \boldsymbol{b}^{(t+1)}) \\
&\gtrsim \frac{e^{(1+K(t)) \|\boldsymbol{q}_1(t)\|^2 - (\delta + \tau) \|\boldsymbol{q}_1(t)\| + b_{z_0}^{(t)}}}{\sum_{z \in \mathcal{Z}} |\mathcal{N}_z^n \cap \mathcal{S}_*^{n,T}| e^{(1+K(t)) \|\boldsymbol{q}_1(T)\|^2 - (\sigma + \tau) \|\boldsymbol{q}_1(T)\| + b_z^{(T)}} + \sum_{z \in \mathcal{Z}} |(\mathcal{N}_z^n \cap \mathcal{S}^{n,T}) - \mathcal{S}_1^{n,T}| e^{b_z^{(T)}}}.
\end{aligned} \tag{172}$$

Similarly, for $r \notin \mathcal{S}_1^{l,t}$ and $l \in \mathcal{S}_1^{l,t}$, we have

$$\begin{aligned}
&\text{softmax}_l (\boldsymbol{x}_r^\top {\boldsymbol{W}_K^{(t+1)}}^\top \boldsymbol{W}_Q^{(t+1)} \boldsymbol{x}_l + \boldsymbol{u}_{(r,l)}^\top \boldsymbol{b}^{(t)}) \\
&\lesssim \frac{e^{b_{z_0}^{(t)}}}{\sum_{z \in \mathcal{Z}} |\mathcal{N}_z^n \cap \mathcal{S}_*^{n,T}| e^{(1+K(t)) \|\boldsymbol{q}_1(T)\|^2 - (\sigma + \tau) \|\boldsymbol{q}_1(T)\| + b_z^{(T)}} + \sum_{z \in \mathcal{Z}} |(\mathcal{N}_z^n \cap \mathcal{S}^{n,T}) - \mathcal{S}_1^{n,T}| e^{b_z^{(T)}}}.
\end{aligned} \tag{173}$$

The same conclusion holds if $l \notin (\mathcal{S}_1^{n,t} \cup \mathcal{S}_2^{n,t})$.

Hence

$$\boldsymbol{q}_1(t+1) = \sqrt{(1+K(t))}\boldsymbol{q}_1(t). \tag{174}$$

$$\boldsymbol{q}_2(t+1) = \sqrt{(1+K(t))}\boldsymbol{q}_2(t). \tag{175}$$

$$\boldsymbol{r}_1(t+1) = \sqrt{(1+Q(t))}\boldsymbol{r}_1(t). \tag{176}$$

$$\boldsymbol{r}_2(t+1) = \sqrt{(1+Q(t))}\boldsymbol{r}_2(t). \tag{177}$$

It can also be verified that this Lemma holds when $t = 1$.

**Proof of Lemma 6:**

$$
\begin{aligned}
&\eta\frac{1}{B}\sum_{n\in\mathcal{B}_b}\frac{\partial\mathbf{Loss}(\boldsymbol{x}_n,y_n)}{\partial\boldsymbol{b}}\\
=&\eta\frac{1}{B}\sum_{n\in\mathcal{B}_b}\frac{\partial\mathbf{Loss}(\boldsymbol{x}_n,y_n)}{\partial F(\boldsymbol{x}_n)}\frac{\partial F(\boldsymbol{x}_n)}{\partial\boldsymbol{b}}\\
=&\eta\frac{1}{B}\sum_{l\in\mathcal{B}_b}(-y_l)\sum_{i=1}^{m}a_i\mathbb{1}[\boldsymbol{W}_{O_{(i,\cdot)}}\sum_{s\in\mathcal{S}^{l,t}}\boldsymbol{W}_V\boldsymbol{x}_l\mathrm{softmax}_l(\boldsymbol{x}_s{}^\top\boldsymbol{W}_K^\top\boldsymbol{W}_Q\boldsymbol{x}_l+\boldsymbol{u}_{(s,l)}^\top\boldsymbol{b}^{(t)})\geq 0]\cdot\Big(\boldsymbol{W}_{O_{(i,\cdot)}}\\
&\cdot\sum_{s\in\mathcal{S}_l}\boldsymbol{W}_V\boldsymbol{x}_s\mathrm{softmax}_l(\boldsymbol{x}_s{}^\top\boldsymbol{W}_K^\top\boldsymbol{W}_Q\boldsymbol{x}_l+\boldsymbol{u}_{(s,l)}^\top\boldsymbol{b}^{(t)})\sum_{r\in\mathcal{S}^{n,t}}\mathrm{softmax}_l(\boldsymbol{x}_r{}^\top\boldsymbol{W}_K^\top\boldsymbol{W}_Q\boldsymbol{x}_l+\boldsymbol{u}_{(s,l)}^\top\boldsymbol{b}^{(t)})\\
&\cdot(\boldsymbol{u}_{(s,l)}-\boldsymbol{u}_{(r,l)})\Big)\\
=&\eta\frac{1}{B}\sum_{l\in\mathcal{B}_b}(-y_l)\sum_{i=1}^{m}a_i\mathbb{1}[\boldsymbol{W}_{O_{(i,\cdot)}}\sum_{s\in\mathcal{S}^{l,t}}\boldsymbol{W}_V\boldsymbol{x}_s\mathrm{softmax}_l(\boldsymbol{x}_s{}^\top\boldsymbol{W}_K^\top\boldsymbol{W}_Q\boldsymbol{x}_l+\boldsymbol{u}_{(s,l)}^\top\boldsymbol{b}^{(t)})\geq 0]\cdot\Big(\boldsymbol{W}_{O_{(i,\cdot)}}\\
&\cdot\sum_{s\in\mathcal{S}_l}\boldsymbol{W}_V\boldsymbol{x}_s\mathrm{softmax}_l(\boldsymbol{x}_s{}^\top\boldsymbol{W}_K^\top\boldsymbol{W}_Q\boldsymbol{x}_l+\boldsymbol{u}_{(s,l)}^\top\boldsymbol{b}^{(t)})(\boldsymbol{u}_{(s,l)}-\sum_{r\in\mathcal{S}^{n,t}}\mathrm{softmax}_l(\boldsymbol{x}_r{}^\top\boldsymbol{W}_K^\top\boldsymbol{W}_Q\boldsymbol{x}_l\\
&+\boldsymbol{u}_{(s,l)}^\top\boldsymbol{b}^{(t)})\boldsymbol{u}_{(r,l)}\Big).
\end{aligned}
\tag{178}
$$

Therefore, we can derive

$$
\begin{aligned}
&\boldsymbol{W}_{O_{(i,\cdot)}} \cdot \sum_{s \in \mathcal{S}^{l,t}} \boldsymbol{W}_V \boldsymbol{x}_s \mathrm{softmax}_l(\boldsymbol{x_s}^\top \boldsymbol{W}_K^\top \boldsymbol{W}_Q \boldsymbol{x}_l + \boldsymbol{u}_{(s,l)}^\top \boldsymbol{b}^{(t)})(u_{(s,l)_z} \\
&- \sum_{r \in \mathcal{S}^{l,t}} \mathrm{softmax}_l(\boldsymbol{x_r}^\top \boldsymbol{W}_K^\top \boldsymbol{W}_Q \boldsymbol{x}_l + \boldsymbol{u}_{(r,l)}^\top \boldsymbol{b}^{(t)})u_{(r,l)_z}) \\
=&\boldsymbol{W}_{O_{(i,\cdot)}} \sum_{s \in \mathcal{S}^{l,t} \cap \mathcal{N}_l^z} \boldsymbol{W}_V \boldsymbol{x}_s \mathrm{softmax}_l(\boldsymbol{x_s}^\top \boldsymbol{W}_K^\top \boldsymbol{W}_Q \boldsymbol{x}_l + \boldsymbol{u}_{(s,l)}^\top \boldsymbol{b}^{(t)})(1 \\
&- \sum_{r \in \mathcal{S}^{l,t} \cap \mathcal{N}_l^z} \mathrm{softmax}_l(\boldsymbol{x_r}^\top \boldsymbol{W}_K^\top \boldsymbol{W}_Q \boldsymbol{x}_l + \boldsymbol{u}_{(r,l)}^\top \boldsymbol{b}^{(t)})) + \boldsymbol{W}_{O_{(i,\cdot)}} \sum_{s \in \mathcal{S}^{l,t} - \mathcal{N}_l^z} \boldsymbol{W}_V \boldsymbol{x}_s \\
&\cdot \mathrm{softmax}_l(\boldsymbol{x_s}^\top \boldsymbol{W}_K^\top \boldsymbol{W}_Q \boldsymbol{x}_l + \boldsymbol{u}_{(s,l)}^\top \boldsymbol{b}^{(t)})(- \sum_{r \in \mathcal{S}^{l,t} \cap \mathcal{N}_l^z} \mathrm{softmax}_l(\boldsymbol{x_r}^\top \boldsymbol{W}_K^\top \boldsymbol{W}_Q \boldsymbol{x}_l + \boldsymbol{u}_{(r,l)}^\top \boldsymbol{b}^{(t)})) \\
=&\boldsymbol{W}_{O_{(i,\cdot)}} \sum_{s \in \mathcal{S}^{l,t} \cap \mathcal{N}_l^z} \boldsymbol{W}_V \boldsymbol{x}_s \mathrm{softmax}_l(\boldsymbol{x_s}^\top \boldsymbol{W}_K^\top \boldsymbol{W}_Q \boldsymbol{x}_l + \boldsymbol{u}_{(s,l)}^\top \boldsymbol{b}^{(t)}) \sum_{r \in \mathcal{S}^{l,t} - \mathcal{N}_l^z} \\
&\cdot \mathrm{softmax}_l(\boldsymbol{x_r}^\top \boldsymbol{W}_K^\top \boldsymbol{W}_Q \boldsymbol{x}_l + \boldsymbol{u}_{(r,l)}^\top \boldsymbol{b}^{(t)}) - \boldsymbol{W}_{O_{(i,\cdot)}} \sum_{s \in \mathcal{S}^{l,t} - \mathcal{N}_l^z} \boldsymbol{W}_V \boldsymbol{x}_s \\
&\cdot \mathrm{softmax}_l(\boldsymbol{x_s}^\top \boldsymbol{W}_K^\top \boldsymbol{W}_Q \boldsymbol{x}_l + \boldsymbol{u}_{(s,l)}^\top \boldsymbol{b}^{(t)}) \sum_{r \in \mathcal{S}^{l,t} \cap \mathcal{N}_l^z} \mathrm{softmax}_l(\boldsymbol{x_r}^\top \boldsymbol{W}_K^\top \boldsymbol{W}_Q \boldsymbol{x}_l + \boldsymbol{u}_{(r,l)}^\top \boldsymbol{b}^{(t)}) \\
=&P_1 + P_2 + P_3.
\end{aligned}
\tag{179}
$$

where the second step is by

$$
(\sum_{s \in \mathcal{S}^{l,t} \cap \mathcal{N}_l^{z-1}} + \sum_{s \in \mathcal{S}^{l,t} - \mathcal{N}_l^{z-1}}) \mathrm{softmax}_l(\boldsymbol{x_s}^\top \boldsymbol{W}_K^\top \boldsymbol{W}_Q \boldsymbol{x}_l + \boldsymbol{u}_{(s,l)}^\top \boldsymbol{b}) = 1,
\tag{180}
$$

Define

$$
\begin{aligned}
P_1 =& \sum_{s \in \mathcal{S}^{l,t} \cap \mathcal{N}_l^z \cap \mathcal{S}_*^{l,t}} \boldsymbol{W}_{O_{(i,\cdot)}} \boldsymbol{W}_V \boldsymbol{x}_s \mathrm{softmax}_l(\boldsymbol{x}_s \boldsymbol{W}_K^\top \boldsymbol{W}_Q \boldsymbol{x}_l + \boldsymbol{u}_{(s,l)}^\top \boldsymbol{b}^{(t)}) \\
&\cdot \sum_{r \in (\mathcal{S}^{l,t} - \mathcal{N}_l^z) \cap \mathcal{S}_*^{l,t}} \mathrm{softmax}_l(\boldsymbol{x_r}^\top \boldsymbol{W}_K^\top \boldsymbol{W}_Q \boldsymbol{x}_l + \boldsymbol{u}_{(r,l)}^\top \boldsymbol{b}^{(t)}) - \sum_{s \in (\mathcal{S}^{l,t} - \mathcal{N}_l^z) \cap \mathcal{S}_*^{l,t}} \boldsymbol{W}_{O_{(i,\cdot)}} \boldsymbol{W}_V \boldsymbol{x}_s \\
&\cdot \mathrm{softmax}_l(\boldsymbol{x}_s \boldsymbol{W}_K^\top \boldsymbol{W}_Q \boldsymbol{x}_l + \boldsymbol{u}_{(s,l)}^\top \boldsymbol{b}^{(t)}) \sum_{r \in \mathcal{S}^{l,t} \cap \mathcal{N}_l^z \cap \mathcal{S}_*^{l,t}} \mathrm{softmax}_l(\boldsymbol{x_r}^\top \boldsymbol{W}_K^\top \boldsymbol{W}_Q \boldsymbol{x}_l + \boldsymbol{u}_{(r,l)}^\top \boldsymbol{b}^{(t)}),
\end{aligned}
\tag{181}
$$

$$
\begin{aligned}
P_2 =& \sum_{s \in \mathcal{S}^{l,t} \cap \mathcal{N}_l^z - \mathcal{S}_*^{l,t}} \boldsymbol{W}_{O_{(i,\cdot)}} \boldsymbol{W}_V \boldsymbol{x}_s \mathrm{softmax}_l(\boldsymbol{x}_s \boldsymbol{W}_K^\top \boldsymbol{W}_Q \boldsymbol{x}_l + \boldsymbol{u}_{(s,l)}^\top \boldsymbol{b}) \sum_{r \in \mathcal{S}^{l,t} - \mathcal{N}_z^l} \mathrm{softmax}_l(\boldsymbol{x_r}^\top \boldsymbol{W}_K^\top \boldsymbol{W}_Q \boldsymbol{x}_l \\
&\cdot + \boldsymbol{u}_{(r,l)}^\top \boldsymbol{b}^{(t)}) - \sum_{s \in (\mathcal{S}^{l,t} - \mathcal{N}_z^l) - \mathcal{S}_*^{l,t}} \boldsymbol{W}_{O_{(i,\cdot)}} \boldsymbol{W}_V \boldsymbol{x}_s \mathrm{softmax}_l(\boldsymbol{x}_s \boldsymbol{W}_K^\top \boldsymbol{W}_Q \boldsymbol{x}_l + \boldsymbol{u}_{(s,l)}^\top \boldsymbol{b}) \sum_{r \in \mathcal{S}^{l,t} \cap \mathcal{N}_l^z} \\
&\cdot \mathrm{softmax}_l(\boldsymbol{x_r}^\top \boldsymbol{W}_K^\top \boldsymbol{W}_Q \boldsymbol{x}_l + \boldsymbol{u}_{(r,l)}^\top \boldsymbol{b}^{(t)}),
\end{aligned}
\tag{182}
$$

$$
\begin{aligned}
P_3 =& \sum_{s \in \mathcal{S}^{l,t} \cap \mathcal{N}_l^z \cap \mathcal{S}_*^{l,t}} \boldsymbol{W}_{O_{(i,\cdot)}} \boldsymbol{W}_V \boldsymbol{x}_s \mathrm{softmax}_l(\boldsymbol{x}_s \boldsymbol{W}_K^\top \boldsymbol{W}_Q \boldsymbol{x}_l + \boldsymbol{u}_{(s,l)}^\top \boldsymbol{b}^{(t)}) \sum_{r \in (\mathcal{S}^{l,t} - \mathcal{N}_z^l) - \mathcal{S}_*^{l,t}} \\
&\cdot \mathrm{softmax}_l(\boldsymbol{x_r}^\top \boldsymbol{W}_K^\top \boldsymbol{W}_Q \boldsymbol{x}_l + \boldsymbol{u}_{(r,l)}^\top \boldsymbol{b}^{(t)}) - \sum_{s \in (\mathcal{S}^{l,t} - \mathcal{N}_z^l) \cap \mathcal{S}_*^{l,t}} \boldsymbol{W}_{O_{(i,\cdot)}} \boldsymbol{W}_V \boldsymbol{x}_s \\
&\cdot \mathrm{softmax}_l(\boldsymbol{x}_s \boldsymbol{W}_K^\top \boldsymbol{W}_Q \boldsymbol{x} + \boldsymbol{u}_{(s,l)}^\top \boldsymbol{b}^{(t)}) \sum_{r \in \mathcal{S}^{l,t} \cap \mathcal{N}_l^z - \mathcal{S}_*^{l,t}} \mathrm{softmax}_l(\boldsymbol{x_r}^\top \boldsymbol{W}_K^\top \boldsymbol{W}_Q \boldsymbol{x}_l + \boldsymbol{u}_{(r,l)}^\top \boldsymbol{b}^{(t)}).
\end{aligned}
\tag{183}
$$

Note that $((\mathcal{S}^{l,t} - \mathcal{N}_l^{z-1}) \cap \mathcal{S}_*^{l,t}) + (\mathcal{S}^{l,t} \cap \mathcal{N}_l^{z-1} \cap \mathcal{S}_*^{l,t}) + (\mathcal{S}^{l,t} - \mathcal{S}_*^{l,t}) = \mathcal{S}^{l,t}$. For $s, j \in \mathcal{S}_*^{l,t}$, by (223) and (224), we have

$$\|\boldsymbol{W}_V^{(t)}\boldsymbol{x}_s - \boldsymbol{W}_V^{(t)}\boldsymbol{x}_j^n\| \leq O(\|z_j(t)\|) \leq \tau. \tag{184}$$

Combining (25), we can obtain

$$|P_1| \leq (\sigma + \tau)\|\boldsymbol{W}_{O_{(i,\cdot)}}^{(t)}\| \frac{|\mathcal{S}_1^{n,t} \cap \mathcal{N}_z^l|}{|\mathcal{S}^{l,t}|} \left( \frac{|\mathcal{S}_1^{n,t}|}{|\mathcal{S}^{l,t}|} - \frac{|\mathcal{S}_1^{n,t} \cap \mathcal{N}_z^l|}{|\mathcal{S}^{l,t}|} \right). \tag{185}$$

Let

$$
\begin{aligned}
T_1 =& \sum_{s \in \mathcal{S}^{l,t} \cap \mathcal{N}_l^z - \mathcal{S}_*^{l,t} - \mathcal{S}_\#^{l,t}} \boldsymbol{W}_{O_{(i,\cdot)}} \boldsymbol{W}_V \boldsymbol{x}_s \text{softmax}_l(\boldsymbol{x}_s \boldsymbol{W}_K^\top \boldsymbol{W}_Q \boldsymbol{x}_l) \\
& \cdot \sum_{r \in \mathcal{S}^{l,t} - \mathcal{N}_z^l} \text{softmax}_l(\boldsymbol{x}_r^\top \boldsymbol{W}_K^\top \boldsymbol{W}_Q \boldsymbol{x}_l + \boldsymbol{u}_{(r,l)}^\top \boldsymbol{b}^{(t)}) - \sum_{s \in (\mathcal{S}^{l,t} - \mathcal{N}_z^l) - \mathcal{S}_*^{l,t} - \mathcal{S}_\#^{l,t}} \boldsymbol{W}_{O_{(i,\cdot)}} \\
& \cdot \boldsymbol{W}_V \boldsymbol{x}_s \text{softmax}_l(\boldsymbol{x}_s \boldsymbol{W}_K^\top \boldsymbol{W}_Q \boldsymbol{x}_l) \sum_{r \in \mathcal{S}^{l,t} \cap \mathcal{N}_l^z} \text{softmax}_l(\boldsymbol{x}_r^\top \boldsymbol{W}_K^\top \boldsymbol{W}_Q \boldsymbol{x}_l + \boldsymbol{u}_{(r,l)}^\top \boldsymbol{b}^{(t)}),
\end{aligned}
\tag{186}
$$

$$
\begin{aligned}
T_2 =& \sum_{s \in \mathcal{S}^{l,t} \cap \mathcal{N}_l^z \cap \mathcal{S}_\#^{l,t}} \boldsymbol{W}_{O_{(i,\cdot)}} \boldsymbol{W}_V \boldsymbol{x}_s \text{softmax}_l(\boldsymbol{x}_s \boldsymbol{W}_K^\top \boldsymbol{W}_Q \boldsymbol{x}_l + \boldsymbol{u}_{(s,l)}^\top \boldsymbol{b}^{(t)}) \\
& \cdot \sum_{r \in (\mathcal{S}^{l,t} - \mathcal{N}_z^l) \cap \mathcal{S}_\#^{l,t}} \text{softmax}_l(\boldsymbol{x}_r^\top \boldsymbol{W}_K^\top \boldsymbol{W}_Q \boldsymbol{x}_l + \boldsymbol{u}_{(r,l)}^\top \boldsymbol{b}^{(t)}) - \sum_{s \in (\mathcal{S}^{l,t} - \mathcal{N}_z^l) \cap \mathcal{S}_\#^{l,t}} \boldsymbol{W}_{O_{(i,\cdot)}} \boldsymbol{W}_V \boldsymbol{x}_s \\
& \cdot \text{softmax}_l(\boldsymbol{x}_s \boldsymbol{W}_K^\top \boldsymbol{W}_Q \boldsymbol{x}_l + \boldsymbol{u}_{(s,l)}^\top \boldsymbol{b}^{(t)}) \sum_{r \in \mathcal{S}^{l,t} \cap \mathcal{N}_l^z \cap \mathcal{S}_\#^{l,t}} \text{softmax}_l(\boldsymbol{x}_r^\top \boldsymbol{W}_K^\top \boldsymbol{W}_Q \boldsymbol{x}_l + \boldsymbol{u}_{(r,l)}^\top \boldsymbol{b}^{(t)}),
\end{aligned}
\tag{187}
$$

$$
\begin{aligned}
T_3 =& \sum_{s \in \mathcal{S}^{l,t} \cap \mathcal{N}_l^z \cap \mathcal{S}_\#^{l,t}} \boldsymbol{W}_{O_{(i,\cdot)}} \boldsymbol{W}_V \boldsymbol{x}_s \text{softmax}_l(\boldsymbol{x}_s \boldsymbol{W}_K^\top \boldsymbol{W}_Q \boldsymbol{x}_l + \boldsymbol{u}_{(s,l)}^\top \boldsymbol{b}^{(t)}) \sum_{r \in \mathcal{S}^{l,t} - \mathcal{N}_z^l - \mathcal{S}_\#^{l,t}} \\
& \cdot \text{softmax}_l(\boldsymbol{x}_r^\top \boldsymbol{W}_K^\top \boldsymbol{W}_Q \boldsymbol{x}_l + \boldsymbol{u}_{(r,l)}^\top \boldsymbol{b}^{(t)}) - \sum_{s \in (\mathcal{S}^{l,t} - \mathcal{N}_z^l) \cap \mathcal{S}_\#^{l,t}} \boldsymbol{W}_{O_{(i,\cdot)}} \boldsymbol{W}_V \boldsymbol{x}_s \\
& \cdot \text{softmax}_l(\boldsymbol{x}_s \boldsymbol{W}_K^\top \boldsymbol{W}_Q \boldsymbol{x}_l + \boldsymbol{u}_{(s,l)}^\top \boldsymbol{b}^{(t)}) \sum_{r \in \mathcal{S}^{l,t} \cap \mathcal{N}_l^z - \mathcal{S}_\#^{l,t}} \text{softmax}_l(\boldsymbol{x}_r^\top \boldsymbol{W}_K^\top \boldsymbol{W}_Q \boldsymbol{x}_l + \boldsymbol{u}_{(r,l)}^\top \boldsymbol{b}^{(t)}).
\end{aligned}
\tag{188}
$$

Therefore,

$$P_2 = T_1 + T_2 + T_3, \tag{189}$$

$$T_1 \leq (\sigma + \tau)\|\boldsymbol{W}_{O_{(i,\cdot)}}^{(t)}\| \cdot \frac{|\mathcal{N}_z^l - \mathcal{S}_*^{l,t} - \mathcal{S}_\#^{l,t}|}{|\mathcal{S}^{l,t}|} \frac{|\mathcal{S}^{l,t} - \mathcal{N}_z^l|}{|\mathcal{S}^{l,t}|}, \tag{190}$$

$$T_2 \leq (\sigma + \tau)\|\boldsymbol{W}_{O_{(i,\cdot)}}^{(t)}\| \cdot \frac{|\mathcal{S}_\#^{l,t} \cap \mathcal{N}_z^l|}{|\mathcal{S}^{l,t}|} \left( \frac{|\mathcal{S}_\#^{l,t}|}{|\mathcal{S}^{l,t}|} - \frac{|\mathcal{S}_\#^{l,t} \cap \mathcal{N}_z^l|}{|\mathcal{S}^{l,t}|} \right). \tag{191}$$

For $y_n = 1$, $s \in \mathcal{S}_*^{l,t}$ and $j \in \mathcal{S}_\#^{l,t}$, by (223), (224), and (225), we have

$$
\begin{aligned}
&\|\boldsymbol{W}_{O_{(i,\cdot)}}^{(t)} (\boldsymbol{W}_V^{(t)} \boldsymbol{x}_s - \boldsymbol{W}_V^{(t)} \boldsymbol{x}_j)\| \\
=&\|\boldsymbol{W}_{O_{(i,\cdot)}}^{(t)} (\boldsymbol{p}_1 - \boldsymbol{p}_2 + \boldsymbol{z}(t) - (\eta \sum_{b=1}^{t} \sum_{i \in \mathcal{W}_m(b)} V_i(b) \boldsymbol{W}_{O_{(i,\cdot)}}^{(b)}{}^\top - \eta \sum_{b=1}^{t} \sum_{i \in \mathcal{U}(b)} V_i(b) \boldsymbol{W}_{O_{(i,\cdot)}}^{(b)}{}^\top) \\
&- (\eta \sum_{b=1}^{t} \sum_{i \notin \mathcal{W}_n(b)} \lambda V_i(b) \boldsymbol{W}_{O_{(i,\cdot)}}^{(b)}{}^\top - \eta \sum_{b=1}^{t} \sum_{i \notin \mathcal{U}(b)} \lambda V_i(b) \boldsymbol{W}_{O_{(i,\cdot)}}^{(b)}{}^\top))\| \\
\gtrsim& \Big(\frac{1-2p}{B} \sum_{n \in \mathcal{B}_b} \frac{\xi \eta (t+1)^2 m}{a^2} (\frac{1}{4B} \sum_{n \in \mathcal{B}_b} p_n(b) - \sigma - \tau) \\
&+ \eta m \frac{1}{2B} \sum_{b=1}^{t} \sum_{n \in \mathcal{B}_b} \frac{p_n(b)}{a} (1 - (\sigma + \tau)) \cdot (\frac{\xi}{aB} \sum_{n \in \mathcal{B}_b} \frac{(1-2p)\eta(t+1)^2}{4B} \sum_{n \in \mathcal{B}_b} \frac{m}{a} p_n(t))^2\Big).
\end{aligned}
\tag{192}
$$

$$
\|\boldsymbol{W}_{O_{(i,\cdot)}}^{(t)} (\boldsymbol{W}_V^{(t)} \boldsymbol{x}_s - \boldsymbol{W}_V^{(t)} \boldsymbol{x}_j^n)\| \lesssim \frac{\xi \eta t^2 m}{a^2} + \frac{\eta t m}{a} \cdot (\frac{\xi \eta t^2 m}{a^2})^2.
\tag{193}
$$

Given $i \in \mathcal{W}_l(0)$, with regard to $P_2$, we first consider the case when $t = 0$. Then with probability at least $1 - |\mathcal{S}^{n,t}|^{-C} \geq 1 - (MZ)^{-C'}$ for $C, C' > 0$, when $z = z_m$,

$$
\begin{aligned}
&\Big|\frac{1}{|\mathcal{S}^{l,t}|} \sum_{j=1}^{N} \mathbb{1}[j \in \mathcal{D}_i \cap \mathcal{N}_z^l \cap \mathcal{S}^{l,t}] - \mathbb{E}[\frac{1}{|\mathcal{S}^{l,t}|} \sum_{j=1}^{N} \mathbb{1}[j \in \mathcal{D}_i \cap \mathcal{N}_z^l \cap \mathcal{S}^{l,t}]]\Big| \\
=&\Big|\frac{|\mathcal{S}_i^{l,t} \cap \mathcal{N}_z^l|}{|\mathcal{S}^{l,t}|} - \frac{|\mathcal{D}_i \cap \mathcal{N}_z^l|}{N}\Big| \leq \sqrt{\frac{\log |\mathcal{S}^{n,t}|}{|\mathcal{S}^{n,t}|}} \leq \frac{1}{\operatorname{poly}(Z)},
\end{aligned}
\tag{194}
$$

$$
\begin{aligned}
&\Big|\frac{1}{|\mathcal{S}^{l,t}|} \sum_{j=1}^{N} \mathbb{1}[j \in \mathcal{S}^{l,t} \cap \mathcal{N}_z^l] - \mathbb{E}[\frac{1}{|\mathcal{S}^{l,t}|} \sum_{j=1}^{N} \mathbb{1}[j \in \mathcal{S}^{l,t} \cap \mathcal{N}_z^l]]\Big| \\
=&\Big|\frac{|\mathcal{S}^{l,t} \cap \mathcal{N}_z^l|}{|\mathcal{S}^{l,t}|} - \frac{|\mathcal{N}_z^l|}{N}\Big| \leq \sqrt{\frac{\log |\mathcal{S}^{n,t}|}{|\mathcal{S}^{n,t}|}} \leq \frac{1}{\operatorname{poly}(Z)}.
\end{aligned}
\tag{195}
$$

For $z = z_m$, if $y_l = 1$

$$
\frac{|(\mathcal{D}_1 \cup \mathcal{D}_2) \cap \mathcal{N}_z^l|}{|(\mathcal{D}_1 \cup \mathcal{D}_2)|} = \frac{|\mathcal{N}_z^l|}{N} \leq \frac{|\mathcal{D}_1 \cap \mathcal{N}_z^l|}{|\mathcal{D}_1|}.
\tag{196}
$$

Therefore, we have

$$
\frac{|\mathcal{S}_*^{l,t} \cap \mathcal{N}_z^l|}{|\mathcal{S}_*^{l,t}|} = \frac{|\mathcal{S}_*^{l,t} \cap \mathcal{N}_z^l|}{|\mathcal{S}_*^{l,t} \cap \mathcal{N}_z^l| + |\mathcal{S}_*^{l,t} \cap (\mathcal{V} - \mathcal{N}_z^l)|} \geq \frac{|\mathcal{N}_z^l|}{|\mathcal{N}_z^l| + |\mathcal{V} - \mathcal{N}_z^l|} - \frac{1}{\operatorname{poly}(Z)}.
\tag{197}
$$

For $i = 3, 4, \cdots, M$, when $z = z_m$, we can derive

$$
\frac{|\mathcal{S}_*^{l,t} \cap (\mathcal{V} - \mathcal{N}_z^l)|}{|\mathcal{S}_*^{l,t} \cap \mathcal{N}_z^l|} \leq \frac{|\mathcal{V} - \mathcal{N}_z^l|}{|\mathcal{N}_z^l|} + \frac{\Theta(1)}{\operatorname{poly}(Z)} \leq \frac{|\mathcal{S}_i^{l,t} \cap (\mathcal{V} - \mathcal{N}_z^l)|}{|\mathcal{S}_i^{l,t} \cap \mathcal{N}_z^l|} + \frac{\Theta(1)}{\operatorname{poly}(Z)},
\tag{198}
$$

$$
\frac{|\mathcal{S}_i^{l,t} \cap (\mathcal{V} - \mathcal{N}_z^l)|}{|\mathcal{S}_i^{l,t} \cap \mathcal{N}_z^l|} \leq \frac{|\mathcal{S}_\#^{l,t} \cap (\mathcal{V} - \mathcal{N}_z^l)|}{|\mathcal{S}_\#^{l,t} \cap \mathcal{N}_z^l|} + \frac{\Theta(1)}{\operatorname{poly}(Z)}.
\tag{199}
$$

Hence, we have

$$
\frac{|\mathcal{S}_*^{l,t} \cap \mathcal{N}_z^l|}{|\mathcal{S}^{l,t}|} \frac{|\mathcal{S}_i^{l,t} \cap (\mathcal{V} - \mathcal{N}_z^l)|}{|\mathcal{S}^{l,t}|} - \frac{|\mathcal{S}_i^{l,t} \cap \mathcal{N}_z^l|}{|\mathcal{S}^{l,t}|} \frac{|\mathcal{S}_*^{l,t} \cap (\mathcal{V} - \mathcal{N}_z^l)|}{|\mathcal{S}^{l,t}|} \geq -\frac{1}{\operatorname{poly}(z)},
\tag{200}
$$

$$
\frac{|\mathcal{S}_i^{l,t} \cap \mathcal{N}_z^l|}{|\mathcal{S}^{l,t}|} \frac{|\mathcal{S}_\#^{l,t} \cap (\mathcal{V} - \mathcal{N}_z^l)|}{|\mathcal{S}^{l,t}|} - \frac{|\mathcal{S}_\#^{l,t} \cap \mathcal{N}_z^l|}{|\mathcal{S}^{l,t}|} \frac{|\mathcal{S}_i^{l,t} \cap (\mathcal{V} - \mathcal{N}_z^l)|}{|\mathcal{S}^{l,t}|} \geq -\frac{1}{\operatorname{poly}(z)}.
\tag{201}
$$

Then take the case where $\boldsymbol{\mu}_1$ is the class-relevant pattern as an example, we have

$$
\begin{aligned}
P_3 + T_3 \gtrsim &\Big((1-\sigma)^2 \cdot \frac{|\mathcal{S}_1^{l,t} \cap \mathcal{N}_l^z|}{|\mathcal{S}^{l,t}|e} \cdot \frac{|(\mathcal{S}^{l,t} - \mathcal{N}_l^z) \cap \mathcal{S}_2^{l,t}|}{|\mathcal{S}^{l,t}|e} - (1+\sigma)^2 \cdot \frac{|(\mathcal{S}^{l,t} - \mathcal{N}_z^l) \cap \mathcal{S}_1^n|}{|\mathcal{S}^{l,t}|e} \\
&\cdot \frac{|\mathcal{S}^{l,t} \cap \mathcal{N}_l^z \cap \mathcal{S}_2^{l,t}|}{|\mathcal{S}^{l,t}|e}\Big) \cdot \eta \frac{(1-2p)^3}{B} \sum_{b=1}^{t} \sum_{n \in \mathcal{B}_b} \frac{p_n(b)m}{a} \cdot \Big(\frac{\xi \eta t^2 m}{a^2}\Big)^2 \|\boldsymbol{p}_1\|^2 + T_4 \\
\gtrsim &(1-\sigma)^2 \cdot \frac{|\mathcal{S}_1^{l,t}|}{|\mathcal{S}^{l,t}|} \frac{|\mathcal{S}_1^{l,t} \cap \mathcal{N}_z^l| - |\mathcal{S}_2^{l,t} \cap \mathcal{N}_z^l|}{|\mathcal{S}^{l,t}|} \cdot \eta \frac{(1-2p)^3}{B} \sum_{b=1}^{t} \sum_{n \in \mathcal{B}_b} \frac{p_n(b)m}{a} \Big(\frac{\xi \eta t^2 m}{a^2}\Big)^2 \|\boldsymbol{p}_1\|^2,
\end{aligned}
\tag{202}
$$

given that

$$
\begin{aligned}
T_4 := &\sum_{s \in \mathcal{S}^{l,t} \cap \mathcal{N}_l^z \cap (\mathcal{S}_\#^{l,t} \cup \mathcal{S}_*^{l,t})} \boldsymbol{W}_{O_{(i,\cdot)}} \boldsymbol{W}_V \boldsymbol{x}_s \mathrm{softmax}_l(\boldsymbol{x}_s \boldsymbol{W}_K^\top \boldsymbol{W}_Q \boldsymbol{x}_l + \boldsymbol{u}_{(s,l)}^\top \boldsymbol{b}^{(t)}) \\
&\cdot \sum_{r \in \mathcal{S}^{l,t} - \mathcal{N}_z^l - \mathcal{S}_\#^{l,t} - \mathcal{S}_*^{l,t}} \mathrm{softmax}_l(\boldsymbol{x}_r^\top \boldsymbol{W}_K^\top \boldsymbol{W}_Q \boldsymbol{x}_l + \boldsymbol{u}_{(r,l)}^\top \boldsymbol{b}^{(t)}) \\
&- \sum_{s \in (\mathcal{S}^{l,t} - \mathcal{N}_z^l) \cap (\mathcal{S}_\#^{l,t} \cup \mathcal{S}_*^{l,t})} \boldsymbol{W}_{O_{(i,\cdot)}} \boldsymbol{W}_V \boldsymbol{x}_s \mathrm{softmax}_l(\boldsymbol{x}_s \boldsymbol{W}_K^\top \boldsymbol{W}_Q \boldsymbol{x}_l + \boldsymbol{u}_{(s,l)}^\top \boldsymbol{b}^{(t)}) \\
&\cdot \sum_{r \in \mathcal{S}^{l,t} \cap \mathcal{N}_l^z - \mathcal{S}_\#^{l,t} - \mathcal{S}_*^{l,t}} \mathrm{softmax}_l(\boldsymbol{x}_r^\top \boldsymbol{W}_K^\top \boldsymbol{W}_Q \boldsymbol{x}_l + \boldsymbol{u}_{(r,l)}^\top \boldsymbol{b}^{(t)}),
\end{aligned}
\tag{203}
$$

and

$$
|T_4| \leq (\sigma + \tau) \frac{\eta^3 t^5 m^3 \xi^2}{a^5} \|\boldsymbol{p}\| \cdot \frac{1}{\mathrm{poly}(Z)}.
\tag{204}
$$

One can obtain the opposite conclusion if

$$
\begin{aligned}
&\frac{|\mathcal{S}_*^{l,t} \cap (\mathcal{V} - \mathcal{N}_z^l)|}{|\mathcal{S}_*^{l,t} \cap \mathcal{N}_z^l|} \geq \frac{|\mathcal{V} - \mathcal{N}_z^l|}{|\mathcal{N}_z^l|} + \frac{\Theta(1)}{\mathrm{poly}(Z)} \geq \frac{|\mathcal{S}_i^{l,t} \cap (\mathcal{V} - \mathcal{N}_z^l)|}{|\mathcal{S}_i^{l,t} \cap \mathcal{N}_z^l|} + \frac{\Theta(1)}{\mathrm{poly}(Z)} \\
&\geq \frac{|\mathcal{S}_\#^{l,t} \cap (\mathcal{V} - \mathcal{N}_z^l)|}{|\mathcal{S}_\#^{l,t} \cap \mathcal{N}_z^l|} + \frac{\Theta(1)}{\mathrm{poly}(Z)}.
\end{aligned}
\tag{205}
$$

We can conclude that $b_z$ will increase during the updates with the condition (198) and decrease with the condition (205). When $t$ is large, given that $|\mathcal{N}_z^l| = \Theta(|\mathcal{S}^{l,t}|)$, define

$$
K := \max_{z \in \mathcal{Z}}\{b_z^{(t)}\} - \min_{z \in \mathcal{Z}}\{b_z^{(t)}\}.
\tag{206}
$$

Therefore,

$$
\begin{aligned}
&P_3 + T_3 \\
\gtrsim &\Big((1-\sigma)^2 \frac{K|\mathcal{S}_1^{l,t} \cap \mathcal{N}_l^z| \cdot |(\mathcal{S}^{l,t} - \mathcal{N}_l^z) \cap \mathcal{S}_2^{l,t}|}{(K|\mathcal{S}^{l,t} \cap \mathcal{N}_z^l| + |\mathcal{S}^{l,t} - \mathcal{N}_z^l|)^2} - (1+\sigma)^2 \cdot \frac{|(\mathcal{S}^{l,t} - \mathcal{N}_z^l) \cap \mathcal{S}_1^n| \cdot K|\mathcal{S}^{l,t} \cap \mathcal{N}_l^z \cap \mathcal{S}_2^{l,t}|}{(K|\mathcal{S}^{l,t} \cap \mathcal{N}_z^l| + |\mathcal{S}^{l,t} - \mathcal{N}_z^l|)^2}\Big) \\
&\cdot \eta \frac{(1-2p)^3}{B} \sum_{b=1}^{t} \sum_{n \in \mathcal{B}_b} \frac{p_n(b)m}{a} \Big(\frac{\xi \eta t^2 m}{a^2}\Big)^2 \|\boldsymbol{p}_1\|^2 \\
\gtrsim &(1-\sigma)^2 \cdot \frac{K|\mathcal{S}_1^{l,t}| \cdot (|\mathcal{S}_1^{l,t} \cap \mathcal{N}_z^l| - |\mathcal{S}_2^{l,t} \cap \mathcal{N}_z^l|)}{(K|\mathcal{S}^{l,t} \cap \mathcal{N}_z^l| + |\mathcal{S}^{l,t} - \mathcal{N}_z^l|)^2} \cdot \eta \frac{(1-2p)^3}{B} \sum_{b=1}^{t} \sum_{n \in \mathcal{B}_b} \frac{p_n(b)m}{a} \Big(\frac{\xi \eta t^2 m}{a^2}\Big)^2 \|\boldsymbol{p}_1\|^2 \\
\gtrsim &(1-\sigma)^2 \cdot \frac{|\mathcal{S}_1^{l,t}|}{|\mathcal{S}^{l,t}|} \frac{|\mathcal{S}_1^{l,t} \cap \mathcal{N}_z^l| - |\mathcal{S}_2^{l,t} \cap \mathcal{N}_z^l|}{K|\mathcal{S}^{l,t}|} \cdot \eta \frac{(1-2p)^3}{B} \sum_{b=1}^{t} \sum_{n \in \mathcal{B}_b} \frac{p_n(b)m}{a} \Big(\frac{\xi \eta t^2 m}{a^2}\Big)^2 \|\boldsymbol{p}_1\|^2.
\end{aligned}
\tag{207}
$$

By combining (185), (190), (191), and 207, we can derive,

$$
-\eta\frac{1}{B}\sum_{n\in\mathcal{B}_b}\frac{\partial\mathbf{Loss}(\boldsymbol{X}^n,y_n)}{\partial b_z}
$$

$$
\gtrsim\eta\frac{1}{B}\sum_{n\in\mathcal{B}_b}\eta\frac{(1-2p)^3}{B}\sum_{b=1}^t\sum_{n\in\mathcal{B}_b}\frac{p_n(b)m^2}{a^2}(\frac{\xi\eta t^2m}{a^2})^2\|\boldsymbol{p}_1\|^2\cdot\frac{|\mathcal{S}_*^{l,t}|}{|\mathcal{S}^{l,t}|}\frac{|\mathcal{S}_*^{l,t}\cap\mathcal{N}_z^l|-|\mathcal{S}_\#^{l,t}\cap\mathcal{N}_z^l|}{K|\mathcal{S}^{l,t}|}.
\tag{208}
$$

If $u_{(s,l)_{z^*}}=1$,

$$
\boldsymbol{u}_{(s,l)}^\top(\boldsymbol{b}^{(t+1)}-\boldsymbol{b}^{(t)})=-\eta\frac{1}{B}\sum_{n\in\mathcal{B}_b}\frac{\partial\mathbf{Loss}(\boldsymbol{X}^n,y_n)}{\partial b_{z^*}}
\tag{209}
$$

$$
\boldsymbol{u}_{(s,l)}^\top\boldsymbol{b}^{(t)}
$$

$$
\geq\eta\frac{1}{B}\sum_{b=1}^t\sum_{n\in\mathcal{B}_b}\eta\frac{(1-2p)^3}{B}\sum_{b=1}^t\sum_{n\in\mathcal{B}_b}\frac{p_n(b)m^2}{a^2}(\frac{\xi\eta t^2m}{a^2})^2\|\boldsymbol{p}_1\|^2\cdot\frac{\gamma_d}{2}\frac{|\mathcal{S}_*^{l,t}\cap\mathcal{N}_z^l|-|\mathcal{S}_\#^{l,t}\cap\mathcal{N}_z^l|}{K|\mathcal{S}^{l,t}|}.
\tag{210}
$$

If we want to compute the difference term $b_{z_m}^{(t)}-b_z^{(t)}$, note that we only need to study the differences in $P_3+T_3$ given the previous analysis. Since that the term $\boldsymbol{W}_{O_{(i,\cdot)}}\boldsymbol{W}_V\boldsymbol{x}_s\mathrm{softmax}_l(\boldsymbol{x}_s\boldsymbol{W}_K^\top\boldsymbol{W}_Q\boldsymbol{x}_l+\boldsymbol{u}_{(s,l)}^\top\boldsymbol{b}^{(t)})$ is larger when $s\in\mathcal{N}_{z_m}^l$, we can bound the difference $P_3+T_3$ using terms in (207). To find the lower bound, we apply the result in (192) and then directly use the fraction of sampled nodes in different neighborhoods because concentration bounds can control the error. To be more specific, on the one hand, if $\mathcal{N}_z^l$ is too small for one $z\in[Z-1]$ and $l\in\mathcal{V}$, the left-hand side of (200), (201), and (202) are close to zero, and these three equations still hold. On the other hand, if we want to see whether terms (200) and (201) with $z=z_m$ are larger them with other $z\neq z_m$, we have the following derivation. Take (200) as an example,

$$
|\mathcal{D}_*^l\cap\mathcal{N}_z^l||\mathcal{D}_i\cap(\mathcal{V}-\mathcal{N}_z^l)|-|\mathcal{D}_i\cap\mathcal{N}_z^l||\mathcal{D}_*^l\cap(\mathcal{V}-\mathcal{N}_z^l)|=|\mathcal{D}_*^l\cap\mathcal{N}_z^l|\cdot|\mathcal{D}_i|-|\mathcal{D}_i\cap\mathcal{N}_z^l|\cdot|\mathcal{D}_*^l|.
\tag{211}
$$

$$
\begin{aligned}
&|\mathcal{D}_*^l\cap\mathcal{N}_{z_m}^l||\mathcal{D}_i\cap(\mathcal{V}-\mathcal{N}_{z_m}^l)|-|\mathcal{D}_i\cap\mathcal{N}_{z_m}^l||\mathcal{D}_*^l\cap(\mathcal{V}-\mathcal{N}_{z_m}^l)|\\
&-(|\mathcal{D}_*^l\cap\mathcal{N}_z^l||\mathcal{D}_i\cap(\mathcal{V}-\mathcal{N}_z^l)|-|\mathcal{D}_i\cap\mathcal{N}_z^l||\mathcal{D}_*^l\cap(\mathcal{V}-\mathcal{N}_z^l)|)\\
&=(|\mathcal{D}_*^l\cap\mathcal{N}_{z_m}^l|-|\mathcal{D}_*^l\cap\mathcal{N}_z^l|)\cdot|\mathcal{D}_i|-(|\mathcal{D}_i\cap\mathcal{N}_{z_m}^l|-|\mathcal{D}_i\cap\mathcal{N}_z^l|)\cdot|\mathcal{D}_*^l|\\
&=(|\mathcal{N}_{z_m}^l|-|\mathcal{N}_z^l|)\cdot\frac{\gamma_d}{2}|\mathcal{D}_i|-(|\mathcal{D}_i\cap\mathcal{N}_{z_m}^l|-|\mathcal{D}_i\cap\mathcal{N}_z^l|)\cdot|\mathcal{D}_*^l|\\
&\quad+(|\mathcal{D}_*^l\cap\mathcal{N}_{z_m}^l|-|\mathcal{N}_{z_m}^l|\frac{\gamma_d}{2}-(|\mathcal{D}_*^l\cap\mathcal{N}_z^l|-|\mathcal{N}_z^l|\frac{\gamma_d}{2}))\cdot|\mathcal{D}_i|\\
&=\frac{1}{2}(|\mathcal{D}_*^l\cap\mathcal{N}_{z_m}^l|-|\mathcal{D}_\#^l\cap\mathcal{N}_{z_m}^l|-(|\mathcal{D}_*^l\cap\mathcal{N}_z^l|-|\mathcal{D}_\#^l\cap\mathcal{N}_z^l|))\cdot|\mathcal{D}_i|\\
&\geq0,
\end{aligned}
\tag{212}
$$

where the first step is by (211), the second step comes from mathematical derivation, the third step is obtained from that $\boldsymbol{\mu}_i,\ i=2,3,\cdots,M$ is uniformly distributed in the whole graph, and the last step is by the definition of $z_m$ in (6). We can derive (201) in the same way. Hence,

$$
b_{z_m}^{(t)}-b_z^{(t)}
$$

$$
\gtrsim\eta\frac{1}{B}\sum_{b=1}^t\sum_{n\in\mathcal{B}_b}\eta\frac{(1-2p)^3}{B}\sum_{b=1}^t\sum_{n\in\mathcal{B}_b}\frac{p_n(b)m^2}{a^2}(\frac{\xi\eta t^2m}{a^2})^2\|\boldsymbol{p}_1\|^2\cdot\frac{\gamma_d}{2}
$$

$$
\cdot(\frac{|\mathcal{S}_*^{l,t}\cap\mathcal{N}_{z_m}^l|-|\mathcal{S}_\#^{l,t}\cap\mathcal{N}_{z_m}^l|}{K|\mathcal{S}^{l,t}|}-\frac{|\mathcal{S}_*^{l,t}\cap\mathcal{N}_z^l|-|\mathcal{S}_\#^{l,t}\cap\mathcal{N}_z^l|}{K|\mathcal{S}^{l,t}|}).
\tag{213}
$$

Note that finally $\eta T=\Theta(1)$. Therefore, $K=\Theta(1)$.

**Proof of Lemma 7**:
For the gradient of $\boldsymbol{W}_V$,

$$
\begin{aligned}
\frac{\partial \overline{\mathbf{Loss}}_b}{\partial \boldsymbol{W}_V} =& \frac{1}{B} \sum_{n \in \mathcal{B}_b} \frac{\partial \mathbf{Loss}(\boldsymbol{X}^n, y_n)}{\partial F(\boldsymbol{X}^n)} \frac{\partial F(\boldsymbol{X}^n)}{\partial \boldsymbol{W}_V} \\
=& \frac{1}{B} \sum_{n \in \mathcal{B}_b} \sum_{i=1}^m (-y_l) a_i \mathbb{1}[\boldsymbol{W}_{O_{(i,\cdot)}} \sum_{s \in \mathcal{S}^{l,t}} \boldsymbol{W}_V \boldsymbol{x}_l \text{softmax}_l(\boldsymbol{x}_s{}^\top \boldsymbol{W}_K^\top \boldsymbol{W}_Q \boldsymbol{x}_l + \boldsymbol{u}_{(s,l)}^\top \boldsymbol{b}) \geq 0] \\
& \cdot \boldsymbol{W}_{O_{(i,\cdot)}}{}^\top \sum_{s \in \mathcal{S}^{l,t}} \text{softmax}_l(\boldsymbol{x}_s{}^\top \boldsymbol{W}_K^\top \boldsymbol{W}_Q \boldsymbol{x}_l + \boldsymbol{u}_{(s,l)}^\top \boldsymbol{b})^\top \boldsymbol{x}_s{}^\top.
\end{aligned}
\tag{214}
$$

Consider a node $n$ where $y_n = 1$. Let $l \in \mathcal{S}_1^{n,t}$

$$
\sum_{s \in \mathcal{S}_1^{n,t}} \text{softmax}_n(\boldsymbol{x}_s{}^\top \boldsymbol{W}_K^{(t)}{}^\top \boldsymbol{W}_Q^{(t)} \boldsymbol{x}_n + \boldsymbol{u}_{(s,n)}^\top \boldsymbol{b}^{(t)}) \geq p_n(t).
\tag{215}
$$

Then for $j \in \mathcal{S}_1^{g,t}$, $g \in \mathcal{V}$,

$$
\begin{aligned}
& \frac{1}{B} \sum_{n \in \mathcal{B}_b} \frac{\partial \mathbf{Loss}(\boldsymbol{X}^n, y_n)}{\partial \boldsymbol{W}_V^{(t)}} \Big| \boldsymbol{W}_V^{(t)} \boldsymbol{x}_j \\
=& \frac{1}{B} \sum_{l \in \mathcal{B}_b} (-y_l) \sum_{i=1}^m a_i \mathbb{1}[\boldsymbol{W}_{O_{(i,\cdot)}}^{(t)} \sum_{s \in \mathcal{S}^{l,t}} \text{softmax}_l(\boldsymbol{x}_s{}^\top \boldsymbol{W}_K^{(t)}{}^\top \boldsymbol{W}_Q^{(t)} \boldsymbol{x}_l + \boldsymbol{u}_{(s,l)}^\top \boldsymbol{b}^{(t)}) \boldsymbol{W}_V^{(t)} \boldsymbol{x}_s \geq 0] \\
& \cdot \boldsymbol{W}_{O_{(i,\cdot)}}^{(t)}{}^\top \sum_{s \in \mathcal{S}^{n,t}} \text{softmax}_l(\boldsymbol{x}_s{}^\top \boldsymbol{W}_K^{(t)}{}^\top \boldsymbol{W}_Q^{(t)} \boldsymbol{x}_l + \boldsymbol{u}_{(s,l)}^\top \boldsymbol{b}^{(t)}) \boldsymbol{x}_s{}^\top \boldsymbol{x}_j \\
=& \Theta(1) \cdot ( \sum_{i \in \mathcal{W}_l(0)} V_i(t) \boldsymbol{W}_{O_{(i,\cdot)}}{}^\top + \sum_{i \notin \mathcal{W}_l(0)} \lambda V_i(t) \boldsymbol{W}_{O_{(i,\cdot)}}{}^\top ),
\end{aligned}
\tag{216}
$$

If $i \in \mathcal{W}_l(0)$, we have

$$
V_i(t) \lesssim \frac{1-2p}{2B} \sum_{n \in \mathcal{B}_{b+}} -\frac{1}{a} p_n(t).
\tag{217}
$$

Similarly, if $i \in \mathcal{U}_l(t)$,

$$
V_i(t) \gtrsim \frac{1-2p}{2B} \sum_{n \in \mathcal{B}_{b-}} \frac{1}{a} p_n(t),
\tag{218}
$$

if $i$ is an unlucky neuron, by Hoeffding's inequality in (19), we have

$$
|V_i(t)| \lesssim \frac{1}{\sqrt{B}} \cdot \frac{1}{a}.
\tag{219}
$$

Therefore, we can derive

$$
\begin{aligned}
& -\eta \sum_{b=1}^t \boldsymbol{W}_{O_{(i,\cdot)}}^{(b)} \sum_{j \in \mathcal{W}_l(0)} V_j(b) \boldsymbol{W}_{O_{(j,\cdot)}}^{(b)}{}^\top \\
\gtrsim& \eta m \frac{1-2p}{2B} \sum_{b=1}^t \sum_{n \in \mathcal{B}_{b+}} \frac{1}{a} p_n(b) \cdot (\frac{\xi}{aB} \sum_{n \in \mathcal{B}_b} \frac{\eta t^2 (1-2p)}{4B} \sum_{n \in \mathcal{B}_b} \frac{m}{a} p_n(t))^2,
\end{aligned}
\tag{220}
$$

$$
\begin{aligned}
& |\eta \sum_{b=1}^t \boldsymbol{W}_{O_{(i,\cdot)}}^{(b)} \sum_{j \in \mathcal{U}_{l,n}(0)} V_j(b) \boldsymbol{W}_{O_{(j,\cdot)}^{(b)}}{}^\top| \\
\lesssim& \frac{\eta}{B} \sum_{b=1}^t \sum_{n \in \mathcal{B}_b} \frac{p_n(b) m}{a} \|\boldsymbol{W}_{O_{(i,\cdot)}}^{(t)}\|^2 \|\boldsymbol{p}_1\|^2,
\end{aligned}
\tag{221}
$$

$$-\eta t \boldsymbol{W}_{O_{(i,\cdot)}} \sum_{j \notin (\mathcal{W}_l(0) \cup \mathcal{U}_{l,n}(0))} V_j(t) \boldsymbol{W}_{O_{(j,\cdot)}}^{\top} \lesssim \frac{\eta t m \|\boldsymbol{p}\|^2}{Ba} \|\boldsymbol{W}_{O_{(i,\cdot)}}^{(t)}\|^2. \tag{222}$$

Hence,
(1) If $j \in \mathcal{S}_1^{n,t}$ for one $n \in \mathcal{V}$,

$$
\begin{aligned}
\boldsymbol{W}_V^{(t+1)} \boldsymbol{x}_j^n &= \boldsymbol{W}_V^{(t)} \boldsymbol{x}_j^n - \eta \Big( \frac{\partial \mathbf{Loss}(\boldsymbol{X}^n, y_n)}{\partial \boldsymbol{W}_V} \Big| \boldsymbol{W}_V^{(t)} \Big) \boldsymbol{x}_j^n \\
&= \boldsymbol{p}_1 - \eta \sum_{b=1}^{t+1} \sum_{i \in \mathcal{W}(_n b)} V_i(b) \boldsymbol{W}_{O_{(i,\cdot)}}^{(b)}{}^{\top} - \eta \sum_{b=1}^{t+1} \sum_{i \notin \mathcal{W}_n(b)} \lambda V_i(b) \boldsymbol{W}_{O_{(i,\cdot)}}^{(b)}{}^{\top} + \boldsymbol{z}_j(t).
\end{aligned}
\tag{223}
$$

(2) If $j \in \mathcal{S}_2^{n,t}$, we have

$$
\begin{aligned}
\boldsymbol{W}_V^{(t+1)} \boldsymbol{x}_j &= \boldsymbol{W}_V^{(0)} \boldsymbol{x}_j^n - \eta \Big( \frac{\partial \mathbf{Loss}(\boldsymbol{X}^n, y_n)}{\partial \boldsymbol{W}_V} \Big| \boldsymbol{W}_V^{(0)} \Big) \boldsymbol{x}_j^n \\
&= \boldsymbol{p}_2 - \eta \sum_{b=1}^{t+1} \sum_{i \in \mathcal{U}(b)} V_i(b) \boldsymbol{W}_{O_{(i,\cdot)}}^{(b)}{}^{\top} - \eta \sum_{b=1}^{t+1} \sum_{i \notin \mathcal{U}(b)} \lambda V_i(b) \boldsymbol{W}_{O_{(i,\cdot)}}^{(b)}{}^{\top} + \boldsymbol{z}_j(t).
\end{aligned}
\tag{224}
$$

(3) If $j \in \mathcal{S}^{n,t}/(\mathcal{S}_1^{n,t} \cup \mathcal{S}_2^{n,t})$, we have

$$
\begin{aligned}
\boldsymbol{W}_V^{(t+1)} \boldsymbol{x}_j^n &= \boldsymbol{W}_V^{(0)} \boldsymbol{x}_j^n - \eta \Big( \frac{\partial \mathbf{Loss}(\boldsymbol{X}^n, y_n)}{\partial \boldsymbol{W}_V} \Big| \boldsymbol{W}_V^{(0)} \Big) \boldsymbol{x}_j^n \\
&= \boldsymbol{p}_k - \eta \sum_{b=1}^{t+1} \sum_{i=1}^{m} \lambda V_i(b) \boldsymbol{W}_{O_{(i,\cdot)}}^{(b)}{}^{\top} + \boldsymbol{z}_j(t).
\end{aligned}
\tag{225}
$$

Here

$$\|\boldsymbol{z}_j(t)\| \le (\sigma + \tau) \tag{226}$$

for $t \ge 1$. Note that this Lemma also holds when $t = 1$.

**Proof of Lemma 9:**
We prove this lemma by induction.
When $t = 0$. For $i \in \mathcal{W}_l(0)$ and $l \in \mathcal{D}_1$, we have that

$$\boldsymbol{W}_{O_{(i,\cdot)}}^{(0)} \Big( \sum_{s \in \mathcal{S}_1^{l,t}} \mathrm{softmax}_l(\boldsymbol{x}_s^{\top} \boldsymbol{W}_K^{(t)}{}^{\top} \boldsymbol{W}_Q^{(t)} \boldsymbol{x}_l + 0) \boldsymbol{p}_1 + \boldsymbol{z}(0) + \sum_{j \ne 1} W_j^n(0) \boldsymbol{p}_j \Big) \gtrsim \xi(\Theta(1) - \sigma - \tau) > 0. \tag{227}$$

Hence, the conclusion holds. When $t = 1$, we have

$$
\begin{aligned}
&\boldsymbol{W}_{O_{(i,\cdot)}}^{(t)} \boldsymbol{V}_l^n(t) \\
&= \boldsymbol{W}_{O_{(i,\cdot)}}^{(t)} \Big( \sum_{s \in \mathcal{S}_1^n} \mathrm{softmax}_l(\boldsymbol{x}_s^{\top} \boldsymbol{W}_K^{(t)}{}^{\top} \boldsymbol{W}_Q^{(t)} \boldsymbol{x}_l + \boldsymbol{u}_{(s,l)}^{\top} \boldsymbol{b}^{(t)}) \boldsymbol{p}_1 + \boldsymbol{z}(t) + \sum_{j \ne 1} W_j^n(t) \boldsymbol{p}_j \\
&\quad - \eta \sum_{b=0}^{t-1} \Big( \sum_{i \in \mathcal{W}_l(0)} V_i(b) \boldsymbol{W}_{O_{(i,\cdot)}}^{(b)}{}^{\top} + \sum_{i \notin \mathcal{W}_l(0)} V_i(b) \lambda \boldsymbol{W}_{O_{(i,\cdot)}}^{(b)}{}^{\top} \Big) \Big).
\end{aligned}
\tag{228}
$$

Denote $\theta_l^i$ as the angle between $\boldsymbol{V}_l(0)$ and $\boldsymbol{W}_{O_{(i,\cdot)}}^{(0)}$. Since that $\boldsymbol{W}_{O_{(j,\cdot)}}^{(0)}$ is initialized uniformed on the $m_a - 1$-sphere, we have $\mathbb{E}[\theta_l^i] = 0$. By Hoeffding's inequality (19), we have

$$\Big\| \frac{1}{|\mathcal{W}_l(0)|} \sum_{i \in \mathcal{W}_l(0)} \theta_l^i - \mathbb{E}[\theta_l^i] \Big\| = \Big\| \frac{1}{|\mathcal{W}_l(0)|} \sum_{i \in \mathcal{W}_l(0)} \theta_l^i \Big\| \le \sqrt{\frac{\log N}{m}}, \tag{229}$$

with probability of at least $1 - N^{-10}$. When $m \gtrsim M^2 \log N$, we can obtain that

$$\Big\| \frac{1}{|\mathcal{W}_l(0)|} \sum_{i \in \mathcal{W}_l(0)} \theta_l^i - \mathbb{E}[\theta_l^i] \Big\| \le O(\frac{1}{M}). \tag{230}$$

Therefore, for $i \in \mathcal{W}_l(0)$, we have

$$\boldsymbol{W}_{O_{(i,\cdot)}} \sum_{b=0}^{t-1} \sum_{i \in \mathcal{W}_l(0)} \boldsymbol{W}_{O_{(i,\cdot)}}^{(b)} > 0. \tag{231}$$

Similarly, we have that $\sum_{b=0}^{t-1} \sum_{i \notin \mathcal{W}_l(0)} \boldsymbol{W}_{O_{(i,\cdot)}}^{(b)}$ is close to $-\boldsymbol{V}_l^n(0)$. Given that $\lambda < 1$, we can approximately acquire that

$$-\boldsymbol{W}_{O_{(i,\cdot)}}^{(0)} \eta \sum_{b=0}^{t-1} \Big( \sum_{i \in \mathcal{W}_l(0)} V_i(b) \boldsymbol{W}_{O_{(i,\cdot)}}^{(b)}{}^\top + \sum_{i \notin \mathcal{W}_l(0)} V_i(b) \lambda \boldsymbol{W}_{O_{(i,\cdot)}}^{(b)}{}^\top \Big) > 0. \tag{232}$$

After the first iterations, we know that $\boldsymbol{u}_{(s,l)}^\top \boldsymbol{b}^{(t)}$ increases the most from $\boldsymbol{u}_{(s,l)}^\top \boldsymbol{b}^{(0)}$ by $\gamma_d$ fraction of discriminative nodes if $s \in \mathcal{N}_{z_m}^l$. Because the softmax is based exponential functions, the most significance increase in $\mathcal{N}_{z_m}^l$ enlarges $\sum_{s \in \mathcal{S}_1^{l,t}} \mathrm{softmax}_l(\boldsymbol{x}_s^\top \boldsymbol{W}_K^{(t)}{}^\top \boldsymbol{W}_Q^{(t)} \boldsymbol{x}_l + \boldsymbol{u}_{(s,l)}^\top \boldsymbol{b}^{(t)})$. Since that $i \in \mathcal{W}_n(0)$, we then have

$$\boldsymbol{W}_{O_{(i,\cdot)}}^{(0)} \Big( \sum_{s \in \mathcal{S}_1^n} \mathrm{softmax}_l(\boldsymbol{x}_s^\top \boldsymbol{W}_K^{(t)}{}^\top \boldsymbol{W}_Q^{(t)} \boldsymbol{x}_l + \boldsymbol{u}_{(s,l)}^\top \boldsymbol{b}^{(t)}) \boldsymbol{p}_1 + \boldsymbol{z}(t) + \sum_{j \neq 1} W_j^n(t) \boldsymbol{p}_j > 0. \tag{233}$$

Therefore, we have

$$\boldsymbol{W}_{O_{(i,\cdot)}}^{(0)} \boldsymbol{V}_l^n(t) > 0. \tag{234}$$

Meanwhile, the addition from $\boldsymbol{W}_{O_{(i,\cdot)}}^{(0)}$ to $\boldsymbol{W}_{O_{(i,\cdot)}}^{(1)}$ is approximately a summation of multiple $\boldsymbol{V}_j(0)$ such that $\boldsymbol{W}_{O_{(i,\cdot)}}^{(0)} \boldsymbol{V}_j(0) > 0$ and $j \in \mathcal{S}_1^n$. Therefore, $\boldsymbol{V}_j(0)^\top \boldsymbol{V}_l(0) > 0$. Therefore, we can obtain

$$\boldsymbol{W}_{O_{(i,\cdot)}}^{(t)} \boldsymbol{V}_l(t) > 0. \tag{235}$$

**(2) Suppose that the conclusion holds when $t = s$. When $t = s + 1$**, we can follow the derivation of the case where $t = 1$. Although the unit vector of $\boldsymbol{W}_{O_{(i,\cdot)}}^{(t)}$ no longer follows a uniform distribution, we know that (229) holds since the angle is bounded and has a mean which is very close to $\boldsymbol{V}_l(0)$. Then, the conclusion still holds.

One can develop the proof for $\mathcal{U}_{l,n}(0)$ following the above steps.

## E  EXTENSION OF OUR ANALYSIS AND ADDITIONAL DISCUSSION

### E.1  ASSUMPTION ON THE PRE-TRAINED MODEL

For the assumption on the pre-trained model, we provide the following discussion.

We would like to clarify that the training problem of the graph transformer (GT) is very challenging to analyze due to its significant non-convexity, and some form of assumptions is needed to facilitate the analysis. In fact, even for the conventional Transformers, the existing state-of-the-art theoretical optimization and generalization analyses all make some assumptions on the data, embedding or initial models, or make further simplifications on the Transformer model. For example, (Oymak et al., 2023) assumes orthogonality on the raw data. Jelassi et al. (2022) simplifies the self-attention layer by only considering the positional encoding (PE). Tian et al. (2023); Li et al. (2023b) use linear activation in the MLP layer. About the initialization, (Li et al., 2023a) assumes orthogonality on the initialization of embeddings. Tarzanagh et al. (2023) requires that the initialization of the query embedding is close to the optimal solution.

The initialization assumption made in our manuscript is the same as (Li et al., 2023a) but for a GT. We would like to emphasize that our initialization assumption is at least no stronger than the existing initialization assumptions in (Li et al., 2023a) and (Tarzanagh et al., 2023). Notably, our work proposes a novel theoretical framework for the training dynamics and generalization of GT for the first time, where the number of trainable parameters is more than the above existing works. Third, although we have assumptions on the initialization for theoretical analysis, our experiments on real-world datasets in Section 5.2 are implemented from random initialization. The performance is aligned with our theoretical findings.

### E.2 EXTENSION TO OTHER POSITIONAL ENCODINGS

Our theoretical analysis is general and can be applied to different positional encodings. Specifically, Theorem 4.1 is based on proving these two parts, (i) the success of positional encoding, i.e., the positional encoding can identify the correct structure information (which is the core neighborhood in our data model), (ii) if structural information is known, analyzing the sample complexity and convergence rate of Graph Transformer. We next discuss the extension of both parts to other positional encoding separately.

For part (i), the success of positional encoding, because different types of positional encoding can learn different types of structure information the best, this analysis needs to be case-by-case for different positional encoding. However, our technique and insight can be potentially useful with some modifications to other positional encodings. For example, Laplacian eigenvectors can essentially divide a graph into several clusters considering its relationship to spectral clustering (Von Luxburg, 2007) and would work best for a data model where data labels depend on clusters. Moreover, Random Walk PE can encode structural information such as whether the node is part of an m-long circle (Rampášek et al., 2022). Degree PE (Ying et al., 2021), one of the standard centrality measures, can capture the local degree information. PE using distance from the centroid of the whole graph (Rampášek et al., 2022) can represent global distance information. Our techniques in analyzing the core neighborhood can be useful in analyzing these positional encodings. Similarly to our framework of the core neighborhood, where a large amount of class-relevant nodes is located, one can respectively construct data models for these positional encodings where class-relevant nodes are dominant for nodes within the corresponding structures, such as a cluster, an m-long circle, a certain degree, and a certain distance to the centroid of the whole graph. The remaining step of the generalization analysis is to learn this data model by Graph Transformer using positional encoding, which is elaborated in detail in the next paragraph.

For part (ii), our proof technique can be easily generalized to other positional encodings with some straightforward transformation. Specifically, positional encoding can be divided into absolute and relative positional encodings. What we study in this work belongs to relative positional encodings. Absolute positional encodings can be formulated as a concatenation to the initial node feature, either by their raw definition (Kreuzer et al., 2021; Rampášek et al., 2022) or by transferring from a bias term (Gabrielsson et al., 2022) ($\boldsymbol{W}\boldsymbol{x} + \boldsymbol{a} = (\boldsymbol{W}, \boldsymbol{W}')(\boldsymbol{x}^\top, \boldsymbol{b}'^\top)^\top$, where the trainable positional encoding $\boldsymbol{a}$ is transferred into a fixed augmented feature $\boldsymbol{b}'$ and a trainable augmented weight $\boldsymbol{W}'$). The structural information is then incorporated into the node representation $(\boldsymbol{x}, \boldsymbol{b}')$. Denote the positional encoding $\boldsymbol{b}'$ for a query node $q$ as $\boldsymbol{b}'_q$. Denote the PE of one core-neighborhood node $c$ and one other neighboring node $o$ for this query node as $\boldsymbol{b}'_c$, and $\boldsymbol{b}'_o$, respectively. Suppose all the $\boldsymbol{b}'$ are normalized. Then, given that the defined positional encoding $\boldsymbol{b}'$ can locate the core neighborhood, i.e., the distance between $\boldsymbol{b}'_c$ and $\boldsymbol{b}'_q$ is much smaller than the distance between $\boldsymbol{b}'_o$ and $\boldsymbol{b}'_q$, we can deduce that the inner product between $\boldsymbol{b}'_c$ and $\boldsymbol{b}'_q$ is much larger than the inner product between $\boldsymbol{b}'_o$ and $\boldsymbol{b}'_q$ This leads to a dominant attention weight between the query node $q$ and the core-neighborhood node $c$ based on the definition of self-attention. Then, one could ignore other neighbors and focus only on core-neighborhood nodes when computing the Graph Transformer output. Then the proof in Theorem 4.1 for part (ii) applies directly.

### E.3 EXTENSION OF THE ANALYSIS ON GAT

From a high-level understanding, a one-layer GAT can be regarded as a Graph Transformer that only uses distance-1 neighborhood information. Therefore, our Theorem 4.3 can be applied to analyze the generalization of a one-layer GAT when its self-attention follows the self-attention mechanism in 1 of our manuscript, given the distance-1 neighborhood as the core neighborhood. From a perspective of training dynamics, GATs also share a common mechanism that computes the aggregation based on the similarity between node features as Graph Transformer does, although the attention layer of GAT (Veličković et al., 2018) is different. In this sense, one-layer GAT can generalize as well as Graph Transformer if the graph satisfies that the latent core neighborhood is the distance-1/distance-small neighborhood, such as homophilous graphs. The generalization analysis of using GATs on graphs with a larger distance of core neighborhoods and its comparison with graph transformers needs more effort, and we will leave it as future work.

### E.4 EXTENSION TO GRAPH CLASSIFICATION PROBLEMS

Since we aim to make a comparison with GCN, which focuses more on node classification tasks, our work also mainly studies node classification. However, our analysis is extendable to graph classification tasks. Consider a supervised-learning binary classification problem on a set of graphs $\{\mathcal{G}_i\}_{i=1}^N$. Denote the feature matrix of the graph $\mathcal{G}_i$ by $\boldsymbol{X}_i$. Following (Ying et al., 2021; Kreuzer et al., 2021), we apply "Mean" or "Sum" as the READOUT function. Hence, we have

$$F(\boldsymbol{X}_i) = K \sum_{n \in \mathcal{T}^i} \boldsymbol{a}_n^\top \text{Relu}(\boldsymbol{W}_O \sum_{s \in \mathcal{T}^i} \boldsymbol{W}_V \boldsymbol{x}_s \text{softmax}_n(\boldsymbol{x}_s^\top \boldsymbol{W}_K^\top \boldsymbol{W}_Q \boldsymbol{x}_n + \boldsymbol{u}_{(s,n)}^\top \boldsymbol{b})). \quad (236)$$

where $K = 1$ if READOUT is "Sum", and $K = 1/|\mathcal{T}_i|$ if READOUT is "Mean". When we compute the gradients, the only difference is that we sum up or average over all nodes in each graph.

**Data Model** The data model follows from Section 4.2. The difference is that the core neighborhood is defined based on the graph label, i.e., we assume the ground truth graph label is determined by the summation/mean of the majority vote of $\boldsymbol{\mu}_1$, $\boldsymbol{\mu}_2$ nodes in the core neighborhood for some nodes in each graph. This is motivated by graph classification on social networks, where the connections between the central person and other people in the graph decide the graph label. For example, if $z_m = 2$ and the distance-$z_m$ neighborhood of nodes in $\mathcal{R}^i$ determines the label, then for the ground truth graph label $\tilde{y}_i = 1$, $|\mathcal{D}_1^i \cap (\cup_{j \in \mathcal{R}^i} \mathcal{N}_{z_m}^j)|$ is larger than $|\mathcal{D}_2^i \cap (\cup_{j \in \mathcal{R}^i} \mathcal{N}_{z_m}^j)|$, where $\mathcal{D}_1^i$ and $\mathcal{D}_2^i$ are the set of $\boldsymbol{\mu}_1$ nodes and $\boldsymbol{\mu}_2$ nodes in $\mathcal{G}_i$. Such a data model ensures that the graph label comes from the graph structure. Meanwhile, it prevents us from assuming a more trivial model where the number of $\boldsymbol{\mu}_1$ nodes and $\boldsymbol{\mu}_2$ nodes in each graph indicates the label and no graph information is used, which is almost the same as that in the ViT work (Li et al., 2023a). Hence, when we compute the graph-level output, the distance-$z_m$ neighborhood of nodes in $\mathcal{R}^i$ still plays a vital role. Then, we can apply the generalization analysis of node classification based on the core neighborhood to the graph classification problem.

### E.5 EXTENSION TO MULTI-CLASSIFICATION

Consider the classification problem with four classes. We use the label $y \in \{+1, -1\}^2$ to denote the corresponding class. Similarly to the previous setup, there are four orthogonal discriminative patterns. We have $\boldsymbol{a} = (\boldsymbol{a}_1, \boldsymbol{a}_2)$, $\boldsymbol{W}_O = (\boldsymbol{W}_{O_1}, \boldsymbol{W}_{O_2})$, $\boldsymbol{W}_V = (\boldsymbol{W}_{V_1}, \boldsymbol{W}_{V_2})$, $\boldsymbol{W}_K = (\boldsymbol{W}_{K_1}, \boldsymbol{W}_{K_2})$, $\boldsymbol{W}_Q = (\boldsymbol{W}_{Q_1}, \boldsymbol{W}_{Q_2})$, and $\boldsymbol{b} = (\boldsymbol{b}_1, \boldsymbol{b}_2)$. Hence, we define

$$\boldsymbol{F}(\boldsymbol{x}_n) = (F_1(\boldsymbol{x}_n), F_2(\boldsymbol{x}_n)), \quad (237)$$

$$F_1(\boldsymbol{x}_n) = \boldsymbol{a}_1^\top \text{Relu}(\boldsymbol{W}_{O_1} \sum_{s \in \mathcal{T}_1^n} \boldsymbol{W}_{V_1} \boldsymbol{x}_s \text{softmax}_n(\boldsymbol{x}_s^\top \boldsymbol{W}_{K_1}^\top \boldsymbol{W}_{Q_1} \boldsymbol{x}_n + \boldsymbol{u}_{(s,n)}^\top \boldsymbol{b}_1)), \quad (238)$$

$$F_2(\boldsymbol{x}_n) = \boldsymbol{a}_2^\top \text{Relu}(\boldsymbol{W}_{O_2} \sum_{s \in \mathcal{T}_2^n} \boldsymbol{W}_{V_2} \boldsymbol{x}_s \text{softmax}_n(\boldsymbol{x}_s^\top \boldsymbol{W}_{K_2}^\top \boldsymbol{W}_{Q_2} \boldsymbol{x}_n + \boldsymbol{u}_{(s,n)}^\top \boldsymbol{b}_2)). \quad (239)$$

The dataset $\mathcal{D}$ can be divided into four groups as

$$\begin{aligned}
\mathcal{A}_1 &= \{(\boldsymbol{X}^n, \boldsymbol{y}_n) | \boldsymbol{y}_n = (1, 1)\}, \\
\mathcal{A}_2 &= \{(\boldsymbol{X}^n, \boldsymbol{y}_n) | \boldsymbol{y}_n = (1, -1)\}, \\
\mathcal{A}_3 &= \{(\boldsymbol{X}^n, \boldsymbol{y}_n) | \boldsymbol{y}_n = (-1, 1)\}, \\
\mathcal{A}_4 &= \{(\boldsymbol{X}^n, \boldsymbol{y}_n) | \boldsymbol{y}_n = (-1, -1)\}.
\end{aligned} \quad (240)$$

The hinge loss function for data $(\boldsymbol{X}^n, \boldsymbol{y}_n)$ will be

$$\text{Loss}(\boldsymbol{x}_n, \boldsymbol{y}_n) = \max\{1 - \boldsymbol{y}_n^\top \boldsymbol{F}(\boldsymbol{x}_n), 0\}. \quad (241)$$

Therefore, when computing the gradient, the problem becomes a binary classification. One can make derivations following the binary case. One notable difference is that we can assume two core neighborhoods for this four-classification problem.

## E.6 Comparision with other frameworks of analysis

In this section, we provide a comparison with other frameworks of analysis.

First, we focus on five other frameworks: Rademacher complexity, algorithmic stability, PAC-Bayesian, model recovery, and neural tangent kernel (NTK). Rademacher complexity (Tolstikhin et al., 2014; Garg et al., 2020; Esser et al., 2021), algorithmic stability (Verma & Zhang, 2019), and PAC-Bayesian (Liao et al., 2021) only focus on the generalization gap, which is the difference between the empirical risk and the population risk function, for a given GCN model with arbitrary parameters and the number of layers (Liao et al., 2021). They do not discuss how to train a model to achieve a small training loss. In contrast, our framework involves the convergence analysis of GCN/Graph Transformers using SGD on a class of target functions and the generalization gap with the trained model. The zero generalization we achieve is zero population risk, which means the learned model from the training is guaranteed to have the desired generalization on the testing data. The model recovery framework (Zhang et al., 2020b) requires a tensor initialization to locate the initial parameter close to the ground truth weight. For the NTK (Du et al., 2019) framework, they need an impractical condition of an infinitely wide network to linearize the model around the random initialization.

Then, we compare existing works on Transformers. As far as we know, the state-of-the-art generalization analysis on other Transformers (Li et al., 2023a; Tarzanagh et al., 2023; Oymak et al., 2023; Tian et al., 2023) did not consider any graph-based labelling function and trainable positional encoding, which are crucial and necessary for node classification tasks. However, we cover these in the formulation and provide the training dynamics and generalization analysis accordingly.

