# OpenReview forum: "What Improves the Generalization of Graph Transformer? A Theoretical Dive into Self-attention and Positional Encoding"
_ICLR.cc/2024/Conference — Submitted to ICLR 2024_

### Official Review · Reviewer_SSQZ · 2023-10-30

**Soundness:** 3 good
**Presentation:** 3 good
**Contribution:** 3 good
**Rating:** 8
**Confidence:** 2

**Summary:**

The paper theoretically investigates the generalizability of Graph Transformers (GTs) by studying the convergence of shallow GT model on a semi-supervised node classification setup where node labels are determined by discriminative patterns in node features and majority voting within core neighborhoods. Interesting findings include 1) larger fraction of discriminative nodes and smaller confusion ratio improves sample complexity, 2) self-attention helps GTs outperform graph convolutional networks (GCNs) by essentially reducing the number of training samples and iterations needed to achieve zero generalization error, and 3) learnable positional encoding in GTs improves generalizability by promoting core neighborhoods. Experiments on synthetic and real-world data verify these theoretical insights.

**Strengths:**

- [S1] **Theoretical novelty and contribution is clear.** The paper is the first to propose a framework for analyzing the generalizability of graph Transformers, and the analysis using the semi-supervised binary node classification setup is very interesting.
- [S2] **Empirical results well support the theoretical claims.**

**Weaknesses:**

- [W1] **It is unclear whether the theoretical claims can be generalized to real-world scenarios and datasets given the restrictive assumption made under the presented results.**
  - In particular, Assumption 1 states that node labels follow a majority vote of discriminative patterns in core neighborhood, which is somewhat similar to network homophily. To verify this assumption on real-world datasets, Appendix E.1 provides additional analysis on the Cora network, which is known to be homophilic [A]. Thus, the question remains: "does the assumption hold for heterophilic networks as well?" Performing similar analyses on additional networks such as Actor and PascalVOC-SP-1G used in Section 5.2 would provide further support towards verifying this assumption.
  - Similarly, the data model assumes that the data is balanced where discriminative and non-discriminative nodes are distributed uniformly across the graph. Is this also a reasonable assumption to make in various real-world networks?

- [W2] **The proof sketch in Subsection 4.6 is not so intuitive and hard to follow.**
  - Considering that the technical contribution of the paper is mostly theoretical, it would be great if the proof sketch stated what each lemma is arguing for intuitively, a rough picture of how each is proven, and how the Lemmas altogether culminate towards Theorem 4.1. The current proof sketch seems to be missing a significant portion that connects the lemmas to the main theorem, which makes the claims presented above somewhat unconvincing on their own.
  - Several questions that could be clarified in the proof sketch are: "Why do we use a two-phase-mini-batch SGD where we first solely train $W_O$?" and "How to the gradient updates of the weights $W_O$, $W_Q$, $W_K$, $W_V$, and $b$ roughly lead to the generalization guarantees in Theorems 4.1-4.3?"

[A] Pei et al., Geom-GCN: Geometric Graph Convolutional Networks. (ICLR 2020)

**Questions:**

- [Q1] In Figure 8, why was the "Uniform sampling" method that samples nodes across the whole graph for feature aggregation chosen as a baseline? I don't think I fully understand what this baseline is hoping to convey.
- [Q2] Also in Figure 8, where in the plot do we find that SPD-based PE "correctly reflects the homophily, heterophily, and long-range dependency of these three datasets"? Is it from where the performance peak of PE-based sampling is located along the x-axis?

There are a number of typos that could use some additional proofreading:
- Page 4: "We assme $\kappa\geq 4\tau$" -> "We assume $\kappa\geq 4\tau$"
- Figure 9 (x-axis title): "Fraction of labeld nodes" -> "labeled nodes"
- The legend on the Left plot of Figure 9 is different from the other two.
- Middle plot of Figure 9 has an additional y-axis.

---

> ### Author Response · Authors · 2023-11-21
> **Response to Reviewer SSQZ Part I**
>
> We thank the reviewer for taking the time to evaluate our manuscript and provide valuable suggestions. Our responses are as follows.
>
> **Q1 ([W1] part 1 by the reviewer)**: "Does the assumption hold for heterophilic networks as well? Performing similar analyses on additional networks such as Actor and PascalVOC-SP-1G used in Section 5.2 would provide further support towards verifying this assumption.”
>
> **A1**: Thank you for this question and the suggestions. **We would like to clarify that our Assumption 1 can be met by some heterophilous and long-range graph datasets. We also want to illustrate that our formulation of the core neighborhood is different from homophily, heterophily, and long-range dependency.** Homophily and heterophily are defined by the fraction of edges that connect nodes from the same class. If such fraction is close to $1$, then it is a homophilous graph. Otherwise, it is a heterophilous graph. Long range graphs are graphs where long-range information is needed for the learning tasks. Our definition of the core neighborhood is the neighborhood where the average winning margin between class-relevant and confusion nodes is the largest.
>
> **We performed similar experiments on PubMed (homophilous graph), Actor (heterophilous graph), and PascalVOC-SP-1G (long range graph) in Section A.1 (It is Section E.1 in the first version).** The experiments include (1) the existence of discriminative nodes for each dataset (2) probing the core distance and justifying that the winning margin on the core distance $\Delta_n(z_m)>0$ (Assumption 1).
>
> To show the existence of discriminative nodes, we first generate Figures [10](https://imgur.com/z2VKqpA), [11](https://imgur.com/3qbJ3zE), [12](https://imgur.com/ySSCagL), and [13](https://imgur.com/W8pagMX) on Cora, PubMed, Actor, and PascalVOC-SP-1G to illustrate that node features of each class can be represented in a low dimension. Then, we use the averaged low-dimensional representation of nodes in each class to identify one discriminative pattern for each class. **Tables [2](https://imgur.com/IddhDqN), [3](https://imgur.com/BpC9B98), [4](https://imgur.com/htl1yWQ), and [5](https://imgur.com/AvM7Bay) indicate that high fractions of nodes are close to each discriminative pattern, which verifies the assumption on the discriminative/non-discriminative nodes.**
>
> To investigate the core distance of each dataset, we compute the fraction of nodes of which the label is aligned with the majority vote of the discriminative nodes in the distance-$z$ neighborhood. To extend the definition from binary classification in our formulation to multi-classification tasks, we use the average number of confusion nodes per class in the distance-$z$ neighborhood as $|\mathcal{D}\_{hash}^n\cap\mathcal{N}_z^n|$ ("hash" stands for #), the number of confusion nodes in the distance-$z$ neighborhood of node $n$. Figure [14](https://imgur.com/A4uhaZ0) shows the value of normalized $\bar{\Delta}(z)$ for $z=1,2,\cdots,12$, where $\bar{\Delta}(z)$ is divided by $|\mathcal{N}_z^n|$ to control the gap of different numbers of nodes in different neighborhoods. **The empirical result indicates that the core distances of the four datasets are all 1 given the largest $\bar{\Delta}(z)$ at $z=1$.**
>
>
> **We also find the difference of $\bar{\Delta}(z)$ value in distance-$z$ neighborhoods with different graph properties.** (1) For the heterophilous graph Actor, the difference between different $\bar{\Delta}(z)$ is very small. (2) For the long-range graph PascalVOC-SP-1G, the value of $\bar{\Delta}(z)$ when $z=12$ is also remarkable. (3) for the homophilous graph PubMed and Cora, the value of $\bar{\Delta}(z)$ when $z=1$ is much larger than the value of $\bar{\Delta}(z)$ when $z$ is large. These show the difference between the four datasets.
>
> We then compute the fraction of nodes that satisfy the winning majority vote, i.e., $\Delta_n(z)>0$. **Table [6](https://imgur.com/c7giuaj) shows that at least 85% nodes of the whole graph satisfy that the majority vote wins, which verifies Assumption 1.**

---

> > ### Author Response · Authors · 2023-11-21
> > **Response to Reviewer SSQZ Part II**
> >
> > **Q2 ([W1] part 2 by the reviewer)**: “Similarly, the data model assumes that the data is balanced where discriminative and non-discriminative nodes are distributed uniformly across the graph. Is this also a reasonable assumption to make in various real-world networks?”
> >
> > **A2**: We thank the reviewer for asking this question. After checking the paper, we noticed that the explanation that “discriminative and non-discriminative nodes are distributed uniformly across the graph” is confusing and does not accurately describe our model. We apologize for this error. **What we want to define for the “balanced dataset” is that the numbers of positive and negative LABELS in the graph are close to each other.** This serves as a  commonly used assumption in the theoretical analysis of neural networks [Shi et al., 2022, Brutzkus et al., 2022, Karp et al., 2021, Li et al., 2023] using the feature learning framework. This condition is also assumed in a recent work [Zhang et al., 2023]  in ICLR 2023 on GNN training using feature learning analysis. In our revision, we rewrite this sentence into “​​**We assume the dataset is balanced, i.e., the gap between the numbers of positive and negative labels is at most $O(\sqrt{N})$.**” Here, $N$ is the total number of nodes in the graph. We are sorry that the previous footnote here may be confusing, and we have moved it to another place.
> >
> > Brutzkus et al., “An Optimization and Generalization Analysis for Max-Pooling Networks
> > ”, UAI 2020.
> >
> > Karp et al., “Local Signal Adaptivity: Provable Feature Learning in Neural Networks Beyond Kernels”, Neurips 2022
> >
> > Shi et al., “A theoretical analysis on feature learning in neural networks: Emergence from inputs and advantage over fixed features”, ICLR 2022.
> >
> > Li et al., “A theoretical understanding of shallow vision transformers: Learning, generalization, and sample complexity”, ICLR 2023.
> >
> > Zhang et al., “Joint Edge-Model Sparse Learning is Provably Efficient for Graph Neural Networks”, ICLR 2023.
> >
> >
> >
> > **Q3 ([W2] part 1 by the reviewer)**: “it would be great if the proof sketch stated what each lemma is arguing for intuitively, a rough picture of how each is proven, and how the Lemmas altogether culminate towards Theorem 4.1.”
> >
> > **A3**: We apologize for the inconvenience of understanding the proof sketch and thank the reviewer for the suggestion. We have made revisions to Section 4.6 in our submission.
> >
> > **For what each lemma conveys**, we state that “By supportive Lemmas 8 and 9, we first characterize two groups of neurons that respectively activate the self-attention layer output of $\mu_1$ and $\mu_2$ nodes from initialization. Then, Lemma 4 shows that the neurons of $W_O$ in these two groups grow along the two directions of the discriminative pattern embeddings. Lemma 7 indicates that the updates of $W_V$ consist of neuron weights from these two groups. Meanwhile, Lemma 5 states that $W_Q$ and $W_K$ evolve to promote the magnitude of query and key embeddings of discriminative nodes. Lemma 6 depicts the training trajectory of the learning parameter of PE that emphasizes the core neighborhood.”
> >
> > **For how each lemma is proven**, we briefly state that “we make the proof of each lemma tractable by studying gradient growth per distance-$z$ neighborhood for each $z$ rather than directly characterizing the gradient growth over the whole graph. Such a technique enables a dynamic tracking of per-parameter gradient updates. As a novel aspect, we prove Lemma 6 by showing that its most significant gradient component is proportional to the average winning margin in the core neighborhood.” in Section 4.6 due to limited space. We add a proof sketch of each lemma at the beginning of Appendix D.
> >
> > **For how these lemmas culminate towards Theorem 4.1**, we state that “First, Lemma 5 and 6 collaborate to illustrate that attention weights correlated with class-relevant nodes become dominant when $\eta t=\Theta(1)$. Second, we compute the network output by Lemmas 4 and 7. By enforcing the output to be either $>1$ or $<-1$ for zero Hinge loss, we derive the sample complexity bound and the required number of iterations by concentration bounds.”

---

> > > ### Author Response · Authors · 2023-11-21
> > > **Response to Reviewer SSQZ Part III**
> > >
> > > **Q4 ([W2] part 2 by the reviewer)**: “"Why do we use a two-phase-mini-batch SGD where we first solely train W_O
> > > ?" and "How to the gradient updates of the weights W_O, W_V, W_Q, W_V, and b roughly lead to the generalization guarantees in Theorems 4.1-4.3?"”
> > >
> > > **A4**: **For the first question on two-phase-mini-batch SGD, we have deleted such a requirement in our revision**. Thank you for asking this inspiring question. Now, we find we can improve the proof, and this condition is no longer necessary. This is because we theoretically verify that the vanilla SGD on all parameters, instead of training $W_O$ solely first, can already direct $W_O$ lucky neurons to activate the MLP layer along the training due to a $\gamma_d=\Theta(1)$ fraction of discriminative nodes in the graph (This is also the goal of the extra first phase of the two-phase training in the previous version). We characterize the updates of the lucky neurons to the desired directions in all iterations in the revised Lemma 9.
> > >
> > > **For the second question on how the updates of weights lead to Theorem 4.1-4.3**, we have the following answer. **First, in A3, we have discussed how the conclusion in Theorem 4.1 is developed**. The main idea is to use Lemmas 5 and 6 to infer dominant attention weights on class-relevant nodes and then use Lemmas 4 and 7 to characterize the required number of samples and iterations for the desired network output of zero Hinge loss. **Second, for the reasoning of Theorem 4.2 and 4.3, we added the discussion in Section 4.6 in the revision**. We state that “When the self-attention layer weights are fixed at $0$ in Theorem 4.2, since that a given core neighborhood and a $\gamma_d=\Theta(1)$ fraction of discriminative nodes still ensures non-trivial attention weights correlated with class-relevant nodes along the training, the updates of $W_O$ and $W_V$ are order-wise the same as Lemmas 4 and 7. Then, we can apply Lemmas 4 and 7 to derive the required number of samples and iterations for zero generalization. Likewise, given a known core neighborhood in Theorem 4.3, the remaining parameters follow the same order-wise update as Lemmas 4, 5, and 7. Hence, Theorems 4.2 and 4.3 can be proved.”
> > >
> > >
> > > **Q5 ([Q1] by the reviewer)**:  “In Figure 8, why was the "Uniform sampling" method that samples nodes across the whole graph for feature aggregation chosen as a baseline? I don't think I fully understand what this baseline is hoping to convey.”
> > >
> > > **A5**: We initially hope to show that the performance of the best PE-based sampling strategy is better than using uniform sampling. To avoid any confusion, we deleted this curve in our revision.
> > >
> > >
> > >
> > > **Q6 ([Q2] by the reviewer)**: “Also in Figure 8, where in the plot do we find that SPD-based PE "correctly reflects the homophily, heterophily, and long-range dependency of these three datasets"? Is it from where the performance peak of PE-based sampling is located along the x-axis?”
> > >
> > > **A6**: Thank you for the question. We realized that our statement that “SPD-based PE also correctly reflects the homophily, heterophily, and long-range dependency of these three datasets” is misleading . Therefore, we deleted this sentence in the revision. **The main message from the performance of PE-based sampling is that (1) the red curves (PE-based sampling) in Figure 8 show different preferences on neighborhoods of the three datasets, indicating the existence of the core neighborhood (2) as $z$ increase, the trend of the red curves matches the trend of entries of the trainable positional encoding parameter $b$, indicating the success of the position encoding.**
> > >
> > > What we meant to have additionally commented is that these three graphs have different patterns of the distance-$z$ neighborhood, and this difference might result from the fact that these graphs are  homophily, heterophily, and long-range dependency, respectively. By different patterns, we mean that the core neighborhood is $z=1$ in Figure 8 left, the impact of all distance $z$ is similar in Figure 8 middle, and both distance 1 and distance 12 have signficant impact in Figure 8 right.
> > >
> > >
> > >
> > > **Q7**: There are a number of typos that could use some additional proofreading:
> > >
> > > **A7**: Thank you for mentioning this problem. We have proofread the paper and made revisions to the submission.
> > >
> > > We would like to clarify that in Figure 9 left, the legend should be that “Red: GT with PE; Green: GT without PE; Blue: GCN”, which is the same as the other two. This shows that the conclusion that GT with PE has a better performance than that without PE and is also better than GCN holds for PubMed dataset.
> > >
> > > The reason that the middle plot of Figure 9 has two y-axis is that GCN performance has a large gap to the Graph Transformer performance with/without PE. If we use one y-axis, the difference between the Graph Transformer performance with and without PE will look very small. To avoid this problem, we draw two y-axis.

---

> ### Comment · Reviewer_SSQZ · 2023-11-23
> **Response to Author Rebuttal by Reviewer SSQZ**
>
> Thank you authors for your commitment into preparing the response. The revision clarifies most of my concerns on whether the assumptions made actually hold for real-world graphs, and also provides a clearer picture of the theoretical contributions. After further consideration, I believe the work makes significant contributions towards characterizing the generalizability of graph Transformers, and thus I raise my score to an accept.

---

> > ### Author Response · Authors · 2023-11-23
> > **Thank you for raising the score to 8**
> >
> > Dear Reviewer SSQZ,
> >
> > Thank you for updating your review and increasing the score. We are encouraged that you acknowledged our contributions both empirically on real-world datasets and theoretically on the generalization analysis.
> >
> > Authors

---

### Official Review · Reviewer_vpyX · 2023-10-30

**Soundness:** 3 good
**Presentation:** 3 good
**Contribution:** 3 good
**Rating:** 6
**Confidence:** 3

**Summary:**

This paper studies the learning and generalization of a shallow graph transformer for semi-supervised node classification with a self-attention layer with relative positional encoding and a two-layer perception. The sample complexity to achieve a zero-generalization error by SGD is provided, which suggests some key factors for improving generalization. Some interesting points for enhancing generalization are also revealed, supported by experimental results on synthetic and real-world benchmarks.

**Strengths:**

This paper is well-motivated and makes a first attempt of studying generalization of graph transformers trained by SGD. Some interesting points have been raised such as the use of graph sampling for attaining better generalization performance.

The paper is well-written and easy to follow. The novelty and contributions are well-organized. This paper develops a novel and extendable feature learning framework for analysing optimization and generalization of graph transformers; this could be of interest to the community.

In addition to the theorems, the remarks provide some interesting comments/insights that would be interesting to the community.

**Weaknesses:**

It is unclear how the generalization obtained for GCNs differs from the existing ones using e.g., Rademacher complexity, algorithmic stability, and/or PAC-Bayesian. Comparisons or discussions should have been provided to make this clearer.

**Questions:**

1.	Given that graph transformers can be represented as GNNs, is it possible to employ existing generalization analysis of GNNs to investigate graph transformer? Why is the proposed analytical framework more suitable for graph transformer compared with the existing ones?
2.	Is the proposed analytical framework applicable to node classifications for heterophily graphs? How about other graph tasks such as graph classification?

---

> ### Author Response · Authors · 2023-11-21
> **Response to Reviewer vpyX Part I**
>
> We thank the reviewer for taking the time to evaluate our manuscript and provide valuable suggestions. Our responses are as follows.
>
> **Q1**: “It is unclear how the generalization obtained for GCNs differs from the existing ones using e.g., Rademacher complexity, algorithmic stability, and/or PAC-Bayesian. Comparisons or discussions should have been provided to make this clearer.”
>
> **A1**: Thank you for the question. **First, our studied problem is different from the listed existing works.** Rademacher complexity [Garg et al.], algorithmic stability [Verma & Zhang], and PAC-Bayesian [Liao et. al.] only focus on the generalization gap, which is the difference between the empirical risk and the population risk function, for a given GCN model with arbitrary parameters and the number of layers [Liao et. al.].  They do not discuss how to train a model to achieve a small training loss. In contrast, our framework involves the convergence analysis of GCN/Graph Transformers using SGD on a class of target functions and the generalization gap with the trained model. The zero generalization we achieve is zero population risk, which means the learned model from the training is guaranteed to have the desired generalization on the testing data.
>
> **We have included such discussions in the revision in Appendix E.6.**
>
> Liao et al. “A PAC-Bayesian Approach to Generalization Bounds for Graph Neural Networks. ”ICLR 2021.
>
> Verma & Zhang. “Stability and Generalization of Graph Convolutional Neural Networks. ”KDD 2019.
>
> Garg et al. “ Generalization and representational limits of graph neural networks. ”ICML 2020.
>
>
>
> **Q2**: “Given that graph transformers can be represented as GNNs, is it possible to employ existing generalization analysis of GNNs to investigate graph transformer? Why is the proposed analytical framework more suitable for graph transformer compared with the existing ones?”
>
> **A2**: This is a great question. **For your first question, the answer is yes, and we can use existing generalization analysis of GNNs to study Graph Transformers, but with limitations.** As discussed in A1, applying existing techniques, such as Rademacher complexity, algorithmic stability, and PAC-Bayesian could derive a generalization gap based on the bounded norm of the model weights, node features, and Lipstchitz constant of the network. Other existing generalization analyses of GNNs include model estimation [Zhang et al., 2020] and NTK [Du et al., 2019]. However, these methods have intrinsic limitations even if they are extended to study Graph Transformers. For Rademacher complexity, algorithmic stability, and PAC-Bayesian,  they do not discuss how to train a model to achieve a small generalization gap. For the model recovery framework, they require a tensor initialization to locate the initial parameter close to the ground truth weight. For NTK, they need an impractical condition of an extremely overparameterized network to linearize the model around the random initialization, and such linearization does not explain the practical success of nonlinear neural networks.
>
> Therefore, **for your second question, the reason is that our proposed analytical framework can investigate the training dynamics by quantifying the feature growth per iteration for attention-based networks.** With such a framework, we are able to theoretically characterize the mechanism that (1) the self-attention layer converges to a sparse attention map that promotes learning class-relevant features (Lemma 1). This leads to a better sample complexity and generalization performance for Graph Transformers compared to GCNs for some graph datasets (Theorem 4.2). (2) The positional encoding attracts attention to the core neighborhood (Lemma 2), which explains the superiority of training with positional encoding over without positional encoding when nodes are properly sampled (Theorem 4.3).
>
> **We have added such discussion in the revision in Appendix E.6.** We also compare our framework and existing works on generalization analysis of other Transformers. Our new contribution is that we consider graph structure and trainable positional encoding in the problem formulation and provide the training dynamics and generalization analysis on this more challenging model.
>
> Zhang et al. “Fast learning of graph neural networks with guaranteed generalizability: one-hidden-layer case”, ICML 2020.
>
> Du et al., “Graph Neural Tangent Kernel: Fusing Graph Neural Networks with Graph Kernels”, Neurips 2019.

---

> > ### Author Response · Authors · 2023-11-21
> > **Response to Reviewer vpyX Part II**
> >
> > **Q3**: “Is the proposed analytical framework applicable to node classifications for heterophily graphs? How about other graph tasks such as graph classification?”
> >
> > **A3**: **The proposed analytical framework can be applied to node classification for heterophilous graphs that satisfy our proposed data assumption in Section 4.2.** To be specific, as long as the heterophilous graph has a neighrborhood that a majority vote of nodes in this neighborhood can decide each label, even if the winning margin is very small, such a heterophilous graph satisfies our data assumptions. We then verify this holds in practice and use the dataset Actor in Appendix A1 as an example. We first use the averaged low-dimensional representation (by principal component analysis) of nodes in each class to identify one discriminative pattern for each class. Table 4 indicates that high fractions of nodes are close to each discriminative pattern, which verifies the assumption on the discriminative/non-discriminative nodes. Figure 14 shows the value of normalized $\bar{\Delta}(z)$ for $z=1,2,\cdots,12$, indicating that the core distance $z_m=1$ for Actor. We then compute the fraction of nodes that satisfying the winning majority vote, i.e., $\Delta_n(z)>0$. Table 6 shows that at least $85%$ nodes of the whole graph satisfy that the majority vote wins, which verifies Assumption 1. By the findings above, we can conclude that some common practical datasets satisfy our data assumption, and the proposed analytical framework is generic.
> >
> >
> > For the extension to graph classification tasks, **we provide a discussion in Section E.4 in the Appendix.** Our analytical framework can be applied to study the generalization of Graph Transformers on graph classification tasks, where the graph labels are determined by a non-trivial mechanism. To be more specific, we assume the existence of the core neighborhood of some central nodes in the graph to decide the ground truth graph label. This is motivated by graph classification on social networks where the connections between the central person and other people in the graph decide the graph label. **Then, we can formulate the graph classification problem as the node classification based on subgraphs.** Our technique can be used to handle this problem.

---

### Official Review · Reviewer_j2s8 · 2023-11-01

**Soundness:** 3 good
**Presentation:** 3 good
**Contribution:** 3 good
**Rating:** 6
**Confidence:** 3

**Summary:**

This article conducts an in-depth theoretical research on the application of shallow Graph Transformers in semi-supervised node classification, yielding several intriguing conclusions. In particular, the author provides a well-reasoned justification for the improvement in generalization observed in shallow Graph Transformers. This enhancement is attributed to several key factors:  larger proportion of discriminative nodes, a more decisive consensus among these nodes, a smaller percentage of incorrect labels, and reduced errors in both the initial model and node patterns. Additionally, the article theoretically investigates the mechanisms through which self-attention leads to superior performance of Graph Transformers compared to Graph Convolutional Networks (GCNs). It also explores how positional embedding contributes to improved generalization by acting upon the core neighborhood.

**Strengths:**

1. This article has a rigorous theoretical derivation and proof. Considering that this is an article that examines the factors affecting GT performance from a theoretical perspective, this rigour is commendable.
2. This article is valuable from a theoretical perspective in its analysis of the mechanisms by which discriminative nodes influence the ability of GTs to generalize.

**Weaknesses:**

1. In the introduction, the article states that "where nodes with discriminative patterns in some so-called core neighborhoods determine the labels by a majority vote, while non-discriminative nodes do not affect the labels." This sentence is a bit confusing for me. What are nodes with discriminative patterns, and how are so-called core neighborhoods defined, since a larger fraction of discriminative nodes improves generalization performance is an important contribution of this paper, perhaps a more detailed explanation or a citation is needed here.

2. It seems to be common knowledge that a smaller fraction of erroneous labels, and smaller errors in the initial model mentioned in the paper affect performance, and there doesn't seem to be a strong incentive to adopt a theory to analyze the mechanism in depth.

3. The article focuses on the factors affecting the performance of the shallow Graph Transformer on the semi-supervised node classification task, and it is unclear whether the conclusions can be generalized to more complex frameworks, or to other downstream tasks.

**Questions:**

Please see the questions in the Weaknesses section.

---

> ### Author Response · Authors · 2023-11-21
> **Response to Reviewer j2s8 Part I**
>
> We thank the reviewer for taking the time to evaluate our manuscript and provide valuable suggestions. Our responses are as follows.
>
> **Q1**: “In the introduction, what are nodes with discriminative patterns, non-discriminative patterns, and core neighborhoods?”
>
> **A1**: We first apologize for the confusion by this sentence. The definition of discriminative patterns is introduced at the beginning of Section 4.2. We assign each node of the graph a pattern $\mu_j, j=1,2,\cdots, M$ with a noise. The nodes of $\mu_1$ or $\mu_2$ are discriminative nodes, while nodes of $\mu_j, 3\leq j\leq M$ are non-discriminative nodes. We assume the existence of a distance-z neighborhood where the ground truth label of every node is decided by such neighborhood. The node is class 1 if there are more nodes with $\mu_1$ patterns than that with $\mu_2$ patterns in the distance-z neighborhood, and the node is class 2 if there are more nodes with $\mu_2$ patterns than that with $\mu_1$ patterns. Such a neighborhood is called the core neighborhood.
>
> For the referred sentence in the Introduction section, to make the presentation concise, we have revised it to “**We focus on a semi-supervised binary node classification problem on structured graph data, where each node feature corresponds to either a discriminative or a non-discriminative pattern, and each ground truth node label is determined by the dominate discriminative pattern in the core neighborhood. **”
>
>
> **Q2**: “It seems to be common knowledge that a smaller fraction of erroneous labels, and smaller errors in the initial model mentioned in the paper affect performance, and there doesn't seem to be a strong incentive to adopt a theory to analyze the mechanism in depth.”
>
> **A2**: Thank you for raising this question.  **We realize this statement can be confusing. What we should have meant is that this paper provides a formal theoretical characterization of how these factors affect generalization quantitatively.** We show that our sample complexity bound in equation 9 is proportional to $1/\gamma_d^2$ and $(1-\epsilon_S)^2$ where $\gamma_d$ is the fraction of discriminative nodes, and $\epsilon_S$ is the confusion rate. This indicates that the graphs with a larger fraction of discriminative nodes or a clearer-cutting vote among discriminative nodes can have a better generalization performance. When $\gamma_d$ is close to 0 and $\epsilon_S$ is close to $1/2$, i.e., the winning margin of the majority vote in the core neighborhood is small, the sample complexity bound proportional to $1/(\gamma_d-(\sigma+\tau))^2$ is large, which means the generalization is sensitive to the error in the initial model. A good pre-trained model is desired in this case. Our result also characterizes that the fraction of erroneous labels $p$ makes the sample complexity and the required number of iterations increase rapidly in an order of $(1-2p)^{-1/2}$ when $p$ is very close to $1/2$.
>
> To address such concern, we have revised the related sentences in the abstract as “**This paper provides the quantitative characterization of the sample complexity and number of iterations for convergence dependent on the fraction of discriminative nodes, the dominant patterns, the fraction of erroneous labels, and the initial model errors.** “

---

> > ### Author Response · Authors · 2023-11-21
> > **Response to Reviewer j2s8 Part II**
> >
> > **Q3**: “It is unclear whether the conclusions can be generalized to more complex frameworks, or to other downstream tasks.”
> >
> > **A3**: This is a great question. **The reason why we focus on the theoretical analysis of one-layer Graph Transformers is due to the theoretical challenges of studying Transformers.** We would like to clarify that, **first, our work is the first to theoretically analyze the generalization of Graph Transformers.** As far as we know, existing works on the generalization or training dynamics of GNNs [Maskey et al., 2022, Tang & Liu 2023, Zhang et al., 2023] do not involve architectures with the attention structure.  **Second, existing works on the generalization analysis of other Transformers cannot be easily extended to study Graph Transformers.** Compared to our work, the state-of-the-art generalization analysis on other Transformers [Li et al., 2023, Tarzanagh et al., 2023, Oymak et al., 2023, Tian et al., 2023] don’t consider any graph-based labelling function and trainable positional encoding, which are crucial and necessary for node classification tasks. **Third, despite one layer, the architecture we formulate has captured major components in GTs, including trainable self-attention, MLP, and positional encoding. The resulting learning problem is already highly nonconvex.** Since we would like to study the training dynamics of the neural network, the complexity of the optimization and generalization problem scales up with the number of trainable parameters in the network. Given multiple trainable parameters, theoretically studying shallow Graph Transformers is already non-trivial.
> >
> > **Although our theoretical analysis is built on one-layer Graph Transformers, empirical results on multi-layer Graph Transformers are aligned with the theory.** Note that in recent works [Zhang et al., 2022, Wu et al., 2022, Chen et al., 2023, Cai et al., 2023, Shirzad et al., 2023, Luo et al., 2023], the number of Graph Transformer layers to learn datasets PubMed, Actor and PascalVOC-SP ranges from 2 to 6. In our empirical experiments on real datasets, we use four-layer Graph Transformers, which are approximately as deep as Graph Transformers proposed recently for node classification.
> >
> > By the way, we provide an extension to other positional encodings and multi-classification tasks in Sections E.2 and E.5 in the Appendix to show the versatility of our analytical framework.
> >
> > **For the extension to other downstream tasks, we provide a discussion on graph classification in Section E.4 in the Appendix.** Our analytical framework can be applied to study the generalization of Graph Transformers on graph classification tasks. To be more specific, we assume the existence of the core neighborhood of some central nodes in each graph to decide the ground truth graph label. This is motivated by graph classification on social networks where the connections between the central person and other people in the graph decide the graph label. Then, we can apply the generalization analysis of node classification based on the core neighborhood to the graph classification problem. Our technique can be used to handle this problem.
> >
> > Maskey et al., “Generalization analysis of message passing neural networks on large random graphs”, Neurips 2022.
> >
> > Tang & Liu, “Towards understanding the generalization of graph neural networks”, ICML 2023.
> >
> > Zhang et al., “Joint edge-model sparse learning is provably efficient for graph neural networks”, ICLR 2023.
> >
> > Li et al., “A theoretical understanding of shallow vision transformers: Learning, generalization, and sample complexity”, ICLR 2023.
> >
> > Oymak et al., “On the role of attention in prompt-tuning”, ICML 2023.
> >
> > Tarzanagh et al., “Max-margin token selection in attention mechanism”, Neurips 20
> > 23.
> > Tian et al., “Scan and snap: Understanding training dynamics and token composition in 1-layer transformer”, Neurips 2023.
> >
> > Zhang et al., “Hierarchical Graph Transformer with Adaptive Node Sampling”, Neurips 2022.
> >
> > Wu et al., “NodeFormer: A Scalable Graph Structure Learning Transformer for Node Classification”, Neurips 2022.
> >
> > Chen et al., “NAGphormer: A Tokenized Graph Transformer for Node Classification in Large Graphs”, ICLR 2023.
> >
> > Cai et al., “On the Connection Between MPNN and Graph Transformer”, ICML 2023.
> >
> > Shirzad et al., “EXPHORMER: Sparse Transformers for Graphs”, ICML 2023.
> >
> > Luo et al., “Transformers over Directed Acyclic Graphs”, Neurips 2023.

---

### Official Review · Reviewer_Eng5 · 2023-11-23

**Soundness:** 2 fair
**Presentation:** 1 poor
**Contribution:** 1 poor
**Rating:** 1
**Confidence:** 5

**Summary:**

This paper studies generalization capability of a graph transformer (a shallow GCN with a single-head self-attention layer) and present the relevant generalization results in terms of training sample complexity and several restrictive assumptions.

**Strengths:**

Studying generalization of graph neural networks with attention is important and interesting.

**Weaknesses:**

My major concerns are about the restrictive assumptions and lack of comparisons about generalization on transudative/semi-supervised graph neural networks.

First of all, the assumptions are rather restrictive compared to existing theoretical results in transductive/semi-supervised learning.

For example, it is assumed that the dataset is balanced, i.e., the gap between the numbers of positive and negative labels is at most $O(\sqrt{N})$. I can hardly believe this will always happen for useful practical scenarios, and existing theoretical results for transductive learning, such as [1-2] (just name a few), do not require such assumption.

Also, Assumption 1 states that every node admits label consistent with majority voting, which is highly restrictive. It is pretty common in practice, such as noisy graph data with noisy labels, that such assumption may not hold.

The survey on generalization results about transudative/semi-supervised learning using graph neural networks is very sketchy, and a much more detailed comparison to this literature including [1-2] is indeed necessary. In particular, one would be curious how the obtained results in this paper can be compared to the sharp generalization bounds using local Rademacher complexity in [1].

Moreover, I encourage the authors to fix the typos and improve the presentation in the next submission. The current submission has too many unnecessarily marked texts, and most formulas are not punctuated.

[1] Localized complexities for transductive learning, COLT 2014.

[2] Learning Theory Can (Sometimes) Explain Generalisation in Graph Neural Networks, NeurIPS 2021.

**Questions:**

See weakness.

**My feedback after reading the authors' response**

I thank the author for their response for my comments. However, several key issues still remain unsolved. In particular, I mentioned that with noisy graph-structured data which is common in practice, the noisy node labels can easily be inconsistent with majority voting so all the theoretical results in this paper do not hold. The authors' argument using clean graph data does not apply to the noisy graph data concerned here. Also, Table 6 does not show that all the nodes satisfy the winning majority vote, while Assumption 1 requires all the nodes to satisfy the winning majority vote. I consider this as a very serious issue. In addition, the authors have a profound misunderstanding about the previous generalization works including [1-2, a]: these works not only give gap between the population risk and empirical risk, but also exhibit models, such as kernel methods as the training model where the training loss can be reduced to arbitrarily small value under mild conditions. As a result, I respectively disagree the authors' claim that other strong theoretical results are not addressing the same problem as that in this paper.

Due to such concern and the highly restrictive assumption 1 which requires that the label of every node is consistent with majority voting, I decrease my score to 1. I encourage the authors to perform a deeper survey about existing research in transudative/semi-supervised learning.

More comment: after another proofreading of the paper, there are other unjustified assumptions, such as $\kappa> 4 \tau $ (under the title of Sec. 4.2), and it is not clear if it is reasonable or when it can hold. After verifying the proof of the paper, it turns out that the main reason for the authors to claim zero generalization error (in Theorem 4.1-4.3) is due to the highly restrictive Assumption 1: indeed, if every node admits a label by majority voting, then it is not surprising one can enjoy 100\% test accuracy. However, again as shown in Table 6, there are still a fraction (relatively small but nonzero) of nodes not satisfying $\Delta_n(z_m)>0$ required in Assumption 1.

[a] Transductive Rademacher Complexity and its Applications. El-Yaniv et al. Journal of Artificial Intelligence Research 2009.

---

> ### Author Response · Authors · 2023-11-23
> **Response to Reviewer Eng5 Part I**
>
> Thank you for the evaluation. Our responses are as follows.
>
> **Q1**: it is assumed that the dataset is balanced, i.e., the gap between the numbers of positive and negative labels is at most $O(\sqrt{N})$. I can hardly believe this will always happen for useful practical scenarios, and existing theoretical results for transductive learning, such as [1-2] (just name a few), do not require such an assumption.
>
> **A1**: We would like to clarify that **this assumption serves as a  commonly used assumption in the optimization and generalization analysis of neural networks** [Shi et al., 2022, Brutzkus et al., 2022, Karp et al., 2021, Li et al., 2023] using the feature learning framework. This condition is also assumed in a recent work [Zhang et al., 2023]  in ICLR 2023 on GNN training using feature learning analysis. These are published works on top ML/AI conferences in the last three years. Note that $O(\sqrt{N})$ could be in practical datasets. Table [9](https://imgur.com/6o0qjVb) shows that for Cora and Actor, this condition holds since the largest gap between the average number of nodes and the number of any class of nodes is smaller than $O(\sqrt{N})$, where we choose $O(\sqrt{N})=10\sqrt{N}$.
>
> **Second, the references [1,2] focus on a different problem from ours.** [1,2] only study the generalization gap, which is the difference between the empirical risk and the population risk function. They do not discuss how to train a model to achieve a small training loss, and thus needs fewer assumptions.  In contrast, our framework involves not only the generalization gap **but also an extra convergence analysis** of GCN/Graph Transformers using SGD and Hinge loss on a class of target functions. The zero generalization we achieve is zero population risk, which means the learned model from the SGD training is guaranteed to have the desired generalization on the testing data. **Our assumption of a balanced dataset is used for the optimization analysis of Graph Transformers.**   We added [1,2] in Appendix E.6 for a comparison in our revision.
>
> **Third, in our manuscript, we always say that our generalization analysis holds with our data model.** For example, in the Abstract, we state “Focusing on a graph data model with discriminative nodes that determine node labels and non-discriminative nodes that are class-irrelevant, we characterize the sample complexity …”; in the Introduction, we state “We focus on a semi-supervised binary node classification problem on structured graph data, where each node feature corresponds to either a discriminative or a non-discriminative pattern, and each ground truth node label is determined by the dominant discriminative pattern in the core neighborhood.”
>
>
>
> Brutzkus et al., “An Optimization and Generalization Analysis for Max-Pooling Networks
> ”, UAI 2020.
>
> Karp et al., “Local Signal Adaptivity: Provable Feature Learning in Neural Networks Beyond Kernels”, Neurips 2022
>
> Shi et al., “A theoretical analysis on feature learning in neural networks: Emergence from inputs and advantage over fixed features”, ICLR 2022.
>
> Li et al., “A theoretical understanding of shallow vision transformers: Learning, generalization, and sample complexity”, ICLR 2023.
>
> Zhang et al., “Joint Edge-Model Sparse Learning is Provably Efficient for Graph Neural Networks”, ICLR 2023.

---

> > ### Author Response · Authors · 2023-11-23
> > **Response to Reviewer Eng5 Part II**
> >
> > Q2: Also, Assumption 1 states that every node admits label consistent with majority voting, which is highly restrictive. It is pretty common in practice, such as noisy graph data with noisy labels, that such assumption may not hold.
> >
> > A2: **We would like to clarify that our Assumption 1 can be met by some commonly used datasets Cora, PubMed (homophilous), Actor (heterophilous) and PascalVOC-SP-1G (long-range).**
> >
> > **We performed experiments on these four datasets in Section A.1 in the updated version.** The experiments include (1) the existence of discriminative nodes for each dataset (2) probing the core distance and justifying that the winning margin on the core distance $\Delta_n(z_m)>0$ (Assumption 1).
> >
> > To show the existence of discriminative nodes, we first generate Figures [10](https://imgur.com/z2VKqpA), [11](https://imgur.com/3qbJ3zE), [12](https://imgur.com/ySSCagL), and [13](https://imgur.com/W8pagMX) on Cora, PubMed, Actor, and PascalVOC-SP-1G to illustrate that node features of each class can be represented in a low dimension. Then, we use the averaged low-dimensional representation of nodes in each class to identify one discriminative pattern for each class. **Tables [2](https://imgur.com/IddhDqN), [3](https://imgur.com/BpC9B98), [4](https://imgur.com/htl1yWQ), and [5](https://imgur.com/AvM7Bay) indicate that high fractions of nodes are close to each discriminative pattern, which verifies the assumption on the discriminative/non-discriminative nodes.**
> >
> > To investigate the core distance of each dataset, we compute the fraction of nodes of which the label is aligned with the majority vote of the discriminative nodes in the distance-$z$ neighborhood. To extend the definition from binary classification in our formulation to multi-classification tasks, we use the average number of confusion nodes per class in the distance-$z$ neighborhood as $|\mathcal{D}\_{hash}^n\cap\mathcal{N}_z^n|$ ("hash" stands for #), the number of confusion nodes in the distance-$z$ neighborhood of node $n$. Figure [14](https://imgur.com/A4uhaZ0) shows the value of normalized $\bar{\Delta}(z)$ for $z=1,2,\cdots,12$, where $\bar{\Delta}(z)$ is divided by $|\mathcal{N}_z^n|$ to control the gap of different numbers of nodes in different neighborhoods. **The empirical result indicates that the core distances of the four datasets are all 1 given the largest $\bar{\Delta}(z)$ at $z=1$.**
> >
> >
> > **We also find the difference of $\bar{\Delta}(z)$ value in distance-$z$ neighborhoods with different graph properties.** (1) For the heterophilous graph Actor, the difference between different $\bar{\Delta}(z)$ is very small. (2) For the long-range graph PascalVOC-SP-1G, the value of $\bar{\Delta}(z)$ when $z=12$ is also remarkable. (3) for the homophilous graph PubMed and Cora, the value of $\bar{\Delta}(z)$ when $z=1$ is much larger than the value of $\bar{\Delta}(z)$ when $z$ is large. These show the difference between the four datasets.
> >
> > We then compute the fraction of nodes that satisfy the winning majority vote, i.e., $\Delta_n(z)>0$. **Table [6](https://imgur.com/c7giuaj) shows that at least 85% nodes of the whole graph satisfy that the majority vote wins, which verifies Assumption 1.**

---

> > > ### Author Response · Authors · 2023-11-23
> > > **Response to Reviewer Eng5 Part III**
> > >
> > > Q3: The survey on generalization results about transudative/semi-supervised learning using graph neural networks is very sketchy, and a much more detailed comparison to this literature including [1-2] is indeed necessary. In particular, one would be curious how the obtained results in this paper can be compared to the sharp generalization bounds using local Rademacher complexity in [1].
> > >
> > > A3: [1,2] study the generalization gap, which is the difference between the empirical risk and the population risk function. They do not discuss how to train a model to achieve a small training loss, and thus needs fewer assumptions.  In contrast, our framework involves not only the generalization gap **but also an extra convergence analysis** of GCN/Graph Transformers using SGD and Hinge loss on a class of target functions. The zero generalization we achieve is zero population risk, which means the learned model from the SGD training is guaranteed to have the desired generalization on the testing data. **Our assumption of a balanced dataset is used for the optimization analysis of Graph Transformers.**   We added [1,2] in Appendix E.6 for a comparison in our revision.
> > >
> > > Q4: Moreover, I encourage the authors to fix the typos and improve the presentation in the next submission. The current submission has too many unnecessarily marked texts, and most formulas are not punctuated.
> > >
> > > A4: Thank you for the suggestion. The reason we use blue marks is to emphasize the changes we made in the revision compared to our initial submission. We already add punctuation to all formulas.

---

### Author Response · Authors · 2023-11-21
**Summary of revisions**

We thank all the reviewers for their valuable evaluation and suggestions. We aim to study the optimization and generalization of Graph Transformers, which is a challenging problem given very limited theoretical references. We thank the reviewers’ appreciation of our efforts.

We would like to highlight a few things in our response and revision.

1. We added Appendix A.1 to verify the graph data assumption using two homophilous graph datasets Cora and PubMed, one heterophilous graph dataset Actor, and one long-range graph dataset PascalVOC-SP-1G. This is to answer [Q1](https://openreview.net/forum?id=aJl5aK9n7e&noteId=F0Knn0F7fz) of Reviewer SSQZ.

2. We made revisions to Section 4.6 in the paper to elaborate more about what each lemma intuitively conveys, how each lemma is proven, and how lemmas lead to Theorem 4.1, 4.2, and 4.3. This is [A3](https://openreview.net/forum?id=aJl5aK9n7e&noteId=DTDm5zfY68) and [A4](https://openreview.net/forum?id=aJl5aK9n7e&noteId=RfrjMR6dCu) to Reviewer SSQZ.

3. We added Section E.6 to compare our framework and other existing generalization works on GNNs and Transformers. This is related to [Q1](https://openreview.net/forum?id=aJl5aK9n7e&noteId=BrAEOmwjxk) and [Q2](https://openreview.net/forum?id=aJl5aK9n7e&noteId=BrAEOmwjxk) of Reviewer vpyX.

4. We state our technical contribution and extension of our work in [A3](https://openreview.net/forum?id=aJl5aK9n7e&noteId=2WzLjq0Ox6) to Reviewer j2s8.

Many thanks, authors

---

### Author Response · Authors · 2023-11-22
**A kind reminder**

Dear Reviewers,

We extend our heartfelt appreciation for your dedicated review of our paper. Your efforts are deeply valued by us.

As the Author-Reviewer discussion period will end in less than one day, could you please let us know whether our response has addressed your questions? If you have further questions, please don't hesitate to reach out. We are fully committed to providing additional responses during this crucial discussion phase.

Thanks, authors

---

### Meta-Review · Area_Chair_by7z · 2023-12-06

**Metareview:**

This article undertakes theoretical research on the application of shallow Graph Transformers in semi-supervised node classification, yielding intriguing conclusions. The author justifies the observed improvement in generalization with shallow Graph Transformers. However, a reviewer has pointed out a critical issue with the paper's reliance on a highly restrictive Assumption 1. Despite the claims made, Table 6 indicates that none of the datasets in question satisfy this assumption. This significant discrepancy undermines the practical utility of the paper's findings.

Upon verification of the paper's proofs, it becomes evident that the authors assert zero generalization error (in Theorem 4.1-4.3) primarily due to the highly restrictive Assumption 1. Essentially, the claim of 100% test accuracy is unsurprising if every node has a label determined by majority voting. Nevertheless, as shown in Table 6, there exists a fraction, albeit relatively small but nonzero, of nodes that do not satisfy the condition $\Delta_n(z_m)>0$ required by Assumption 1. The assumption, however, mandates that all nodes satisfy $\Delta_n(z_m)>0$."

After discussion with reviewers, I recommend rejection.

**Justification For Why Not Higher Score:**

N/A

**Justification For Why Not Lower Score:**

N/A

---

### Decision · Program_Chairs · 2024-01-16

Reject